# SECURE DISTRIBUTED TRAINING AT SCALE

## ABSTRACT

Some of the hardest problems in deep learning can be solved via pooling together computational resources of many independent parties, as is the case for scientific collaborations and volunteer computing. Unfortunately, any single participant in such systems can jeopardize the entire training run by sending incorrect updates, whether deliberately or by mistake. Training in presence of such peers requires specialized distributed training algorithms with Byzantine tolerance. These algorithms often sacrifice efficiency by introducing redundant communication or passing all updates through a trusted server. As a result, it can be infeasible to apply such algorithms to large-scale distributed deep learning, where models can have billions of parameters. In this work, we propose a novel protocol for secure (Byzantine-tolerant) decentralized training that emphasizes communication efficiency. We rigorously analyze this protocol: in particular, we provide theoretical bounds for its resistance against Byzantine and Sybil attacks and show that it has a marginal communication overhead. To demonstrate its practical effectiveness, we conduct large-scale experiments on image classification and language modeling in presence of Byzantine attackers.

## 1 INTRODUCTION

Many hard scientific problems were solved through collaboration between many nations, groups and individuals. This is especially evident in natural sciences, where researchers formed multinational collaborations to run large-scale experiments and share compute infrastructure (Aad et al., 2012; Ruttley et al., 2017; Abbott et al., 2016). Projects like Folding@home (Beberg et al., 2009) and BOINC (Anderson, 2004) push this trend even further by recruiting volunteers that donate their compute to collectively run computational experiments at an unprecedented scale (Merritt, 2020).

Similar techniques were recently proposed for machine learning. They aim to solve the challenges related to the sheer computational complexity of many machine learning tasks, such as pretraining transformers for NLP (Devlin et al., 2019; Brown et al., 2020; Liu et al., 2019) or learning on huge datasets in vision (Sun et al., 2017; Kolesnikov et al., 2020; Goyal et al., 2021). Recent works propose several systems (Kijsipongse et al., 2018; Ryabinin & Gusev, 2020; Atre et al., 2021; Diskin et al., 2021) that can share the computation across many volunteers that donate the idle time of their computers to train large models on public datasets.

Despite their strengths, volunteer computing systems have so far seen limited practical applications (Kijsipongse et al., 2018). A major roadblock towards the global adoption of these techniques is trust in reliability of each participant. For distributed training, all progress made by the collaboration can be undermined if a single peer sends incorrect outputs due to an error in computation (Smith, 2019) or malicious intent (Tolpegin et al., 2020).

Prior art in decentralized optimization proposed several optimization algorithms that are resistant to such "Byzantine" faults. However, most Byzantine-tolerant training protocols require either passing all updates through a trusted central server or exchanging additional messages that increase the network load by several times (Chen et al., 2018; Rajput et al., 2019). This is a major problem for large-scale distributed deep learning, where hundreds of peers must exchange updates for millions of parameters at regular intervals (Li et al., 2020; Sergeev & Balso, 2018; Shoeybi et al., 2019). Thus, in many practical scenarios, the computation and communication overhead of Byzantine-tolerant algorithms outweighs the potential benefits of collaborating with others.

In this work, we set out to solve this problem by proposing a novel distributed training protocol designed for large-scale deep learning workloads. Our approach combines the scalability and communication efficiency of modern distributed training techniques such as All-Reduce SGD (Sergeev & Balso, 2018) with resilience against Byzantine and Sybil attackers. To achieve this, we leverage distributed system security techniques to verify the integrity of training with minimal overhead that does not depend on the model size. Our protocol does not require trusted peers, operating under the assumption that anyone can be an attacker. Our contributions can be summarized as follows:

- We propose a novel strategy for decentralized Byzantine-tolerant training on data available to all participants, where the extra communication cost does not depend on the number of parameters.

- We rigorously analyze the proposed strategy and prove convergence bounds for convex and non-convex losses with Byzantine attackers. Furthermore, we derive accelerated convergence rates for the same task under realistic assumptions about model gradients.

- Based on the above algorithm, we describe a system that allows multiple parties to train a shared model with zero trust assumptions. We prove that this system is resistant to both Byzantine and Sybil attacks from a computationally constrained attacker.

- We verify the effectiveness of our algorithm through both controlled experiments[1] and actual large-scale training runs. Specifically, we start with ResNet-18 for CIFAR-10 classification and follow up with pretraining ALBERT-large in a setup where almost half of all peers are malicious.

## 2 RELATED WORK

### 2.1 DISTRIBUTED DEEP LEARNING

Nowadays, training neural networks often requires the amount of computation that is infeasible to achieve on any single machine. As a result, one has to train such models on multiple machines using specialized distributed training methods. Most of these methods fall into two groups: in *data-parallel* training, each worker trains the entire model by sampling batches from the training data (Sergeev & Balso, 2018; Goyal et al., 2017); in contrast, *model-parallel* training allocates parts of the model on different workers (Huang et al., 2019; Narayanan et al., 2019; Shoeybi et al., 2019). In this study, we consider only the first group; notably, most model-parallel systems still rely on data parallelism between nodes at the same stage (Rajbhandari et al., 2020; Narayanan et al., 2021).

Usually, data-parallel training consists of two phases: first, each worker computes the gradients over its data; then, all workers aggregate the gradients and run an SGD step. The simplest aggregation strategy is known as Parameter Servers (PS) (Li, 2014; Dean et al., 2012; Recht et al., 2011): one of the servers stores and updates the model parameters, while all others iteratively compute the gradients, send them to the PS, and download the updated parameters. This strategy can be quite efficient with a small number of workers; as it increases, the parameter server eventually becomes unable to handle the load. Gradient compression (Seide et al., 2014; Lin et al., 2018; Mishchenko et al., 2019; Koloskova et al., 2020) or local updates (Zinkevich et al., 2010) can partially alleviate this issue, but it remains a fundamental bottleneck of the approach.

In practice, most distributed training systems leverage All-Reduce (AR) (Goyal et al., 2017; Mikami et al., 2019; You et al., 2020) — a family of collective communication protocols that allow servers to average their data and receive the result on each machine. The resulting method, named All-Reduce SGD (AR-SGD), runs AR on local gradients of each peer to compute the global average. Usually, AR-SGD uses bandwidth-optimal versions of All-Reduce (Sergeev & Balso, 2018; Patarasuk & Yuan, 2009); depending on the exact algorithm, they require each peer to transfer $O(d)$ or $O(d \log n)$ data when averaging a vector of size $d$ across $n$ peers (compared to $O(dn)$ for PS).

### 2.2 BYZANTINE-TOLERANT OPTIMIZATION

Standard distributed training methods are not robust against Byzantine attacks. In the vanilla parallel SGD, one malicious worker can break the convergence of the whole method by shifting the mean of the resulting vector in an arbitrary way. Therefore, the research community invented specialized

---

[1]Source code for the experiments is available at `https://github.com/iclr-paper/BTARD`

algorithms that can train models even in this setup. These algorithms are different in nature and provide an extra layer of complexity on top of distributed training methods described in Section 2.1.

**Parameter-server (PS) based approaches.** Majority of the algorithms designed to be Byzantine-resilient rely on the existence of a trusted parameter-server. In such approaches, the standard mean estimator, e.g., the one used in parallel SGD, is typically substituted by a more robust aggregation rule (Blanchard et al., 2017; Yin et al., 2018; Damaskinos et al., 2019; El Mhamdi et al., 2018; Pillutla et al., 2019). However, recent works show that it is not enough via proposing the special types of Byzantine attacks (Baruch et al., 2019; Xie et al., 2020) and showing that permutation-invariant algorithms cannot converge to any predefined accuracy of the solution (Karimireddy et al., 2020).

Although several approaches aiming to circumvent this issue exist, most of them have significant limitations such as no convergence analysis (Chen et al., 2018; Rajput et al., 2019; Rodríguez-Barroso et al., 2020; Xu & Lyu, 2020), too restrictive assumptions in the analysis (Alistarh et al., 2018; Allen-Zhu et al., 2021; Regatti et al., 2020), or the usage of variance-reduced estimators (Wu et al., 2020), which are known to converge slowly in deep learning applications (Defazio & Bottou, 2019). The only paper without such limitations is (Karimireddy et al., 2020), where the authors propose a new aggregation rule called CENTEREDCLIP, apply it to SGD with client momentum, and prove convergence results for the obtained method in the non-convex case under reasonable assumptions. We provide more details on Byzantine-tolerant PS based approaches in Appendix A.1.1.

**Decentralized approaches** for Byzantine-tolerant optimization are studied only in a few papers. Unfortunately, the known approaches are not well-suited for distributed deep learning since they either rely on full gradient computations (Yang & Bajwa, 2019a;b) or use redundant communications with multiple servers (El-Mhamdi et al., 2020), or require peer-to-peer communication of full vectors at each step (Gupta et al., 2021; Gupta & Vaidya, 2021), which is not scalable, or provide the convergence guarantees that are inferior to non-parallel SGD (Peng et al., 2021), which has prohibitively slow convergence on modern deep learning tasks. Further details on Byzantine-tolerant decentralized approaches are deferred to Appendix A.1.2.

## 2.3 SECURITY IN DISTRIBUTED SYSTEMS

In this work, we circumvent the restrictions of existing Byzantine-tolerant techniques using the following approaches from the field of distributed system security.

**Broadcast channels.** Several key stages of our algorithm require peers to send a certain value to all their groupmates. Since we rely exclusively on peer-to-peer connections, a malicious peer could violate this process by deliberately sending different values to each participant. To protect against this attack, distributed systems can use broadcast channels built on top of the protocol for Byzantine broadcast from Dolev & Strong (1983). This protocol ensures that the peers agree on the same broadcasted value even if some of them are malicious. We provide more details on its guarantees in Appendix A.2.1 and review its communication cost in Appendix B.

**Multi-party random number generators.** To ensure that peers compute gradients honestly, our approach verifies a random subset of all computed gradients. These verifications would not be effective if malicious peers could predict (or influence) whether they are going to be verified. Hence, we need to choose who is going to be checked in such a way that the attackers can neither predict nor influence the random draw. This can be done with a multi-party random number generator (MPRNG) based on a multi-party coin tossing protocol, such as the protocol from Blum (1983). We provide an overview of this protocol in Appendix A.2.2 and review its communication cost in Appendix B.

## 3 METHOD

We consider secure distributed training on public datasets, where each peer can access the entire training data and communicate with any other peer. In this scenario, multiple parties cooperate by combining their computational resources for a single large-scale training run. More precisely, we consider a data-parallel training setup with All-Reduce SGD, as described previously in Section 2.1. We describe our strategy in several stages, progressively moving from the theoretical setup to real-world distributed training:

- Section 3.1 outlines our approach for **B**yzantine-**T**olerant **A**ll-**Red**uce (BTARD).

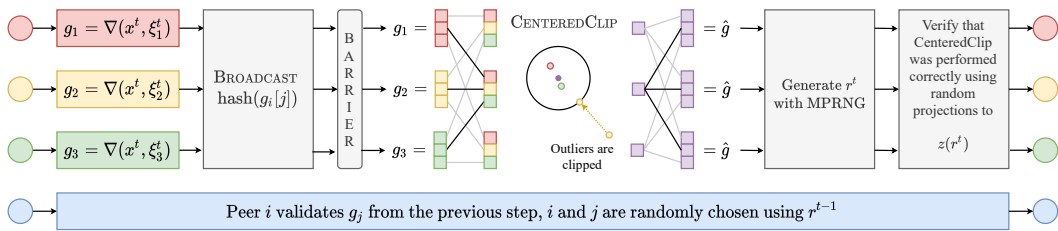

Figure 1: An intuitive scheme demonstrating one step of Byzantine-Tolerant All-Reduce. This is a part of the Algorithm 1 executed between the consecutive SGD steps. Here, $t$ is the step number, $x^t$ is the model weights, and $\xi_i^t$ is a publicly known random seed for sampling a minibatch.

- In Section 3.2, we formulate the underlying optimization problem and derive convergence bounds.
- In Section 3.3, we propose a decentralized system design for distributed training with zero trust.

## 3.1 BYZANTINE-TOLERANT ALL-REDUCE

We assume that some workers can be malicious, i.e., they can arbitrarily deviate from our algorithm: for instance, send arbitrary vectors instead of the stochastic gradients or violate the communication protocol. Such workers are denoted as *Byzantine nodes* or just *Byzantines*. We assume them to be omniscient (Karimireddy et al., 2020) (except for the honest nodes' private keys and the internals of MPRNG) and able to collude with each other. We denote the set of all "good" workers as $\mathcal{G}$ and the set of Byzantine workers as $\mathcal{B}$. We further assume that $\mathcal{B}$ is fixed throughout the optimization process, and less than a half of the nodes are Byzantine: $|\mathcal{B}| \leq \delta n$, where $\delta \in [0, 1/2)$. Finally, we assume that all workers have access to the data defining the objective function, so they can sample minibatches from the full dataset.[2]

We design our algorithm in such a way so that all types of Byzantine faults have limited effect and a chance of being discovered. Together, these properties impose a limit on the total damage that an attacker can do over the entire training run. To control the magnitude of attacks over a single SGD step, we modify All-Reduce with a robust aggregation technique known as CENTEREDCLIP (Karimireddy et al., 2020). Specifically, we use Butterfly All-Reduce[3] (Li et al., 2017) and apply CENTEREDCLIP in parallel to each partition of the gradient vector. We refer to this procedure as BUTTERFLYCLIP (Algorithm 2).

However, Byzantine peers can circumvent robust aggregation by attacking over many iterations. To protect against this, BTARD periodically chooses random peers to serve as *validators*. The validators must recalculate the gradients of other peers and report any discrepancies instead of computing their own gradients. However, such tests are only effective if the attackers cannot predict when they will be validated[4]. To ensure that, we use the multi-party random number generator described in Section 2.3.

After each training step, peers use MPRNG to choose $m$ validators and $m$ peers to validate (each validator checks one peer). As a result, malicious peers cannot predict "safe" iterations before they commit to an attack. Thus, any persistent attacker will eventually be found by an honest validator.

However, since validators can also be malicious, BTARD uses a separate *accuse* procedure to root out false reports. Before each averaging round, peers broadcast the hash of their gradients for that round. These values serve the same purpose as commitments in MPRNG. If peer $i$ accuses peer $j$ of modifying gradients, all other peers must also recalculate $j$'s gradients. If the majority finds that $j$ is innocent, the accusing peer $i$ is banned instead (Hammurabi & Harper, 1904).

The resulting algorithm is resilient to attacks made through incorrect gradients. However, malicious peers may also harm training by violating the CENTEREDCLIP procedure for the portion of gradients they are aggregating. Fortunately, we can design a test through which peers can verify that the vector

---

[2] He et al. (2020) show that it is impossible to achieve any predefined accuracy of the solution without this assumption, i.e., in the heterogeneous case (see discussion in Appendix E.2).

[3] We choose Butterfly All-Reduce so that peers aggregate non-overlapping parts of the gradient vector. This helps to identify the attacker if the gradients are aggregated incorrectly. Jiang et al. (2020) report that Butterfly All-Reduce is near-optimal for distributed training over high-latency networks such as the Internet.

[4] Otherwise, Byzantine peers can simply defer the attack to subsequent steps when they are not validated.

they received was indeed the output of CENTEREDCLIP. To formulate this test, we need to view CENTEREDCLIP as a fixed-point iteration for the equation:

$$\sum_{i=1}^{n} (\vec{g}_i - \vec{x}) \min \left\{ 1, \frac{\tau}{\|\vec{g}_i - \vec{x}\|} \right\} = 0 \tag{1}$$

The workers are not able to test whether (1) holds directly since collecting $\vec{g}_i$ would lead to $O(dn)$ extra communication, defeating the purpose of our algorithm. Instead, workers should use MPRNG to sample a random direction $\vec{z}$ in the space of model gradients. Then, each peer computes the inner product (2) and sends it through the *broadcast channel* (as described in Section 2.3):

$$s_i = \left\langle \vec{z}, (\vec{g}_i - \vec{x}) \min \left\{ 1, \frac{\tau}{\|\vec{g}_i - \vec{x}\|} \right\} \right\rangle \tag{2}$$

Finally, all peers verify that $\sum_{i=1}^{n} s_i = 0$. Similarly to our previous use of MPRNG, all aggregators must commit to their aggregation results before they learn $\vec{z}$. This ensures that a malicious aggregator cannot modify the results in such a way that the difference would be orthogonal to $\vec{z}$ (this and more complex attack vectors are reviewed and analyzed in Appendices C and D.3).

We combine all these procedures in Algorithm 1 (see its intuitive scheme in Figure 1 and a more formal version in Algorithm 6). While gradients $g_i$, random vector $z$, and some other variables are not the same during different steps, we omit the step index $t$ in their notation for brevity. We review communication and computational complexity of the algorithm in Appendix B.

---

**Algorithm 1** BTARD-SGD for peer $i$ (informal)

**Input:** rank $i$, model $x^0$, seed $\xi_i^0$, step count $T$, peer count $n$
1: **for** $t \in 0, \ldots, T-1$ **do**
2:     $g_i = \text{COMPUTEGRADIENTS}(x^t, \xi_i^t)$
3:     $\hat{g} = \text{BUTTERFLYCLIP}(i, g_i)$
4:     $r^t = \text{MPRNG}()$
5:     $z = \text{GETRANDOMVECTOR}(r^t)$
6:     **for** $j \in 1, \ldots, n$ **do**
7:         // $\hat{g}[j]$ is the aggregated part from peer $j$
8:         $\Delta_i^j = (g_i[j] - \hat{g}[j]) \min \left\{ 1, \frac{\tau}{\|g_i[j] - \hat{g}[j]\|_2} \right\}$
9:         **broadcast** $s_i^j = \langle z[j], \Delta_i^j \rangle$
10:    **for** $j \in 1, \ldots, n$ **do**
11:       // We know $\Delta_j^i$ from CENTEREDCLIP
12:       **if** $s_j^i \neq \langle z[j], \Delta_j^i \rangle$ **then**
13:          ACCUSE(i, j, $s_j^i$ is wrong)
14:       **if** $\sum_t^n s_t^j \neq 0$ **then**
15:          // Peer $j$ verified that $s^j$ are correct
16:          ACCUSE(i, j, $\hat{g}[j]$ is wrong)
17:    $x^{t+1} = \text{SGDSTEP}(x^t, \hat{g})$
18:    $\xi_i^{t+1} = \text{hash}(r^t || i)$
19:    **if** $i \in \text{CHOOSEVALIDATORS}(r^t)$ **then**
20:       $j = \text{CHOOSETARGET}(r^t, i)$
21:       VALIDATEPEER($j, x^t, \xi_j^t, c_j, h_j^*, s_j^*$)
22:       // ... instead of computing gradients
23:       //    for step $t+1$
24: **return** $x^T$

---

**Algorithm 2** BUTTERFLYCLIP for peer $i$

**Input:** rank $i$, gradients $g_i \in \mathbb{R}^d$
1: $g_i[1], \ldots, g_i[n] = \text{SPLIT}(g_i, n)$
2: **broadcast** $\forall j, \ h_i^j = \text{hash}(g_i[j])$
3: **send** $\forall j, \ g_i[j] \to \text{peer}_j$
4: **receive** $\forall j, \ g_j[i] \leftarrow \text{peer}_j$
5: **verify** $\forall j, \ \text{hash}(g_j[i]) = h_j^i$
6: $\hat{g}_i = \text{CENTEREDCLIP}(g_1[i], \ldots, g_n[i])$
7: **send** $\forall j, \ \hat{g}_i \to \text{peer}_j$
8: **receive** $\forall j, \ \hat{g}_j \leftarrow \text{peer}_j$
9: **return** $\text{MERGE}(\hat{g}_1, \ldots, \hat{g}_n)$

---

**Algorithm 3** ACCUSE (i, j, allegation)

**Input:** accuser $i$, target $j$
1: $g_j = \text{COMPUTEGRADIENTS}(x^t, \xi_j^t)$
2: **if** $\exists k : (\text{hash}(g_j[k]) \neq h_j^k$
3:    **or** $s_j^k \neq \langle z[k], \Delta_j^k \rangle)$ **or** $\sum_{k=1}^{n} s_k^j \neq 0$
   **then**
4:    VOTEFORBAN($\text{peer}_j$)
5:    // ... and everyone who covered it up
6: **else**
7:    VOTEFORBAN($\text{peer}_i$)
8: **for** $k \in 1, \ldots, n$ **do**
9:    **if** $\text{NUMVOTES}(\text{peer}_k) \geq n/2$ **then**
10:       BAN($\text{peer}_k$)

---

## 3.2 CONVERGENCE ANALYSIS

From the perspective of optimization theory, our task is the expectation minimization problem:

$$\min_{x \in Q \subseteq \mathbb{R}^d} \{ f(x) := \mathbb{E}_{\xi \sim \mathcal{D}} [f(x, \xi)] \} \tag{3}$$

Here, the objective function $f$ is smooth and uniformly lower bounded, $Q \subseteq \mathbb{R}^d$ is a closed convex set of admissible parameters and $\xi$ is the source of stochasticity, such as minibatch indices. We assume that the problem (3) can be solved in a distributed manner, i.e., one can use $n$ workers performing (mini-batched) stochastic gradients calculations in parallel and communicating according to some protocol. For simplicity, we will define the set of workers as $[n] := \{1, 2, \ldots, n\} = \mathcal{G} \sqcup \mathcal{B}$.

There are many ways for Byzantines to affect the training. For example, a malicious peer may perform: **(i) gradient attacks**, where Byzantines modify their $g_i^k$, but otherwise behave normally; **(ii) aggregation attacks**, where Byzantine aggregator returns wrong $\hat{g}_i$ and relies on others to cover it up by misreporting $s_i$; **(iii) reputation attacks** such as frame-up or slander via false $\text{ACCUSE}(i, j, \cdot)$; and finally, **(iv) protocol errors** are any other deviations from the steps of Algorithm 1, e.g. refusing to send any data. We elaborate on each attack type in Appendix C.

For the purpose of this analysis, the latter two attacks can be repelled with an extra policy that allows an active worker to eliminate any other worker at the cost of also being banned. Whenever a benign peer $i$ encounters a protocol error from another peer $j$, it invokes that policy to remove both himself and peer $j$ from training. The design of this policy ensures that every invocation, whether by normal or Byzantine peers, eliminates at least 1 Byzantine peer and at most 1 benign peer. Thus, if a Byzantine minority uses this against benign peers, this will only decrease their relative numbers: $(\delta n - 1)/(n - 2) < \delta$. This leaves us with two attacks that both target the aggregated gradients.

We provide convergence guarantees for variants of BTARD-SGD with $Q = \mathbb{R}^d$ under different sets of assumptions about the function $f$ and its stochastic gradients. Our first two setups assume, that:

**Assumption 3.1.** *There exist such constant $\sigma \geq 0$, $s_0 \in [d]$ that for any set of indices $S = (i_1, \ldots, i_s)$, $1 \leq i_1 < i_2 < \ldots < i_s \leq d$, $s \geq s_0$ stochastic gradient $\nabla f(x, \xi)$ satisfy*

$$\mathbb{E}[\nabla f(x, \xi)] = \nabla f(x), \quad \mathbb{E}\left[\left\|\nabla_{[S]} f(x, \xi) - \nabla_{[S]} f(x)\right\|^2\right] \leq \frac{s\sigma^2}{d}, \tag{4}$$

*where $\nabla_{[S]} f(x, \xi) = (\nabla_{i_1} f(x, \xi), \ldots, \nabla_{i_s} f(x, \xi))^\top$, $\nabla_{[S]} f(x) = (\nabla_{i_1} f(x), \ldots, \nabla_{i_s} f(x))^\top$, and $\nabla f_j(x, \xi), \nabla_j f(x)$ are $j$-th components of $\nabla f(x, \xi)$ and $f(x)$ respectively.*

Here, (4) is an extension of the classical uniformly bounded variance (UBV) assumption (Nemirovski et al., 2009; Ghadimi & Lan, 2012; 2013) ensuring that the noise in all subvectors of large enough dimension has the variance dependent on the ratio between the dimension of the subvector $s$ and the dimension of the full vector $d$. For example, it holds when the noise is isotropic. Moreover, one can relax this assumption to the standard UBV assumption, if blocks for aggregation in BTARD are chosen uniformly at random (see Appendix E.3.1 for further details). In order to further reduce overhead from **Verification 3** in the full Algorithm 6, we also assume that the stochastic gradient distributions have sub-quadratically decreasing tails (see details in Appendix E.3.1).

**Assumption 3.2.** *There exist such constant $\sigma \geq 0$, $s_0 \in [d]$ that for any set of indices $S = (i_1, \ldots, i_s)$, $1 \leq i_1 < i_2 < \ldots < i_s \leq d$, $s \geq s_0$ stochastic gradient $\nabla f(x, \xi)$ satisfy*

$$\mathbb{P}\left\{\left\|\frac{1}{k}\sum_{i=1}^{k}\nabla_{[S]} f(x, \xi_i) - \nabla_{[S]} f(x)\right\|^2 > \frac{ts\sigma^2}{kd}\right\} < \frac{1}{t^2}, \quad \forall t > 0, \tag{5}$$

*where $\xi_1, \ldots, \xi_k$ are i.i.d. samples from $\mathcal{D}$, and $\nabla_{[S]} f(x, \xi)$, $\nabla_{[S]} f(x)$ are defined in As. 3.1.*

Under these assumptions, we derive the following convergence bounds for strongly convex, generally convex, and non-convex objectives (see Table 1). The respective proofs are deferred to Appendix E.4.

Let us briefly discuss the main properties of the derived results. When $\delta = 0$, i.e., there are no Byzantine peers, we recover the tightest known rates for parallel SGD for strongly convex, generally convex, and non-convex objectives with both sets of assumptions. Next, we notice that in all complexity bounds in the known $|\mathcal{B}_k^a|$ case, the term depending on the ratio of Byzantine workers $\delta$ (the third one in all bounds) has better dependence on the accuracy of the solution $\varepsilon$ than the classical variance term (the second one in all bounds). Therefore, for sufficiently small $\varepsilon$, the derived complexity bounds are the same as in the case when there are no Byzantine workers and parallel SGD is used. However, these bounds are obtained under the assumption that all participants know the exact number of attacking Byzantine workers at each iteration, which is not realistic but helps to better adjust clipping parameter $\tau$ in CENTEREDCLIP.

Table 1: Summary of complexity bounds for BTARD-SGD in different scenarios. By complexity we mean the number of iterations sufficient to find such point $\widehat{x}$ that $\mathbb{E}[\|\nabla f(\widehat{x})\|^2] \leq \varepsilon^2$ for non-convex problems and $\mathbb{E}[f(\widehat{x}) - f(x^*)] \leq \varepsilon$ for convex and $\mu$-strongly convex problems (see Def. E.2) with $x^*$ being the solution. Notation: "known $|\mathcal{B}_k^a|$" = the exact number of attacking Byzantine workers at iteration $k$ is known to each participant, $L$ = smoothness constant (see Def. E.1), $\Delta_0 = f(x^0) - f_*$, $f_*$ = uniform lower bound for $f$, $\sigma^2$ = variance parameter from As. 3.1, $n$ = the initial number of peers, $b$ = the initial number of Byzantine workers, $\delta = {}^b/n$, $m$ = number of peers checked at each iteration, $R_0 = \|x^0 - x^*\|$.

| Assumptions | Convexity of $f$ | | |
| --- | --- | --- | --- |
| | Non-convex | Convex | Strongly convex |
| As. 3.1 + As. 3.2 + known $|\mathcal{B}_k^a|$ | $\frac{L\Delta_0}{\varepsilon^2} + \frac{L\Delta_0\sigma^2}{n\varepsilon^4} + \frac{n\delta\sigma^2}{m\varepsilon^2}$ | $\frac{LR_0^2}{\varepsilon} + \frac{\sigma^2 R_0^2}{n\varepsilon^2} + \frac{n\sqrt{\delta}\sigma R_0}{m\varepsilon}$ | $\frac{L}{\mu}\log\frac{\mu R_0^2}{\varepsilon} + \frac{\sigma^2}{n\mu\varepsilon} + \frac{n\sqrt{\delta}\sigma}{m\sqrt{\mu\varepsilon}}$ |
| As. 3.1 + As. 3.2 | $\frac{L\Delta_0}{\varepsilon^2} + \frac{L\Delta_0\sigma^2}{n\varepsilon^4} + \frac{n^2\delta\sigma^2}{m\varepsilon^2}$ | $\frac{LR_0^2}{\varepsilon} + \frac{\sigma^2 R_0^2}{n\varepsilon^2} + \frac{n^2\delta\sigma R_0}{m\varepsilon}$ | $\frac{L}{\mu}\log\frac{\mu R_0^2}{\varepsilon} + \frac{\sigma^2}{n\mu\varepsilon} + \frac{n^2\delta\sigma}{m\sqrt{\mu\varepsilon}}$ |

As for the more general case, the third term is much worse than the corresponding term in the previous setup. Nevertheless, the term that depends on the ratio of Byzantine workers $\delta$ has the same dependence on $\varepsilon$ as in the known $|\mathcal{B}_k^a|$ case. This implies that for sufficiently small $\varepsilon$ the derived complexity bounds are the same as in the case when there are no Byzantine workers and parallel SGD is used. For complete formulations, proofs and other details we refer the reader to Appendix E.3.

So far, all our convergence results rely on As. 3.2, i.e., that the stochastic gradients have not too heavy tails. This assumption holds for many real-world neural networks. However, there are important NLP tasks such as BERT training (Zhang et al., 2020), where the noise in the stochastic gradient has such a heavy noise that As. 3.2 becomes unlrealistic. The third and final setup in our analysis aims to address such heavy-tailed problems with BTARD-CLIPPED-SGD (see full Algorithm 8 in appendix). We analyse the method under the assumption that $\alpha$-th moments of the stochastic gradients are uniformly upper-bounded for some $\alpha \in (1, 2]$. We notice that for $\alpha < 2$ this assumption allows the variance of the stochastic gradient to be unbounded. In this setting, we prove that BTARD-CLIPPED-SGD finds an $\varepsilon$-solution of the convex problem after $\mathcal{O}\left(\varepsilon^{-\alpha/(\alpha-1)}\left(1 + \left({}^{n\sqrt{\delta}}/m\right)^{\alpha/(\alpha-1)}\right)\right)$ iterations when the number of attacking Byzantine peers is known at each iteration and $\mathcal{O}\left(\varepsilon^{-\alpha/(\alpha-1)}\left(1 + \left({}^{n^2\delta^2}/m\right)^{\alpha/(\alpha-1)}\right)\right)$ iterations otherwise. One can find the full statements and complete proofs of our results in Appendix E.

### 3.3 RESISTING SYBIL ATTACKS

The algorithm described in Section 3.1 operates with a pre-defined list of peers that can only decrease in size. However, many real-world scenarios would benefit from new peers joining midway through training. Unfortunately, this exposes the system to Sybil attacks (Douceur, 2002), when a single computationally constrained attacker adopts multiple pseudonymous identities in order to establish a dishonest majority and break the algorithm.

To handle this, one may augment BTARD with a heuristic protocol that dictates how new peers can join. A new participant must prove that it has honestly computed enough gradients over multiple continious iterations before it is allowed to actually contribute to the training. This ensures that the influence of Sybil attackers is proportional to their computing power (see details in Appendix F).

## 4 EXPERIMENTS

### 4.1 CIFAR10 CLASSIFICATION

First, we evaluate our approach with a realistic image-classification workload in controlled conditions. Our setup is a ResNet-18 (He et al., 2015) model trained to solve the CIFAR10 classification task (Krizhevsky et al.). We train the model on 16 peers (each peer processes 8 samples per batch) using the SGD with Nesterov (1983) momentum and the cosine annealing learning rate (Loshchilov & Hutter, 2017). We deliberately use a tuned setup that achieves 93.5% test accuracy in order to measure how Byzantine attacks affect this training outcome.

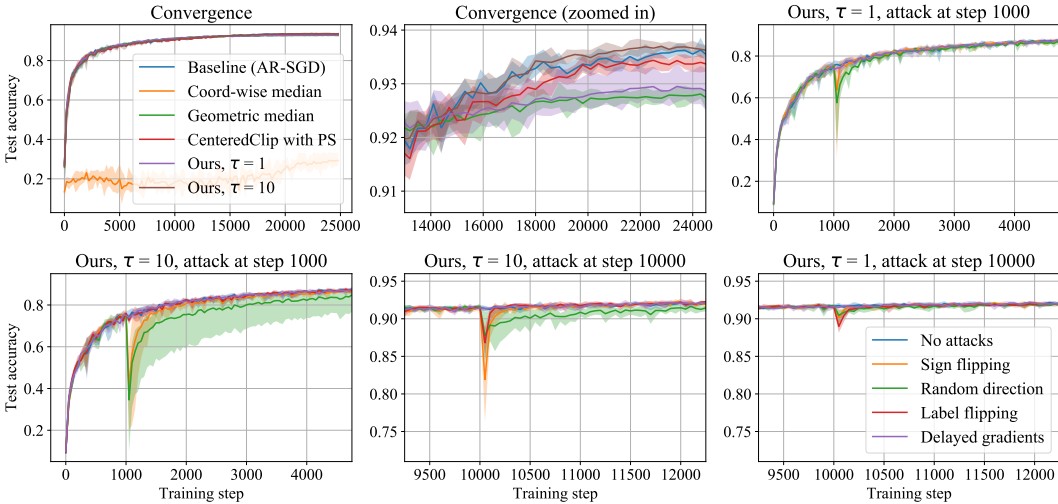

Figure 2: **(Upper-Left, Upper-Middle:)** ResNet-18 test accuracy with different robust aggregation techniques (without attacks). **(Other plots:)** Effectiveness of Byzantine attacks on BTARD-SGD.

We evaluate our method with constant $\tau = 10$ (weaker clipping) and with $\tau = 1$ (stronger clipping). These values were chosen based on the maximal standard deviation of the gradient parts averaged by the workers during normal training, so that almost no vectors are clipped for the weaker clipping and approximately half of the vectors are clipped for the stronger clipping scenario. BTARD randomly selects 1 validator on each step. If the validator happens to be Byzantine, it does not accuse its peers.

We compare our method to the regular All-Reduce without clipping and the baselines that use a trusted parameter server (the original variant of CENTEREDCLIP (Karimireddy et al., 2020), the coordinate-wise and geometric medians). Some other popular robust aggregation techniques are omitted because they were shown to be inferior (Karimireddy et al., 2020). We run all iterative algorithms (such as CENTEREDCLIP) to convergence with $\epsilon = 10^{-6}$, as we have found that limiting the number of iterations can significantly decrease the final model quality (see Fig. 7 in Appendix I.1).

In addition to training convergence, we evaluate our setup in presence of malicious peers. To test pessimistic conditions, we pick a setting where 7 of 16 peers are Byzantine (other setups can be found in Appendix I.1). We experiment with the following attack types:

- SIGN FLIPPING: each attacker sends the opposite of its true gradient.

- RANDOM DIRECTION: all attackers send large vectors pointed at a common random direction.

- LABEL FLIPPING: each attacker computes its gradient based on the cross-entropy loss with flipped labels. For CIFAR-10, we replace label $l \in \{0, ..., 9\}$ with $9 - l$.

- DELAYED GRADIENT: attackers send their real gradients delayed by 1000 steps.

We further amplify the Byzantine gradients from the first two attacks by a large coefficient $\lambda = 1000$ so they would dominate the aggregated gradient if no clipping is used. While in practice such attacks can be identified right away due to the large gradient norms, we deliberately avoid doing that to test our clipping approach. We also evaluate common low-magnitude attacks (Baruch et al., 2019; Xie et al., 2020; Allen-Zhu et al., 2021) in Appendix I.4.

For each experiment configuration, Byzantines behave honestly prior to step $s$, then simultaneously attack on each subsequent step until they are banned (another setup with the Byzantines attacking periodically is reported in Appendix I.1). We consider attacks in two training regions: early stages ($s = 1000$) and closer to convergence ($s = 10,000$). We repeat each experiment 5 times and report the mean and range of the test accuracy during at least 2000 steps after all Byzantines are banned. In our experiments, this usually happened within 150 steps after $s$.

The results are shown in the Fig. 2. Comparing to the All-Reduce baseline, we note that our method does not worsen the speed of convergence. On average, the final test accuracy is 0.6% worse for $\tau = 1$ and 0.1% better for $\tau = 10$. The two most effective attacks (in terms of accuracy) are the

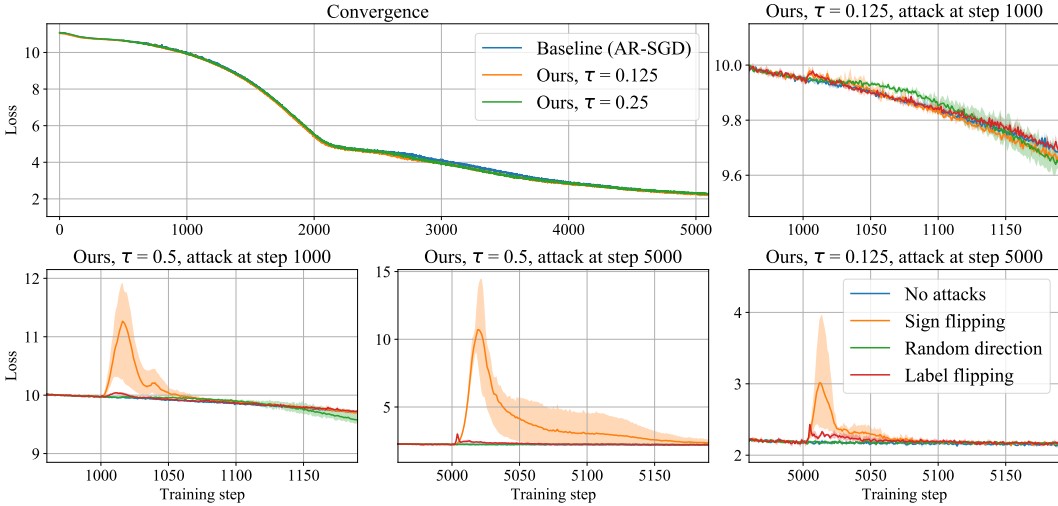

Figure 3: **(Upper-Left:)** ALBERT-large training objective using AR-SGD and BTARD-Clipped-SGD (without attacks). **(Remaining plots:)** Effectiveness of Byzantine attacks on BTARD.

random direction and sign flipping. The effect of label flipping is smaller, and the effect of delayed gradients is almost undetectable.

## 4.2 PRE-TRAINING TRANSFORMERS

For our second experiment, we choose a more compute-intensive and hyperparameter-sensitive model with adaptive optimizers to demonstrate that our approach may be applied to models that are commonly used in distributed training scenarios. Our setup is pre-training the ALBERT-large model (Lan et al., 2019) on the Wikitext 103 dataset (Merity et al., 2017) using LAMB (You et al., 2020). Since the original ALBERT setup uses gradient clipping, we use BTARD-CLIPPED-SGD (see Alg. 8 in Appendix). We train the model on 16 machines that jointly accumulate 4096 samples for every batch. Similarly to the previous section, we evaluate two configurations with $\tau = 0.5$ (weaker clipping) and $\tau = 0.125$ respectively, in addition to an All-Reduce baseline. To evaluate Byzantine tolerance, we also use 7 out of 16 Byzantine workers, 1 validator and two attack regions: $s = 1000$ and $s = 5000$. We omit reporting of the delayed gradient attack as we have found it completely ineffective. The full configuration of this experiment is provided in Appendix H.

The results shown in Figure 3 demonstrate a pattern similar to the previous section. During normal training, both $\tau$ values had no significant effect on the training progress, reaching 1.3% larger loss in the worst case. However, $\tau = 0.125$ shows significantly faster recovery from all three attacks. One important observation from these experiments is that while some attacks significantly increase the loss function, the model returns to the previous loss value much faster than it takes to reach the same loss when training from scratch. We further study the computation overhead of BTARD in this setup in Appendix I.2 and provide the experiments with a larger number of peers in Appendix I.3.

## 5 CONCLUSION

In this work, we formulated BTARD-SGD — a Byzantine-tolerant training strategy for large neural networks. We verified its robustness and effectiveness through rigorous theoretical analysis and large-scale distributed training experiments. While our research is mostly algorithmical, it can open new opportunities in many deep learning applications.

Perhaps the most important one is making it possible to train large neural networks in a cooperative manner. BTARD-SGD could allow small research groups to host open cooperative training projects where the training hardware is crowdsourced by volunteers around the world. Alternatively, a group of small companies could collectively compete with larger corporations by combining their compute clusters. While these applications also require engineering effort to become practical, our algorithm ensures that they can run securely without the need to carefully screen every potential participant.

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

# Supplementary Material

## Table of contents

## A  ADDITIONAL RELATED WORK

### A.1  BYZANTINE-TOLERANT OPTIMIZATION: ADDITIONAL DETAILS

In this section, we provide extra details on the related work discussed in Section 2.2. The summary of complexity results is presented in Table 2.

#### A.1.1  PARAMETER-SERVER (PS) BASED APPROACHES

There is a quite large number of papers on Byzantine-tolerant optimization that aim to robustify parallel SGD in the case when a trusted parameter-server (PS) is available. Since in the classical parallel SGD even one Byzantine worker can break the convergence of the whole method by shifting the mean of the resulting vector in an arbitrary way, it is natural to substitute averaging of the vectors received from the workers by a more robust aggregation rule, e.g., Krum (Blanchard et al., 2017), coordinate-wise median, trimmed median (Yin et al., 2018), Multi-Krum (Damaskinos et al., 2019), Bulyan (El Mhamdi et al., 2018), geometric median (Pillutla et al., 2019). However, all these methods were shown to be brittle and not robust to special types of Byzantine attacks (Baruch et al., 2019; Xie et al., 2020; Karimireddy et al., 2020). Moreover, Karimireddy et al. (2020) show that all permutation-invariant algorithms cannot converge to any predefined accuracy of the solution, meaning that simple application of some aggregation rules on top of SGD does not lead to Byzantine tolerance.

There are several approaches to circumvent this issue. Alistarh et al. (2018) propose BYZANTINESGD and prove the convergence results for convex problems. Allen-Zhu et al. (2021) extend this approach to handle non-convex problems as well. In both papers, the key idea is based on applying the concentration properties of the sums depending on the stochastic gradients as well as iterative removing of Byzantine peers. However, theoretical guarantees from Alistarh et al. (2018); Allen-Zhu et al. (2021) rely on the restrictive assumption that the noise in the stochastic gradients is uniformly bounded with probability 1. Bulusu et al. (2020) propose similar approach to the one from (Allen-Zhu et al., 2021) but analyze their method under more restrictive assumptions (boundedness of the gradient). Next, Wu et al. (2020) propose a Byzantine-tolerant version of parallel SAGA (Defazio et al., 2014), i.e., variance-reduced version of SGD, with geometric median as an aggregation rule — BYRD-SAGA – and prove its convergence for strongly convex objectives. However, the authors do not establish the convergence of BYRD-SAGA to any predefined accuracy of the solution. Moreover, variance-reduced methods are known to converge slowly in deep learning applications (Defazio & Bottou, 2019), which limits the practical utility of BYRD-SAGA. Finally, Karimireddy et al. (2020) propose a new aggregation rule called CENTEREDCLIP, apply it to SGD with client momentum, and prove convergence results for the obtained method in the non-convex case under reasonable assumptions. Alternative lines of work achieve Byzantine-tolerant optimization through redundant computations (Chen et al., 2018; Rajput et al., 2019) or reputation-based approaches (Rodríguez-Barroso et al., 2020; Regatti et al., 2020; Xu & Lyu, 2020). Unfortunately, these papers either do not contain theoretical (non-asymptotic) convergence results for the proposed methods or rely on too restrictive assumptions in the analysis. See more references in the recent survey by Lyu et al. (2020).

#### A.1.2  DECENTRALIZED APPROACHES

Byzantine-tolerant optimization methods for decentralized communication architectures are studied only in a couple of papers. Yang & Bajwa (2019a;b) consider a specific scenario when workers compute full gradients, local loss functions on peers are heterogeneous, and the trimmed coordinate-wise median is used as an aggregation rule. In this setup, the authors prove convergence results in the strongly convex case to some accuracy depending on the heterogeneity level of local loss functions, which is natural in the presence of Byzantine peers. However, these results are not applicable to a wide range of practically important problems where stochastic gradients have to be used. This issue was partially resolved in Peng et al. (2021), where the authors propose a version of GOSSIP SGD applied to the equivalent reformulation of the original problem based on TV-regularization (Ben-Ameur et al., 2013). However, the established convergence results in the strongly convex case do not show any benefits of using communications with other workers in the homogeneous data regime that appears in large-batch training of deep learning models. Li et al. (2019) use the same idea for a parameter-server architecture. Next, there are approaches requiring peer-to-peer communications of full vectors at each step (Gupta et al., 2021; Gupta & Vaidya, 2021), which is not scalable.

Table 2: Summary of the complexity results for Parameter-Server (PS) based and distributed Byzantine-tolerant optimization. By default, columns "Non-convex", "Convex", and "Strongly convex" contain the complexity bounds for $L$-smooth non-convex, convex, and $\mu$-strongly convex problems respectively. By complexity we mean the number of iterations sufficient to find such point $\widehat{x}$ that $\mathbb{E}[\|\nabla f(\widehat{x})\|^2] \le \varepsilon^2$ for non-convex problems and $\mathbb{E}[f(\widehat{x}) - f(x^*)] \le \varepsilon$ for convex and $\mu$-strongly convex problems (see Def. E.2) with $x^*$ being the solution. For simplicity we omit numerical factors, logarithmic terms depending on the parameters of the problem, and factors, quantifying suboptimality of the starting point, i.e., $R_0 = \|x^0 - x^*\|$ and $f(x^0) - \inf_{x\in\mathbb{R}^d} f(x)$. Notation: $\delta = {}^{|\mathcal{B}|}/_n$, $m$ = number of peers checked at each iteration. The results from (Yang & Bajwa, 2019a;b) are not included in the table since they rely on full-gradient computations.

| Non-PS? | Work | Non-convex | Convex | Strongly convex |
|---|---|---|---|---|
| ✗ | (Alistarh et al., 2018)[1,2] | ✗ | $\frac{1}{\varepsilon} + \frac{\sigma^2}{n\varepsilon^2} + \frac{\delta^2\sigma^2}{\varepsilon^2}$ | $\frac{1}{\mu} + \frac{\sigma^2}{n\mu\varepsilon} + \frac{\delta^2\sigma^2}{\mu\varepsilon}$ |
| | (Allen-Zhu et al., 2021)[1,3] | $\frac{1}{n\varepsilon^4} + \frac{\delta^2}{\varepsilon^4}$ | ✗ | ✗ |
| | (Wu et al., 2020)[4] | ✗ | ✗ | $\frac{L^2}{\mu^2}$ [5] |
| | (Karimireddy et al., 2020)[6] | $\frac{1}{\varepsilon^2} + \frac{\sigma^2}{n\varepsilon^4} + \frac{\delta\sigma^2}{\varepsilon^4}$ | ✗ | ✗ |
| ✓ | (Peng et al., 2021)[6,7] | ✗ | ✗ | $\frac{1}{\mu\varepsilon} + \frac{n\sigma^2}{\mu^2\varepsilon} + \frac{\lambda^2 d\overline{N}^2}{\mu^2\varepsilon}$ |
| | **This work**[8] | $\frac{1}{\varepsilon^2} + \frac{\sigma^2}{n\varepsilon^4} + \frac{n\delta\sigma^2}{m\varepsilon^2}$ | $\frac{1}{\varepsilon} + \frac{\sigma^2}{n\varepsilon^2} + \frac{n\sqrt{\delta}\sigma}{m\varepsilon}$ | $\frac{1}{\mu} + \frac{\sigma^2}{n\mu\varepsilon} + \frac{n\sqrt{\delta}\sigma}{m\sqrt{\mu\varepsilon}}$ |
| | **This work**[9] | $\frac{1}{\varepsilon^2} + \frac{\sigma^2}{n\varepsilon^4} + \frac{n^2\delta^2\sigma^2}{m\varepsilon^2}$ | $\frac{1}{\varepsilon} + \frac{\sigma^2}{n\varepsilon^2} + \frac{n^2\delta\sigma}{m\varepsilon}$ | $\frac{1}{\mu} + \frac{\sigma^2}{n\mu\varepsilon} + \frac{n^2\delta\sigma}{m\sqrt{\mu\varepsilon}}$ |
| | **This work**[10] | ✗ | $\left(\frac{G\Lambda_1}{\varepsilon}\right)^{\frac{\alpha}{\alpha-1}}$ | $\left(\frac{G^2\Lambda_1}{\mu\varepsilon}\right)^{\frac{\alpha}{2(\alpha-1)}}$ |
| | **This work**[11] | ✗ | $\left(\frac{G\Lambda_2}{\varepsilon}\right)^{\frac{\alpha}{\alpha-1}}$ | $\left(\frac{G^2\Lambda_2}{\mu\varepsilon}\right)^{\frac{\alpha}{2(\alpha-1)}}$ |

[1] The results are proven under uniformly bounded noise assumption: $\|\nabla f(x, \xi) - \nabla f(x)\| \le \sigma$ for all $x$ and $\xi$. High-probability guarantees are established, i.e., it is proven that with probability at least $1 - \beta$ algorithms from (Alistarh et al., 2018) find $\hat{x}$ such that $f(\hat{x}) - f(x^*) \le \varepsilon$ and algorithms from (Allen-Zhu et al., 2021) find $\hat{x}$ such that $\|\nabla f(\hat{x})\| \le \varepsilon$.

[2] Dependencies on $\beta$ are logarithmic and, therefore, omitted. Optimization problems is assumed to be define on a bounded set, the rates depends on the diameter of this set.

[3] The results are derived for the case $\sigma = 1$. Allen-Zhu et al. (2021) also derive convergence guarantees for finding second-order stationary points.

[4] Wu et al. (2020) consider finite-sum case of (3), i.e., $f(x) = \frac{1}{N}\sum_{j=1}^N f(x, j)$. The results are derived under uniformly bounded variance assumption: $\mathbb{E}_j[\|\nabla f(x, j) - \nabla f(x)\|^2] \le \sigma^2$ for all $x \in \mathbb{R}^d$, where $j$ is sampled uniformly at random from $\{1, \ldots, N\}$. Wu et al. (2020) also derive convergence guarantees under $\zeta$-bounded dissimilarity assumption, i.e., when $f(x) = \frac{1}{|\mathcal{G}|}\sum_{i\in\mathcal{G}} f_i(x)$, $f_i(x) = \frac{1}{N}\sum_{j=1}^N f_i(x, j)$ for all $i \in \mathcal{G}$, and $\frac{1}{|\mathcal{G}|}\sum_{i\in\mathcal{G}} \|\nabla f_i(x) - \nabla f(x)\|^2 \le \zeta^2$.

[5] This result is obtained the main result of (Wu et al., 2020) and states that the method from (Wu et al., 2020) finds $\hat{x}$ such that $f(\hat{x}) - f(x^*) \le \varepsilon$ only for $\varepsilon \ge {}^{\sigma^2}/_{\mu^2(\frac{1}{2}-\delta)^2}$, which can be large.

[6] The result is derived under uniformly bounded variance assumption, i.e., $\mathbb{E}_{\xi\sim\mathcal{D}}[\|\nabla f(x, \xi) - \nabla f(x)\|^2] \le \sigma^2$ for all $x \in \mathbb{R}^d$.

[7] Peng et al. (2021) consider the case, when peers are allowed to communicate with their neighbors that are defined via some communication graph. The result establishes the total number of iterations/communication rounds needed to find $\hat{x}$ such that $\mathbb{E}\|\hat{x} - x^*\|^2 \le \varepsilon$ for $\varepsilon \ge \frac{\lambda^2 d}{\mu^2}\sum_{i\in\mathcal{G}} |\mathcal{B}_i|^2$, where $\lambda \ge 0$ is any non-negative number and $\mathcal{B}_i$ is the set of Byzantine peers neighboring with the $i$-th peer. In the complexity result, we use the notation $\overline{N}^2 = \sum_{i\in\mathcal{G}}(|\mathcal{G}_i|^2 + |\mathcal{B}_i|^2)$, where $\mathcal{G}_i$ is the set of good neighbors of the $i$-th peer. When $\lambda = 0$, the workers do not communicate at all. Moreover, Peng et al. (2021) analyze the case of heterogeneous local functions, composite optimization problems and time-varying setup but in that case $\lambda$ is lower bounded by a strictly positive quantity depending on the heterogeneity level and minimal non-zero singular value of the node-edge incidence matrix, i.e., any predefined accuracy cannot be achieved.

[8] The results are derived for BTARD-SGD (in the strongly convex case, for RESTARTED-BTARD-SGD) under Assumptions 3.1 and 3.2 in the case when the exact number of attacking Byzantine workers at iteration $k$ is known to each participant. See Theorems E.2, E.4, and E.6.

[9] The results are derived for BTARD-SGD (in the strongly convex case, for RESTARTED-BTARD-SGD) under Assumptions 3.1 and 3.2. See Theorems E.3, E.5, and E.7.

[10] The results are derived for BTARD-CLIPPED-SGD (in the strongly convex case, for RESTARTED-BTARD-CLIPPED-SGD) under Assumption E.1 without any additional assumptions on the tails of the distribution. Moreover, it is assumed that the exact number of attacking Byzantine workers at iteration $k$ is known to each participant. See Theorems E.8 and E.10. In the complexity results, we use the notation $\Lambda_1 = 1 + \frac{n\sqrt{\delta}}{m}$.

[11] The results are derived for BTARD-CLIPPED-SGD (in the strongly convex case, for RESTARTED-BTARD-CLIPPED-SGD) under Assumption E.1 without any additional assumptions on the tails of the distribution. See Theorems E.9 and E.11. In the complexity results, we use the notation $\Lambda_2 = 1 + \frac{n^2\delta}{m}$.

Finally, El-Mhamdi et al. (2020) propose an algorithm based on the usage of multiple servers. The authors assume that both workers and servers can be Byzantines, which is a realistic scenario. However, their approach requires the workers to send their gradients to all servers at each iteration and receive parameters from all servers as well. This leads to a significant communication overhead in practice. Moreover, El-Mhamdi et al. (2020) do not provide non-asymptotic convergence rates, making it problematic to provide an in-depth comparison with existing works and with our results as well. Therefore, it is unclear whether the usage of multiple servers speeds up training or it just leads to overhead in the communications and computations.

In contrast, our results do benefit from the communications between workers. First of all, as one can see from Table 2, the terms depending on the fraction $\delta$ of Byzantine peers in our complexity bounds for BTARD-SGD and RESTARTED-BTARD-SGD (the third terms) have better dependence on the target accuracy $\varepsilon$ than the corresponding terms in the complexity bounds from *all* previous works (even from those relying on the existence of a PS). Moreover, for sufficiently small $\varepsilon$ these terms in our complexity results are smaller than the second terms, which correspond to the main term in the complexity of parallel SGD. That is, BTARD-SGD/RESTARTED-BTARD-SGD applied to the problem with Byzantine peers has convergence guarantees that are not worse than the corresponding guarantees for parallel SGD applied to the problem without any Byzantine workers. In such regimes, our theoretical convergence results outperform even ones derived for PS-based algorithms.

We notice that Assumptions 3.1 and 3.2 used in the analysis of BTARD-SGD/RESTARTED-BTARD-SGD are slightly stronger than uniformly bounded variance assumption used in (Wu et al., 2020; Karimireddy et al., 2020; Peng et al., 2021). However, as we explain in Appendix E.3.1, our analysis allows to relax Assumptions 3.1 to uniformly bounded variance assumption, and Assumption 3.2 is reasonable for many practically important problems. Finally, we also propose and analyze BTARD-CLIPPED-SGD and RESTARTED-BTARD-CLIPPED-SGD under Assumption E.1 that may hold even in the case of *unbounded* variance of the stochastic gradient. To the best of our knowledge, this is the first time in the literature on the Byzantine-tolerant optimization when the complexity results are obtained without assuming boundedness of the stochastic gradient's variance.

## A.2 SECURITY IN DISTRIBUTED SYSTEMS: ADDITIONAL DETAILS

In this section, we provide extra details on the related work discussed in Section 2.3.

### A.2.1 BROADCAST CHANNELS

Many distributed systems rely exclusively on direct peer-to-peer connections, avoiding any centralized servers to increase reliability and avoid the performance bottleneck. In presence of malicious participants, this introduces additional security challenges since an attacker can send corrupted data to one participant and behave honestly with others. If a peer accuses another peer in sending corrupted data, it is impossible for remaining peers to determine whether the accusation is fair since only these two peers had access to the contents of the communication channel between them.

To overcome this, distributed systems can build secure broadcast channels over the peer-to-peer connections using the protocols for Byzantine broadcast. These protocols guarantee that if a peer $p$ sends a message, (a) all honest peers receive the same message and (b) the received message coincides with the original one if $p$ is honest.

Pease et al. (1980) suggest such a protocol for the case when the share of malicious peers $\delta < 1/3$, and Dolev & Strong (1983) suggest a protocol tolerating any $\delta < 1$ assuming the presence of a public key infrastructure and usage of the digital signatures (Rivest et al., 1978; Goldwasser et al., 1988).

Hirt & Raykov (2014) review how the communication complexity of various Byzantine broadcast protocols depends on the maximal tolerated $\delta$ and the length of the broadcasted message. Abraham et al. (2019) review protocols with additional practical assumptions improving the communication complexity.

### A.2.2 MULTI-PARTY RANDOM NUMBER GENERATORS

Many distributed systems may benefit from the multi-party random number generators (MPRNG) where a group of malicious peers would have little influence (bias) on the generator output. MPRNGs

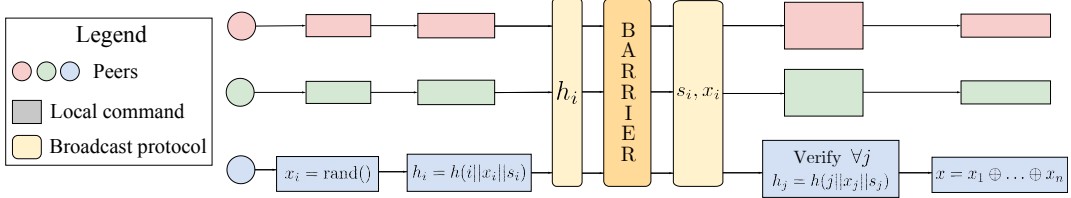

Figure 4: An intuitive scheme of MPRNG based on the generalization of Blum (1983). Here, $||$ denotes concatenation, $\oplus$ denotes bitwise XOR, $h(x)$ is a common cryptographic hash function. The hashed values include the peer identifier $i$ to protect from replay attacks and a large random string $s_i$ to resist dictionary attacks.

are usually based on multi-party coin tossing protocols, such as the protocol from Blum (1983). As an example, MPRNG allows to choose a participant winning a lottery or choose a peer whose calculations are going to be validated by other peers to detect possible cheating.

While Blum (1983) formally introduces a protocol for one bit and two parties, its generalization to multiple bits and parties (as necessary for MPRNG) is trivial assuming the presence of the broadcast channel. This modification is widely known in literature, e.g., described in Zhang et al. (2019). According to this generalization, peers should execute the following protocol to obtain $k$ random bits (see the intuitive scheme in Figure 4):

1. Each peer generates its own random string $x_i$ made of $k$ bits.

2. Each peer broadcasts *commitment* $h_i = h(i||x_i||s_i)$, where $||$ denotes concatenation, $h(x)$ is a common cryptographic hash function, $i$ is the peer's unique identifier (known by other peers), and $s_i$ is a large random string.

3. Peers wait until all of them finish broadcasting the commitments. After that, no peer can alter its $x_i$ to influence the protocol output (otherwise, peers would notice that the new value $x_i'$ does not match the commitment).

4. Each peer *reveals* their random string by broadcasting its $x_i$ and $s_i$.

5. Each peer verifies that all other peers revealed values $x_j$ and $s_j$ that match their commitments $h_j = h(j||x_j||s_j)$.

6. If a peer detects that peer $j$ aborted the procedure or its commitment does not match its revealed values, it concludes that we cannot trust peer $j$. Since other peers read the same broadcast channel, all of them can make the same conclusion. In this case, the system repeats the protocol.

7. If peers do not detect any mismatches, they calculate the protocol output $x = x_1 \oplus ... \oplus x_n$, where $\oplus$ denotes the bitwise XOR operation.

In this protocol, the commitments include the peer identifier $i$ to protect from *replay attacks* (when an attacker repeats someone else's message) and the large random string $s_i$ to resist *dictionary attacks* (when an attacker reverses the hash function using a large dictionary of its values).

While there are MPRNGs (Rabin & Ben-Or, 1989) with a negligible bias for the case when more than a half parties are honest (assuming the presence of the broadcast channel), Cleve (1986) proves that it is impossible to reach the negligible bias for the case of dishonest majority, which may be reached in practice with the Sybil attacks.

However, we note that the bias in Blum (1983) (and its modification above) appears only in the case when an attacker learns the result earlier than other peers and forces the protocol to be repeated. If we are using MPRNG to choose a peer that to be checked for cheating, we may ban all peers that aborted the procedure and restart from scratch without them, therefore eliminating the bias problem.

## B   Network and compute overhead of BTARD-SGD

**Communication overhead.**   Despite having complex structure, BTARD-SGD has only limited communication overhead, when compared to regular All-Reduce SGD. A single step of BTARD requires each peer to send each gradient tensor exactly once for aggregation, then download the

results, exactly as in Butterfly All-Reduce. On top of that, peers are only required to broadcast $O(n)$ scalars that are independent of the total size of the trained model. This includes the communication cost for MPRNG: as shown in Figure 4, one round of the MPRNG based on the generalization of Blum (1983) requires each peer to only broadcast 3 scalars.

Hirt & Raykov (2014) discuss that a simple modification of the protocol from Dolev & Strong (1983) leads to the communication complexity of $O(n^3 + ln^2)$ bits for each peer in the worst case, where $l$ is the length of the broadcasted message. For the $O(n)$ scalars, this gives $O(n^3)$ communication.

As a result, the communication complexity of a single BTARD-SGD step for each peer is $O(d + n^3)$ bits. We note that for models that benefit from distributed training, the $n^3$ component is usually dominated by the vector size $d$, as they usually contain at least tens of millions of trainable parameters.

Still, if necessary, the $n^3$ component may be improved using the Byzantine broadcast protocols with additional practical assumptions, such as allowing a small probability of error. Abraham et al. (2019) review such protocols, their assumptions, and communication complexity.

We do not restrict the way of getting the data (whether the samples are downloaded on each step, the whole dataset is distributed before the training, or the data is generated on the fly) and do not consider it a part of our algorithm, leaving the related communication cost out of scope of this analysis.

**Synchronization points.** Another important aspect of BTARD performance is synchronization. The naive implementation of Algorithm 1 would have many global synchronization "barriers" per step: one for aggregating gradients, another for choosing a random direction $z$, yet another for electing validators, etc. These frequent synchronizations could undermine the practical training performance of BTARD in high-latency networks, such as when training over the Internet.

Fortunately, it is possible to reduce the number of synchronizations by bundling them together. For instance, peers use a single MPRNG round for sampling $z$ and for electing validators. Furthermore, this MPRNG round and subsequent checks can be done in background, while a peer accumulates gradients for the next step. The only restriction is that this "shared" MPRNG round must be performed after all peers declare their checksums for that round.

With these optimizations, BTARD-SGD requires only two points of synchronization per round. The first one occurs right before gradient aggregation, and the second one is in a background task that performs verifications. Finally, there is a non-regular need for synchronization when one peer accuses another of being Byzantine. However, as we elaborated earlier, each accusation will result in at least one Byzantine being banned. Therefore, this additional cost will occur only a limited number of times over the training run.

**Computation overhead.** In terms of computation, BTARD-SGD introduces two main overheads: from validators and CENTEREDCLIP respectively. As we have shown empirically, both BTARD-SGD and BTARD-CLIPPED-SGD can withstand attacks even with 1 random validator chosen from 16 peers. As such, the computation overhead for these validators is under $10\%$ of the total compute.

As for the CENTEREDCLIP, our algorithm executes the same amount of computation as the original CENTEREDCLIP (Karimireddy et al., 2020), except that now the extra load is distributed evenly across all peers. We provide an empirical evaluation of such overhead in Appendix I.2.

Finally, we note that generating a shared vector $z$ from a scalar seed $r^t$ (as defined in Algorithm 1) has a negligible cost and can be done with any standard pseudo-random number generator. For instance, generating $z$ for ALBERT-large (the setup from Section 4.2) takes $30 \pm 1.2$ ms on the same T4 GPU that we use in our experiments.

## C   OVERVIEW OF ATTACK VECTORS

In Section 3.2, we have outlined the 4 main types of Byzantine attacks that can affect BTARD-SGD. Here, we analyze each of these types in detail and provide a list of attacks that fit these types.

**Gradient attacks.** This attack vector encompasses all attacks where Byzantine peers replace their true gradients with something else, but otherwise act normally. With this attack, $b$ Byzantine peers can collectively shift the outputs of CENTEREDCLIP by up to $\tau \cdot b/n$ in any chosen direction. However,

since Byzantine peers will need to commit hash of their incorrect gradients, *every honest validator* can accuse one of these peers with probability $b/n$ .

**Aggregation attacks.** A similar, but opposite attack type can be attempted when a Byzantine peer performs gradient aggregation. Instead of honestly computing CENTEREDCLIP, an attacker may modify the returned vector to incorporate the same kinds of changes as in gradient attacks (see above). This time, the maximum difference that can be applied through such attacks is larger, but it only affects $b/n$ of vector coordinates that are aggregated by Byzantines.

Done naively, such attacks can be detected and banned by the gradient checksum (see L15-17 in Algorithm 1). In order to ensure that the above check passes, Byzantines can misreport their $s_i^j$ in such a way that $\sum_i s_i^j = 0$. However, since actual $s_i^j$ depend only on $g_i^k$ and $\hat{g}^k$, these values can be verified by the chosen validators, and, in case of mismatch, reported via ACCUSE. We rigorously prove this in Appendix D.3.

Furthermore, if an honest validator finds that a certain peer has broadcast incorrect $s_i^j$, the validator can simultaneously accuse the corresponding Byzantine aggregator $j$ that *should have* notified about the incorrect $s_i^j$ (see L12-14 in Algorithm 1).

**Reputation abuse.** Since BTARD-SGD provides means by which benign participants can ban Byzantine attackers, it is important to ensure that the same means cannot be exploited by Byzantine peers to eliminate benign ones or otherwise abuse the system. There are three potential attack vectors that fit this description:

- Falsely accusing a benign peer,
- Persistently calling the ACCUSE procedure to slow down training,
- Automatically approving gradients without actual validation,

In BTARD-SGD, we protect against slander (issues 1. and 2.) by the design of ACCUSE protocol, by which a peer that initiates false allegations will itself be banned. As such, Byzantines can only invoke ACCUSE protocol a limited number of times before they are all permanently banned.

In turn, the attack vector (3.) is more effective: if one Byzantine was chosen as validator for another Byzantine, they can automatically report successful validation without negative consequences for either of them. However, since all validators are chosen through MPRNG, an attacker has no way of predicting whether its validator will be benign or Byzantine. Thus, any malicious activity will always have a chance of being caught by an honest validator.

**Protocol violations.** Finally, a Byzantine attacker can deviate from the protocol prescribed by BTARD-SGD in simpler ways ways, for instance:

1. Not committing the hash of its gradient when required by 4,
2. Not sending data to a particular peer when required (or sending data twice),
3. Deliberately broadcasting a hash that mismatches the subsequently sent data,
4. Sending metadata (e.g. gradient norm) that is inconsistent with previously sent gradient part,
5. Sending $s_i$ that is inconsistent with previously sent gradient,
6. Not validating when chosen as validator, validating when **not** chosen, or validating a different peer than was chosen by BTARD-SGD.

For protocol deviations that are visible to all benign participants, such as in (1.) or (6.), benign peers can ban the offender instantaneously. However, this is not the case for attacks such as (2.), where the deviation is only visible to one or few peers.

As described earlier in Section 3.2, we address this issue with a special procedure that allows any peer to ban any other peer at the cost of also being banned. Thus, if an attacker sends inconsistent gradients, norms or inner products to only one benign peer, that peer can still get the attacker banned even though it wouldn't be able to call ACCUSE.

Protecting from attacks 3, 4 and 5 from the above list also relies on this mutual elimination procedure. Specifically, if an attacker sends provably incorrect data to a benign peer, that peer will immediately

trigger the mutual elimination procedure. The only exception to this rule is if one Byzantine peer sends incorrect data to another Byzantine peer: this behavior is neither punishable nor, in itself, harmful. In turn, the mutuality of this elimination procedure prevents potential misuse by Byzantines: if an attacker decides to ban someone through this procedure, that attacker will also be banned.

## D  Detailed algorithm description

In this section, we provide more formal versions of the BTARD (Alg. 4) and BTARD-SGD (Alg. 6) algorithms, as well as auxiliary subroutines and further details. For completeness, we describe our approach in a bottom-up manner. First, in Appendix D.1, we describe Algorithm 4 and one of its main building block called CENTEREDCLIP (Karimireddy et al., 2020). Then, we formulate the ACCUSE and ELIMINATE subroutines for blocking malicious peers in Algorithm 5 and comment on them in Appendix D.2. Finally, we formulate the full BTARD-SGD in Algorithm 6 using the above subroutines as building blocks and rigorously analyze its robustness to the violations of its steps by Byzantine peers in Appendix D.3.

### D.1  BTARD and CenteredClip

We begin with a glossary of basic functions used in the algorithms:

- CHECKCOMPUTATIONS($j$) or VALIDATEPEER — run COMPUTEGRADIENTS($x^t, \xi_j^t$) and compare against the $c_j, h_j^*, s_j^*$ broadcasted by that peer. If there is mismatch, ACCUSE.
- VOTEFORBAN(peer$_j$) — send a message declaring an intent to ban peer $j$ over the broadcast channel (the message is signed with the sender's private key).
- NUMVOTES(peer$_j$) — count the number of messages declaring the intent to ban peer $j$ and received via the broadcast channel.
- BAN(peer$_j$) — add peer $j$ to a local blocklist, ignore any subsequent messages from that peer, and continue training without it.
- SPLIT($v, n$) — split vector $v$ of size $d$ into $n$ parts. The first $d \mod n$ parts are of size $\lceil d/n \rceil$ and the remaining parts have size $\lfloor d/n \rfloor$.
- MERGE($v_1, \dots, v_n$) — concatenate vectors $v_1, \dots, v_n$ into one.

Algorithm 4 defines a single gradient aggregation step (outlined earlier in Alg. 2) with additional verifications needed to reduce the negative influence of Byzantine peers. For simplicity, we assume that workers run each line in a synchronous manner (e.g. wait for all peers to broadcast hash($g_i$) before communicating the actual gradients). In practice, this restriction can be lifted in favor of asynchronous steps with several explicit synchronization barriers, but that would further complicate the pseudo-code.

One of the key building blocks of BTARD is CENTEREDCLIP – a robust aggregation rule proposed in (Karimireddy et al., 2020). Unlike a number of other aggregation rules as coordinate-wise median, Krum, geometric median, CENTEREDCLIP is provably robust against Byzantine attacks (see Theorem III from (Karimireddy et al., 2020) and Lemma E.1).

Let $\mathcal{G}$ be the set of good peers, $\mathcal{B}$ be the set of Byzantine workers, and, for simplicity, let $[n] = \mathcal{G} \sqcup \mathcal{B}$, $|\mathcal{B}| = \delta n \le \delta_0 n < {}^n/_2$. Assume that we have $n$ random vectors $x_1, \dots, x_n$, such that $\forall i, j \in \mathcal{G}$

$$\mathbb{E}[x_i] = \mathbb{E}[x_j] = x, \quad \mathbb{E}[\|x_i - x_j\|^2] \le \sigma^2,$$

and for all $i \in \mathcal{B}$ vectors $x_i$ can be arbitrary. CENTEREDCLIP works as follows: it is an iterative procedure generating a sequence $\{v_l\}_{l \ge 0}$ satisfying

$$v^{l+1} = v^l + \frac{1}{n} \sum_{i=1}^n (x_i - v^l) \min\left\{1, \frac{\tau_l}{\|x_i - v^l\|}\right\}, \qquad \text{(CenteredClip)}$$

where

$$\tau_l = 4\sqrt{\frac{(1-\delta)\left(B_l^2/3 + \sigma^2\right)}{\sqrt{3}\delta}}, \quad B_{l+1}^2 = 6.45\delta B_l^2 + 5\sigma^2. \qquad (6)$$

---

**Algorithm 4** **B**yzantine-**T**olerant **A**ll-**Red**uce (BTARD)

---

**Input:** number of workers $n$, gradient vectors on the workers $g_1, g_2, \ldots, g_n \in \mathbb{R}^d$, $d > n$, $\Delta_{\max} > 0$
    – parameter for verification 3

1: **for** workers $i = 1, \ldots, n$ in parallel **do**
2:    Split $g_i$ into $n$ parts: $g_i = (g_i(1)^\top, \ldots, g_i(n)^\top)^\top$, $g_i(j) \in \mathbb{R}^{d_j}$ for all $i, j \in [n]$
3:
4:    **broadcast** $c_i = \text{hash}(g_i)$
5:    **for** j = 1, ..., n **do**
6:        **broadcast** $c_i(j) = \text{hash}(g_i(j))$
7:
8:    **Aggregate gradients (same as Alg. 2):**
9:    Send $g_i(j)$ to peer $j$ for all $j \neq i$ and receive $g_j(i)$ from peer $j$ for all $j \neq i$
10:   **for** $j = 1, \ldots, n$ **do**
11:      **if** $\text{hash}(g_j(i)) \neq c_j(i)$ **then**
12:         ELIMINATE$(i, j)$    // Signed with peer$_i$ private key
13:   $\widehat{g}(i) = \text{CENTEREDCLIP}(g_1(i), g_2(i), \ldots, g_n(i))$
14:   **broadcast** $\widehat{c}(i) = \text{hash}(\widehat{g}(i))$
15:   Send $\widehat{g}(i)$ to each worker and receive $\widehat{g}(j)$ for all $j \neq i$ from other workers
16:   **for** $j = 1, \ldots, n$ **do**
17:      **if** $\text{hash}(\widehat{g}(j)) \neq \widehat{c}(j)$ **then**
18:         ELIMINATE$(i, j)$    // Signed with peer$_i$ private key
19:   $\widehat{g} = \text{MERGE}(\widehat{g}(1), \ldots, \widehat{g}(n))$
20:
21:   **Send metadata for verification:**
22:   Generate $r$ via MPRNG
23:   $z = \text{GETRANDOMVECTOR}(r)$
24:   **for** $j \in 1, ..., n$ **do**
25:      $\Delta_i^j = (g_i(j) - \widehat{g}(j)) \cdot \min\left\{1, \frac{\tau}{\|g_i(j) - \widehat{g}(j)\|_2}\right\}$
26:      **broadcast** $s_i^j = \langle z[j], \Delta_i^j \rangle$
27:      **broadcast** $\text{norm}_{ij} = \|g_i(j) - \widehat{g}(j)\|_2$
28:      **for** $l = 1, \ldots, n$ **do**
29:         $w_{lj} = \min\left\{1, \frac{\tau}{\text{norm}_{lj}}\right\}$
30:
31:   **for** $j = 1, \ldots, n$ **do**
32:      **Verification 1:**
33:      **if** $\text{norm}_{ji} \neq \|g_j(i) - \widehat{g}(i)\|_2$ **then**
34:         ACCUSE(i, j, $\text{norm}_{ji}$ does not mach $c_j(i)$)
35:      **Verification 2:**
36:      // peer $i$ knows $\Delta_j^i$ from CenteredClip
37:      **if** $s_j^i \neq \langle z^k[j], \Delta_j^i \rangle$ **then**
38:         ACCUSE(i, j, $s_i^j$ does not match $c_j(i)$)
39:      **if** $\sum_i^n s_i^j \neq 0$ **then**
40:         // peer$_j$ already verified that all $s^j$ are correct
41:         ACCUSE(i, j, $\widehat{g}(j)$ is wrong)
42:      **Verification 3:**
43:      **broadcast** $\text{check}_{ij} = [\|g_i(j) - \widehat{g}(j)\|_2 > \Delta_{\max}]$
44:      **if** $\sum_l \text{check}_{lj} > \frac{n}{2}$ **then**
45:         CHECKAVERAGING$(j)$
46:   **return** $\widehat{g}$

---

Intuitively, CENTEREDCLIP behaves like the mean for all points within the sphere of radius $\tau$ and like the median for "outliers". In turn, choosing different values of $\tau$ allows one to smoothly interpolate between the mean ($\tau \to \inf$) and the geometric median ($\tau \to 0$) aggregation rules.

The goal of this procedure is natural: find good enough approximation $\widehat{x}$ of $\overline{x} = \frac{1}{|\mathcal{G}|}\sum_{i \in \mathcal{G}} x_i$. In (Karimireddy et al., 2020), it is shown[5] that for $\delta \leq 0.1$ the sequence $\{v_l\}_{l \geq 0}$ generated by CENTEREDCLIP satisfies

$$\mathbb{E}[\|v^l - \overline{x}\|^2] \leq (9.7\delta)^l 3\mathbb{E}[\|v_0 - \overline{x}\|^2] + 4000\delta\sigma^2. \tag{7}$$

Moreover, Karimireddy et al. (2020) prove that for all possible aggregation rules producing $\widehat{x}$ and given $\delta_0, \sigma$ there exists such set of vectors $x_1, \dots, x_n$ and such a partition $[n] = \mathcal{G} \sqcup \mathcal{B}$ that

$$\mathbb{E}[\|\widehat{x} - \overline{x}\|^2] = \Omega(\delta\sigma^2).$$

Therefore, CENTEREDCLIP can be seen as an optimal aggregation rule neglecting numerical constants. The usage of CENTEREDCLIP helps the good peer $i$ to produce a good enough approximation of the ideal average of the $i$-th parts of stochastic gradients among good peers in BTARD.

Moreover, since $\delta \leq 0.1$ we have that $6.45\delta \leq 0.645$ implying that $B_l^2 \to B^2 \sim \sigma^2$ when $l \to \infty$, and $\tau_l \to \tau \sim \sqrt{\sigma^2/\delta}$. These limits can be easily computed from (6). Next, for $l \to \infty$ CenteredClip converges to the solution of the following equation:

$$\sum_{i=1}^{n} (x_i - v) \min\left\{1, \frac{\tau}{\|x_i - v\|}\right\} = 0. \tag{8}$$

In other words, CenteredClip for large enough $l$ approximates the fixed point iteration process of solving (8). This property plays a key role in **Verification 2** of BTARD.

---

**Algorithm 5** ACCUSE (i, j, allegation), detailed version

---

**Input:** accuser $i$, target $j$, peer count $n$, all values exchanged in Algorithm 4
1: Recalculate $g_j^k = \text{COMPUTEGRADIENTS}(x^k, \xi_j^k)$
2: Split $g_i$ into $n$ parts: $g_i = (g_i(1)^\top, \dots, g_i(n)^\top)^\top$, $g_i(j) \in \mathbb{R}^{d_j}$ for all $i, j \in [n]$
3:
4: **for** $l = 1 \dots n$ **do**
5:      **if** $\text{hash}(g_j^k) \neq c_j^k$ **or** $\text{hash}(g_j^k(l)) \neq h_j^l$ **then**
6:          VOTEFORBAN($\text{peer}_j$)    // For gradient attack
7:
8:      $\Delta_l^j = (g_l(j) - \widehat{g}(j)) \cdot \min\left\{1, \frac{\tau}{\|g_l(j) - \widehat{g}(j)\|_2}\right\}$
9:      **if** $\|g_j(l) - \widehat{g}(l)\|_2 \neq \text{norm}_{jl}$ **or** $\langle \Delta_l^j, z_j \rangle \neq s_l^j$ **or** $\sum_{l=1}^{n} s_l^j \neq 0$ **then**
10:          VOTEFORBAN($\text{peer}_j$)    // For aggregation attack
11:          **for** $o = 1, \dots, n$ **do**
12:              **if** peer $o$ approved $\text{norm}_{jo}$ or $s_j^o$ **then**
13:                  VOTEFORBAN($\text{peer}_o$)    // For covering up the $j$-th peer's aggregation attack
14:
15: **for** $l = 1 \dots n$ **do**
16:      **if** NUMVOTES($\text{peer}_l$) $\geq n/2$ **then**
17:          BAN($\text{peer}_l$)

---

### D.2 PROTOCOLS FOR BANNING BYZANTINE PEERS

ACCUSE and ELIMINATE are the two protocols by which peers ban Byzantine attackers from training. The ACCUSE protocol is only invoked if there the malicious activity of the target peer can be proven to others (we detail the exact mechanism in Algorithm 3). In contrast, ELIMINATE is a mechanism that allows any peer $i$ to ban any other peer $j$ from training without proof — but at the cost of peer $i$ also being banned. We have described this protocol earlier as a countermeasure for protocol violations (see Appendix C).

---

[5]In fact, Karimireddy et al. (2020) derive this result for two-staged version of CENTEREDCLIP. One can derive similar result for the original CENTEREDCLIP under the assumption that for all $i, j \in \mathcal{G}$ we have $\mathbb{E}[\|x_i - x_j\|^4] \leq \sigma^4$.

Both ACCUSE$(i, j)$ and ELIMINATE$(i, j)$ imply that peer $i$ uses the broadcast channel to declare its intent to ban peer $j$. All peers collect such messages during a training step and process them at the end of the step in some specific order (e.g. sorted by $(\text{type}, \text{public\_key}_i, \text{public\_key}_j)$, where type $\in \{\text{ACCUSE}, \text{ELIMINATE}\}$ and ACCUSE $<$ ELIMINATE). If processing one of the messages results in banning peer $p$, further messages involving $p$ are ignored regardless of the $p$'s role. This way, it is impossible for a Byzantine to eliminate more than one honest peer along with itself. Peers reach consensus since their decisions on banning someone are based solely on the messages from the broadcast channel (sorted in the common order) and the calculations with identical results.

### D.3  BTARD-SGD AND DETECTING PROTOCOL VIOLATIONS

Finally, the Algorithm 6 incorporates the two above procedures into a secure decentralized SGD training loop. This algorithm is intended as a more formal version of Alg. 1 from Section 3.1.

---

**Algorithm 6** BTARD-SGD

**Input:** $x^0$ – starting point, $\gamma$ – stepsize, $K$ – number of iterations, $\{s_{i,k}\}_{i,k=0,0}^{n,K-1}$ – seeds for batches computations
1: $C_0 = \text{Banned}_{-1} = \varnothing$
2: **for** $k = 0, 1, \ldots, K - 1$ **do**
3:      Worker $i$ computes $g_i^k = \begin{cases} \nabla f(x^k, \xi_{i,k}), & \text{if } i \in \mathcal{G}_k \setminus \mathcal{C}_k, \\ *, & \text{if } i \in \mathcal{B}_k \setminus \mathcal{C}_k, \end{cases}$, where $\xi_{i,k}$ is generated via seed $s_{i,k}$ available to every worker
4:      $\left(\widehat{g}^k, \text{public\_info}_k\right) = \text{BTARD}(g_{i_1^k}^k, g_{i_1^k}^k, \ldots, g_{i_{a_k}^k}^k)$, where $\{i_1^k, \ldots, i_{a_k}^k\} = (\mathcal{G}_k \cup \mathcal{B}_k) \setminus \mathcal{C}_k$
5:      // BTARD is described in Algorithm 4
6:      Choose $2m$ workers $c_1^{k+1}, \ldots, c_m^{k+1}, u_1^{k+1}, \ldots, u_m^{k+1}$ uniformly at random without replacement, $\mathcal{C}_{k+1} = \{c_1^{k+1}, \ldots, c_m^{k+1}\}, \mathcal{U}_{k+1} = \{u_1^{k+1}, \ldots, u_m^{k+1}\}$
7:      $\text{Banned}_k = \text{CHECKCOMPUTATIONS}(\mathcal{C}_{k+1}, \mathcal{U}_{k+1}, \text{public\_info}_k)$
8:      $x^{k+1} = \text{proj}_Q(x^k - \gamma \widehat{g}^k) := \text{argmin}_{x \in Q} \|x - (x^k - \gamma \widehat{g}^k)\|$
9:      $\mathcal{G}_{k+1} = \mathcal{G}_k \setminus \text{Banned}_{k-1}$
10:      $\mathcal{B}_{k+1} = \mathcal{B}_k \setminus \text{Banned}_{k-1}$

---

**Verifications 1 and 2.** While good peers always run CENTEREDCLIP, Byzantine peers can arbitrary violate the protocol meaning that they can send an arbitrary vector instead of sending the result of CENTEREDCLIP. **Verification 1** and **2** are needed to prevent such violations and make it possible to identify them during the check of computations.

First of all, both verifications are split into 2 rounds in order to let the aggregators of the corresponding part accuse those peers who send inconsistent norms or inner products. Next, in theory, we assume that all good peers find exactly the solution of CENTEREDCLIP equiation (8). Therefore, it is possible to compute the weights from (8) for each worker $i$ and each component $j$ knowing only a norm of the difference of corresponding vectors, i.e., one can compute $\min\{1, \frac{\tau}{\|g_i(j) - \widehat{g}(i)\|}\}$ by $\|g_i(j) - \widehat{g}(i)\|$. That is, if Byzantine peer $i$ sends $\text{norm}_{ij} \neq \|g_i(j) - \widehat{g}(j)\|$, it will be either revealed by $j$-th worker if $j \in \mathcal{G}$ or it will be revealed with some probability during the subsequent checks of computations.

However, **Verification 1** is insufficient to prevent malicious behavior: at iteration $k$ Byzantine peer can send $g_i^k(j)$ such that $\|g_i^k(j) - \widehat{g}^k(j)\| = \|\nabla_{(j)} f(x^k, \xi_{i,k}) - \widehat{g}^k(j)\|$. If $j \in \mathcal{B}$, then it can be the case that $i$-th worker commits the hash of $\nabla_{(j)} f(x^k, \xi_{i,k})$ and the check of gradient computation will not identify the violation of the protocol. That is why, **Verification 2** is required.

GETRANDOMVECTOR is a function that generates a random unit vector $z$ in the space of model parameters. This vector is based on a random seed $r$ obtained from MPRNG.

The goal of **Verification 2**, is to check that CENTEREDCLIP equation (8) holds for the received vector. The idea is simple: if

$$\sum_{l=1}^{n} (g_l(i) - \widehat{g}(i)) \min\left\{1, \frac{\tau}{\|g_l(i) - \widehat{g}(i)\|}\right\} = 0, \tag{9}$$

then for any $z_i$ of an appropriate dimension

$$\sum_{l=1}^{n} \langle g_l(i) - \widehat{g}(i), z_i \rangle \min \left\{ 1, \frac{\tau}{\|g_l(i) - \widehat{g}(i)\|} \right\} = 0. \tag{10}$$

Since $z_i$ in BTARD is generated from the uniform distribution on the unit Euclidean sphere, we have

$$\mathbb{P}\{(9) \text{ does not hold} \quad \& \quad (10) \text{ holds}\} = 0. \tag{11}$$

However, it is impossible to verify (10) explicitly for workers $j \neq i$. Therefore, in the algorithm, good workers check

$$\sum_{l=1}^{n} s_l^i = 0, \quad \text{where } s_l^i = \begin{cases} \langle g_l(i) - \widehat{g}(i), z_i \rangle \min \left\{ 1, \frac{\tau}{\|g_l(i) - \widehat{g}(i)\|} \right\}, & \text{if } l \in \mathcal{G}, \\ *, & \text{if } l \in \mathcal{B}. \end{cases} \tag{12}$$

Unfortunately, Byzantine peers can send arbitrary $s_l^i$. This can lead to the situations when (12) holds while (10) and, as a consequence, (9) do not. Below, we rigorously show that all possible violations of the protocol that are not detected by verifications of BTARD can be detected by the auxiliary check of computations with some probability.

**Verification 3.** This is an additional verification that serves to limit the potential scope of *aggregation attacks* (as described in Appendix C). If the result of CenteredClip landed far from too many benign participants, BTARD will verify it by re-running the same aggregation across all peers. While this procedure is costly, our analysis proves that it is has a very small probability of triggering unless some of the peers perform aggregation attacks. In the latter case, verifying the gradient accumulation will root out such attacks and ban the corresponding peers.

**Check of computations.** As we mentioned earlier, it is possible to violate the protocol without being detected by the verifications of BTARD. Therefore, extra checks of computations are required. In particular, after each aggregation in BTARD-SGD $2m$ workers are selected uniformly at random: $m$ workers check the computations at the previous step of other $m$ workers. That is, each Byzantine peer is checked at iteration $k$ with probability $\sim m/n$ by some good worker (see the proof of Thm. E.2). Consider an arbitrary Byzantine peer $j$ and all possible violations of the protocol at iteration $k$ that are not detected by verifications of BTARD.

First of all, we notice that if $c_j(i) \neq \text{hash}(\nabla_{(i)} f(x^k, \xi_{j,k}))$, then it will be detected during the check of computations with some probability[6]. Moreover, if $i \in \mathcal{G}$, then $j$-th worker has to send $c_j(i) = \text{hash}(g_j(i))$ to avoid ban.

Therefore, the only non-trivial case is when $i \in \mathcal{B}$ as well. In this case, $j$-th worker can commit $c_j(i) = \text{hash}(\nabla_{(i)} f(x^k, \xi_{j,k}))$ since it is meaningless for $i$-th worker to accuse $j$-th one. Since $\text{norm}_{ij}$, $s_i^j$ and $\widehat{g}(i)$ are known for all $i$ and $j$, $j$-th worker has to broadcast $\text{norm}_{ji} = \|\nabla_{(i)} f(x^k, \xi_{j,k}) - \widehat{g}(i)\|$ and $s_j^i = \langle \nabla_{(i)} f(x^k, \xi_{j,k}) - \widehat{g}(i), z_i \rangle \min \left\{ 1, \frac{\tau}{\|\nabla_{(i)} f(x^k, \xi_{j,k}) - \widehat{g}(i)\|} \right\}$ to avoid the ban during the check of the computations. Therefore, regardless to the choice $g_j(i)$, to pass **Verification 2** $i$-th worker should send such $\widehat{g}(i)$ that

$$\sum_{l \in \mathcal{G} \cup \{j\}} \langle \nabla_{(i)} f(x^k, \xi_{l,k}) - \widehat{g}(i), z_i \rangle \min \left\{ 1, \frac{\tau}{\|\nabla_{(i)} f(x^k, \xi_{l,k}) - \widehat{g}(i)\|} \right\} + \sum_{l \in \mathcal{B} \setminus \{j\}} s_l^i = 0.$$

In this case, the behavior of the $j$-th worker along $i$-th component is equivalent to the behavior of the good one. It means, that to avoid ban during the check of computations, each Byzantine worker $l$ should broadcast $\text{norm}_{li} = \|\nabla_{(i)} f(x^k, \xi_{l,k}) - \widehat{g}(i)\|$ and $s_l^i = \langle \nabla_{(i)} f(x^k, \xi_{l,k}) - \widehat{g}(i), z_i \rangle \min \left\{ 1, \frac{\tau}{\|\nabla_{(i)} f(x^k, \xi_{l,k}) - \widehat{g}(i)\|} \right\}$ implying that $i$-th worker should send such $\widehat{g}(i)$ that

$$\sum_{l=1}^{n} \langle \nabla_{(i)} f(x^k, \xi_{l,k}) - \widehat{g}(i), z_i \rangle \min \left\{ 1, \frac{\tau}{\|\nabla_{(i)} f(x^k, \xi_{l,k}) - \widehat{g}(i)\|} \right\} = 0.$$

In view of (11), it implies that

$$\widehat{g}(i) = \text{CenteredClip}(\nabla_{(i)} f(x^k, \xi_{1,k}), \nabla_{(i)} f(x^k, \xi_{2,k}), \dots, \nabla_{(i)} f(x^k, \xi_{2,k})),$$

i.e., there are no violations of the protocol along the $i$-th component.

---

[6]Here and below, this means that the attack/violation will be detected iff a non-Byzantine peer is chosen to validate the perpetrator.

# E CONVERGENCE ANALYSIS: MISSING PROOFS AND EXTRA DETAILS

## E.1 PRELIMINARIES

For convenience, we provide the classical definitions and facts on smooth and strongly convex functions below.

**Definition E.1** (*L*-smoothness). *We say that function* $f : Q \to \mathbb{R}$, $Q \subseteq \mathbb{R}^d$ *is L-smooth if it is differentiable and*

$$\forall x, y \in Q \quad \|\nabla f(x) - \nabla f(y)\| \leq L\|x - y\|. \tag{13}$$

One can show (Nesterov, 2003) that *L*-smoothness implies

$$\forall x, y \in Q \quad f(y) \leq f(x) + \langle \nabla f(x), y - x \rangle + \frac{L}{2}\|y - x\|^2, \tag{14}$$

$$\forall x \in Q \quad \|\nabla f(x)\|^2 \leq 2L \left( f(x) - f_* \right), \tag{15}$$

where $f_*$ is a uniform lower bound for $f$.

**Definition E.2** ($\mu$-strong convexity). *Differentiable function* $f : Q \to \mathbb{R}$, $Q \subseteq \mathbb{R}^d$ *is called $\mu$-strongly convex if*

$$\forall x, y \in Q \quad f(y) \geq f(x) + \langle \nabla f(x), y - x \rangle + \frac{\mu}{2}\|y - x\|^2. \tag{16}$$

## E.2 IMPOSSIBILITY OF BYZANTINE-TOLERANT LEARNING IN HETEROGENEOUS CASE

Several papers on Byzantine-tolerant optimization consider non-homogeneous setup, when good workers have different local functions (Wu et al., 2020; He et al., 2020). Formally, it means that instead of solving

$$\min_{x \in Q \subseteq \mathbb{R}^d} \left\{ f(x) := \mathbb{E}_{\xi \sim \mathcal{D}} \left[ f(x, \xi) \right] \right\}, \tag{17}$$

where good peers sample stochastic gradients from the full dataset (i.e., they can sample $\xi$ from $\mathcal{D}$), the following problem is considered:

$$\min_{x \in Q \subseteq \mathbb{R}^d} \left\{ f(x) := \frac{1}{|\mathcal{G}|} \sum_{i \in \mathcal{G}} f_i(x) \right\}, \tag{18}$$

where $f_i(x) = \mathbb{E}_{\xi_i \sim \mathcal{D}_i} \left[ f(x, \xi_i) \right]$ and there exists $\zeta \geq 0$ such that for all $x \in Q$

$$\frac{1}{|\mathcal{G}|} \sum_{i \in \mathcal{G}} \|\nabla f_i(x) - \nabla f(x)\|^2 \leq \zeta^2. \tag{19}$$

However, under $\zeta$-bounded heterogeneity assumption (19) it is impossible in general to solve (18) with any predefined accuracy in the presence of Byzantine peers (He et al., 2020). Moreover, this is true even when trusted Parameter-Server is available.

**Theorem E.1** (Theorem III from (He et al., 2020)). *For any optimization method* Alg *there exist $n$ functions $f_1(x), \ldots, f_n(x)$ such that at least $(1 - \delta)n$ of them are good (corresponding workers belong to $\mathcal{G}$), 1-smooth, $\mu$-strongly convex and satisfy (19) such that the output $\widehat{x}$ of* Alg *given the access to these $n$ functions has an error at least*

$$\mathbb{E}\left[ f(\widehat{x}) - \min_{x \in \mathbb{R}^d} f(x) \right] \geq \Omega\left( \frac{\delta \zeta^2}{\mu} \right) \quad \text{and} \quad \mathbb{E}\left[ \|\nabla f(\widehat{x})\|^2 \right] \geq \Omega\left( \delta \zeta^2 \right), \tag{20}$$

*where the expectation is taken w.r.t. the randomness of* Alg.

The intuition behind this negative result is as following: since the only assumption on the similarity of "good" functions is (19), Byzantine peers can shift the gradients by a vector with a norm $\sim \zeta$ without being detected. In this case, it is impossible to distinguish good peers from Byzantines but the solution of (18) depends on which workers are good and which are bad. Therefore, the best one can hope for is the convergence to some neighborhood of the solution.

The lower bounds from (20) are proportional to $\delta\zeta^2$ and cannot be made arbitrary small for given $\delta$ and $\zeta^2$. It means that the convergence to any predefined accuracy of the solution is impossible to achieve when local loss functions are $\zeta$-heterogeneous. In this sense, Byzantine-tolerant learning is impossible in the heterogeneous case. Moreover, in some practical applications (e.g., in Federated Learning (McMahan et al., 2017)), $\zeta$ from (19) can be large implying that one cannot achieve reasonable accuracy of the solution when $\delta$ is not too small (e.g., $\delta \geq 0.01$). Finally, strong convexity parameter $\mu$ is typically much smaller than 1 (assuming that the smoothness parameter is 1). In these cases, $\delta\zeta^2/\mu$ can be too large and, as a result, all methods are not converging at all.

### E.3 Convergence guarantees for BTARD-SGD

#### E.3.1 On Assumptions 3.1 and 3.2

First of all, Assumption 3.1 holds whenever standard uniformly bounded variance (UBV) assumption is satisfied. Indeed, if $\mathbb{E}_{\xi\sim\mathcal{D}}[\|\nabla f(x,\xi)-\nabla f(x)\|^2] \leq \widehat{\sigma}^2$, then $\mathbb{E}_{\xi\sim\mathcal{D}}[(\nabla_i f(x,\xi)-\nabla_i f(x))^2] \leq \widehat{\sigma}^2$ for all $i = 1,\ldots,d$, since $\|\nabla f(x,\xi) - \nabla f(x)\|^2 = \sum_{i=1}^d (\nabla_i f(x,\xi) - \nabla_i f(x))^2$. This implies that Assumption 3.1 holds with $\sigma^2 \leq d\widehat{\sigma}^2$. However, $\sigma^2$ can be significantly smaller than $d\widehat{\sigma}^2$. For example, if the noise in stochastic gradients is isotropic, e.g., Gaussian, then

$$\mathbb{E}_{\xi\sim\mathcal{D}}[(\nabla_1 f(x,\xi) - \nabla_1 f(x))^2] = \ldots = \mathbb{E}_{\xi\sim\mathcal{D}}[(\nabla_d f(x,\xi) - \nabla_d f(x))^2],$$

implying that

$$\mathbb{E}_{\xi\sim\mathcal{D}}[(\nabla_i f(x,\xi) - \nabla_i f(x))^2] = \frac{1}{d}\mathbb{E}_{\xi\sim\mathcal{D}}[(\nabla f(x,\xi) - \nabla f(x))^2] \leq \frac{\widehat{\sigma}^2}{d}$$

for all $i = 1,\ldots,d$. Therefore, in this case, Asssumption 3.1 holds with $\sigma^2 = \widehat{\sigma}^2$.

Next, it is possible to relax Assumption 3.1 to the classical UBV assumption. Indeed, in our proofs, we use Assumption 3.1 to bound the variance in the blocks of the stochastic gradients, where the blocks of components are chosen for workers to execute BTARD. If these blocks are chosen uniformly at random, i.e., the vector is split into several parts of the given sizes uniformly at random, then it is enough to have

$$\mathbb{E}\left[\|\nabla f_{[S]}(x,\xi) - \nabla_{[S]} f(x)\|^2\right] \leq \frac{s\sigma^2}{d} \tag{21}$$

for a random subset $S$ of $\{1,\ldots,d\}$ such that $|S| = s$, where expectation is taken w.r.t. $\xi$ and $S$. To derive inequality (21) from UBV assumption $\mathbb{E}_{\xi\sim\mathcal{D}}[\|\nabla f(x,\xi) - \nabla f(x)\|^2] \leq \widehat{\sigma}^2$ we use tower property of the expectation:

$$
\begin{aligned}
\mathbb{E}\left[\|\nabla f_{[S]}(x,\xi) - \nabla_{[S]} f(x)\|^2\right] &= \mathbb{E}_{\xi\sim\mathcal{D}}\left[\mathbb{E}_S\left[\|\nabla f_{[S]}(x,\xi) - \nabla_{[S]} f(x)\|^2\right]\right] \\
&= \mathbb{E}_{\xi\sim\mathcal{D}}\left[\sum_{i=1}^d \mathbb{P}\{i \in S\}(\nabla_i f(x,\xi) - \nabla_i f(x))^2\right] \\
&= \frac{s}{d}\mathbb{E}_{\xi\sim\mathcal{D}}\left[\sum_{i=1}^d (\nabla_i f(x,\xi) - \nabla_i f(x))^2\right] \\
&= \frac{s}{d}\mathbb{E}_{\xi\sim\mathcal{D}}\left[\|\nabla f(x,\xi) - \nabla f(x)\|^2\right] \leq \frac{s\widehat{\sigma}^2}{d},
\end{aligned}
$$

i.e., (21) holds for $\sigma^2 = \widehat{\sigma}^2$.

Finally, as we show in Lemmas E.2 and E.4, under As. 3.2 **Verification 3** at BTARD leads to extra checking of computations with probability $\sim 1/n$ at each iteration when all workers honestly follow the protocol and under a proper choice of $\Delta_{\max}$. Therefore, extra computations either appear due to malicious manipulations of Byzantine peers, and lead eventually to the ban for the Byzantine peers who deviate from the protocol, or, when all workers honestly follow the protocol, only once per $n$ iterations on average. There are a number of important machine learning tasks, such as training ResNet-50 on Imagenet (Zhang et al., 2020) and many others image classification problems, where the noise in the stochastic gradient has much "lighter" (sub-Gaussian) tails. That is, As. 3.2 is reasonable for a large class of practically important problems. Moreover, in Appendix E.4, we also provide an analysis of BTARD-CLIPPED-SGD and RESTARTED-BTARD-CLIPPED-SGD without any assumptions on the tails of the stochastic gradients distribution.

### E.3.2 Quality of the aggregation

The quality of the aggregation at each iteration of BTARD-SGD significantly affects the rate of the method. That is, properties of $\widetilde{g}^k$ are highly important for the convergence of BTARD-SGD. This aggregator is obtained via BTARD that requires to know a tight estimate of the total number of Byzantine workers violating the protocol at iteration $k$ – clipping parameter $\tau$ depends on this quantity. Therefore, it is natural to start with relatively simple setup when the number of Byzantine workers violating the protocol is known at each iteration.

Before we formulate the first result we introduce some useful notations. Let $n_k$ be the total number of peers at iteration $k$, $b_k$ be the total number of Byzantine peers at iteration $k$, $\widehat{b}^k$ be the total number of Byzantine peers violating the protocol at iteration $k$, and $\delta_k = \frac{b_k}{n_k}, \widehat{\delta}_k = \frac{\widehat{b}_k}{n_k - m}$. In view of new notation, we start with the ideal situation when $\widehat{b}_k$ is known for each worker at each iteration $k$. First of all, it is needed to to estimate the quality of the aggregation for good workers.

**Lemma E.1** (Theorem IV from Karimireddy et al. (2020)). *Let As. 3.1 hold, $\delta \leq 0.1(n - m)$, and $i \in \mathcal{G}_k \setminus \mathcal{C}_k$. Assume that $\widehat{b}_k$ is known for each worker at iteration $k$ and $\delta = \widehat{\delta}_k$ is used to compute clipping parameter $\tau_l$ for CenteredClip. If the total number of iterations $T$ of CenteredClip satisfies $T \geq \log_{9.7\delta} \frac{\delta\sigma^2}{3\mathbb{E}[\|v^0 - \overline{g}^k\|^2]}$, then*

$$\mathbb{E}\left[\|\widehat{g}^k(i) - \overline{g}^k(i)\|^2 \mid x^k\right] \leq 4001\widehat{\delta}_k \frac{\sigma^2}{n_k - m}, \tag{22}$$

*where $\overline{g}^k(i) = \frac{1}{|\mathcal{G}_k \setminus \mathcal{C}_k|} \sum\limits_{j \in \mathcal{G}_k \setminus \mathcal{C}_k} g_j^k(i)$.*

*Proof.* The proof follows directly from (7). □

Unlike the good peers, Byzantine workers can cooperate and shift the result of CenteredClip in the components they aggregate without being revealed at **Verification 2** of BTARD. However, they cannot produce an arbitrary large shifts due to **Verification 3**. The next lemma estimates the maximal possible magnitude of a shift together with probability of triggering CheckAveraging at iteration $k$ for at least one worker.

**Lemma E.2.** *Let As. 3.1 and 3.2 hold, $b \leq 0.1(n - m)$, and $i \in \mathcal{B}_k \setminus \mathcal{C}_k$. Assume that $\widehat{b}_k$ is known for each worker at iteration $k$, $\Delta_{\max}^k = \frac{(1+\sqrt{3})\sqrt{2}\sigma}{\sqrt{n_k - m}}$ and $\delta = \widehat{\delta}_k$ is used to compute clipping parameter $\tau_l$ for CenteredClip. If the total number of iterations $T$ of CenteredClip satisfies $T \geq \log_{9.7\delta} \frac{\delta\sigma^2}{3\mathbb{E}[\|v^0 - \overline{g}^k\|^2]}$ and CheckAveraging(i) is not triggered, then*

$$\mathbb{E}\left[\|\widehat{g}^k(i) - \overline{g}^k(i)\|^2 \mid x^k\right] \leq \frac{4\left((1 + \sqrt{3})^2 + 4\right)\sigma^2}{n_k - m}, \tag{23}$$

*where $\overline{g}^k(i) = \frac{1}{|\mathcal{G}_k \setminus \mathcal{C}_k|} \sum\limits_{j \in \mathcal{G}_k \setminus \mathcal{C}_k} g_j^k(i)$. Moreover, if $\widehat{b}_k = 0$ and $n_k - m \geq 170$, then $\widehat{g}^k(i) = \overline{g}^k(i)$ and*

$$\mathbb{P}\left\{\text{CheckAveraging is triggered for } \geq 1 \text{ peer} \mid x^k\right\} \leq \frac{149}{49(n_k - m)}. \tag{24}$$

*Proof.* If CheckAveraging(i) is not triggered at iteration $k$, then for $r_k \geq \frac{n_k - m}{2}$ good workers $i_1, i_2, \ldots, i_{r_k} \in \mathcal{G}_k \setminus \mathcal{C}_k$ we have $\|g_{i_j}^k(i) - \widehat{g}^k(i)\| \leq \Delta_{\max}^k$. Therefore, due to the independence of

$g_i^k$, $i \in \mathcal{G}_k \setminus \mathcal{C}_k$ for fixed $x^k$ we have

$$
\begin{aligned}
\mathbb{E}\left[\left\|\widehat{g}^k(i) - \overline{g}^k(i)\right\|^2 \mid x^k\right] &\leq 2\mathbb{E}\left[\left\|\widehat{g}^k(i) - \frac{1}{r_k}\sum_{j=1}^{r_k} g_{i_j}^k(i)\right\|^2 \mid x^k\right] \\
&\quad + 2\mathbb{E}\left[\left\|\frac{1}{r_k}\sum_{j=1}^{r_k} g_{i_j}^k(i) - \overline{g}^k(i)\right\|^2 \mid x^k\right] \\
&\leq 2\mathbb{E}\left[\frac{1}{r_k}\sum_{j=1}^{r_k}\|\widehat{g}^k(i) - g_{i_j}^k(i)\|^2\right] + 4\mathbb{E}\left[\|\nabla_{(i)}f(x^k) - \overline{g}^k(i)\|^2 \mid x^k\right] \\
&\quad 4\mathbb{E}\left[\left\|\frac{1}{r_k}\sum_{j=1}^{r_k} g_{i_j}^k(i) - \nabla_{(i)}f(x^k)\right\|^2 \mid x^k\right] \\
&\leq 2(\Delta_{\max}^k)^2 + \frac{4\sigma^2}{|\mathcal{G}_k \setminus \mathcal{C}_k|} + \frac{8\sigma^2}{n_k - m} \\
&\leq \frac{4\left((1 + \sqrt{3})^2 + 4\right)\sigma^2}{n_k - m},
\end{aligned}
$$

where we use $|\mathcal{G}_k \setminus \mathcal{C}_k| \geq r_k \geq \frac{n_k - m}{2}$ and $\nabla f_{(i)}(x^k) = \mathbb{E}[g_{i_j}^k \mid x^k]$. Finally, let us estimate the probability of triggering CHECKAVERAGING when all workers follow the protocol. In this case, $\widehat{g}(i) = \overline{g}^k(i)$. Next, due to As. 3.2 and $b \leq 0.1(n - m)$ we have

$$
\mathbb{P}\left\{\|\overline{g}^k(i) - \nabla f_{(i)}(x^k)\|_2 > \sqrt{\frac{\sigma^2}{n_k - m}} \mid x^k\right\} \leq \frac{1}{|\mathcal{G}_k \setminus \mathcal{C}_k|^2} \leq \frac{100}{49(n_k - m)^2}
$$

and for all $j \in \mathcal{G}_k \setminus \mathcal{C}_k$

$$
\mathbb{P}\left\{\|g_j^k(i) - \nabla f_{(i)}(x^k)\|_2 > \sqrt{\frac{3\sigma^2}{n_k - m}} \mid x^k\right\} \leq \frac{1}{9}.
$$

Consider the independent random variables $\eta_j$, $j \in \mathcal{G}_k \setminus \mathcal{C}_k$, where

$$
\eta_j = \begin{cases} 1, & \text{if } \|g_j^k(i) - \nabla f_{(i)}(x^k)\|_2 \leq \sqrt{\frac{3\sigma^2}{n_k - m}}, \\ 0, & \text{otherwise,} \end{cases}
$$

where $x^k$ is fixed. Then, $\eta_j$ is a Bernoulli random variable with parameter of "success" $q \geq 8/9$. Applying Hoeffding's inequality we get that

$$
\begin{aligned}
\mathbb{P}\left\{\sum_{j \in \mathcal{G}_k \setminus \mathcal{C}_k} \eta_j \leq \frac{n_k - m}{2} \mid x^k\right\} &\leq \exp\left(-2(n_k - m)\left(q - \frac{n_k - m}{2|\mathcal{G}_k \setminus \mathcal{C}_k|}\right)^2\right) \\
&\leq \exp\left(-2(n_k - m)\left(\frac{8}{9} - \frac{n - m}{1.4(n - m)}\right)^2\right) \\
&= \exp\left(-\frac{242(n_k - m)}{3969}\right).
\end{aligned}
$$

Since for all $j \in \mathcal{G}_k \setminus \mathcal{C}_k$ we have $\|\overline{g}^k(i) - g_j^k(i)\|_2 \leq \|\overline{g}^k(i) - \nabla_{(i)}f(x^k)\|_2 + \|\nabla_{(i)}f(x^k) - g_j^k(i)\|_2$ the obtained bounds imply that CHECKAVERAGING is triggered for at least one worker at iteration $k$ with probability not greater than

$$
\frac{100}{49(n_k - m)} + (n_k - m)\exp\left(-\frac{242(n_k - m)}{3969}\right) \leq \frac{149}{49(n_k - m)},
$$

where we use that $\exp\left(-\frac{242x}{3969}\right) \leq \frac{1}{x^2}$ for all $x \geq 170$. $\qquad\square$

We notice that Byzantine peers can trigger CHECKAVERAGING by violating the protocol. However, each Byzantine is checked at iteration $k$ with probability $p \sim m/n$ (see Thm. E.2). Therefore, Byzantine workers can trigger only $\mathcal{O}\left(bn/m\right)$ extra rounds of communications and computations on average via triggering CHECKAVERAGING. In contrast, when there are no Byzantine workers or all workers follow the protocol CHECKAVERAGING is triggered only once per $\mathcal{O}(n-m)$ iterations that is a negligible communication an computation overhead when $n$ is large.

Combining two previous lemmas we get the following result.

**Lemma E.3.** *Let As. 3.1 hold and $b \le 0.1(n-m)$. Assume that $\widehat{b}_k$ is known for each worker at iteration $k$, $\Delta_{\max}^k = \frac{(1+\sqrt{3})\sqrt{2}\sigma}{\sqrt{n_k-m}}$ and $\delta = \widehat{\delta}_k$ is used to compute clipping parameter $\tau_l$ for CenteredClip. If the total number of iterations $T$ of CenteredClip satisfies $T \ge \log_{9.7\delta} \frac{\delta\sigma^2}{3\mathbb{E}[\|v^0-\overline{g}^k\|^2]}$ and CHECKAVERAGING is not triggered for any worker, then*

$$\mathbb{E}\left[\|\widehat{g}^k - \overline{g}^k\|^2 \mid x^k\right] \le C\widehat{\delta}_k\sigma^2, \tag{25}$$

$$\mathbb{E}\left[\|\widehat{g}^k\|^2 \mid x^k\right] \le 2C\widehat{\delta}_k\sigma^2 + 2\|\nabla f(x^k)\|^2 + \frac{2\sigma^2}{n-2b-m}, \tag{26}$$

*where $\overline{g}^k = \frac{1}{|\mathcal{G}_k\setminus\mathcal{C}_k|}\sum\limits_{j\in\mathcal{G}_k\setminus\mathcal{C}_k} g_j^k$ and $C = 4001 + 4\left((1+\sqrt{3})^2 + 4\right)$.*

*Proof.* We have

$$
\begin{aligned}
\mathbb{E}\left[\|\widehat{g}^k - \overline{g}^k\|^2 \mid x^k\right] &= \sum_{i\in\mathcal{G}_k\setminus\mathcal{C}_k}\mathbb{E}\left[\|\widehat{g}^k(i)-\overline{g}^k(i)\|^2 \mid x^k\right] + \sum_{i\in\mathcal{B}_k\setminus\mathcal{C}_k}\mathbb{E}\left[\|\widehat{g}^k(i)-\overline{g}^k(i)\|^2 \mid x^k\right] \\
&\overset{(22),(23)}{\le} (1-\widehat{\delta}_k)(n_k-m)\cdot 4001\widehat{\delta}_k\frac{\sigma^2}{n_k-m} + \widehat{\delta}_k(n_k-m)\cdot\frac{4\left((1+\sqrt{3})^2+4\right)\sigma^2}{n_k-m} \\
&= C\widehat{\delta}_k\sigma^2.
\end{aligned}
$$

Next, using the independence of $g_j^k$ for $j \in \mathcal{G}_k \setminus \mathcal{C}_k$ and fixed $x^k$ we derive

$$
\begin{aligned}
\mathbb{E}\left[\|\widehat{g}^k\|^2 \mid x^k\right] &\le 2\mathbb{E}\left[\|\widehat{g}^k - \overline{g}^k\|^2 \mid x^k\right] + 2\mathbb{E}\left[\|\overline{g}^k\|^2 \mid x^k\right] \\
&\overset{(25)}{\le} 2C\widehat{\delta}_k\sigma^2 + 2\|\nabla f(x^k)\|^2 + 2\mathbb{E}\left[\|\overline{g}^k - \nabla f(x^k)\|^2 \mid x^k\right] \\
&\le 2C\widehat{\delta}_k\sigma^2 + 2\|\nabla f(x^k)\|^2 + \frac{2\sigma^2}{|\mathcal{G}_k\setminus\mathcal{C}_k|} \\
&\le 2C\widehat{\delta}_k\sigma^2 + 2\|\nabla f(x^k)\|^2 + \frac{2\sigma^2}{n-2b-m}.
\end{aligned}
$$

$\square$

In view of the definition of $(\delta,c)$-robust aggregator from Karimireddy et al. (2020), the result of BTARD at iteration $k$ is $(\widehat{\delta}_k, C)$-robust. However, we derive this property under assumption that $\widehat{b}_k$ is known to all workers at each iteration $k$, which is impractical.

When $\widehat{b}_k$ is unknown the situation changes dramatically: in general, good peers can only know some upper bound for the fraction of Byzantine peers at iteration $k$. Unfortunately, if used without bans, this is not enough to converge to any accuracy of the solution since BTARD-SGD is a permutation-invariant algorithm in terms of Karimireddy et al. (2020). Therefore, in this case, we always use CENTEREDCLIP with $\tau_l = \infty$ for all $l \ge 0$, i.e., good peers compute an exact average. In this settings, even 1 Byzantine worker can significantly shift the average in all parts of the vector. The next lemma quantifies the negative effect of Byzantine workers in this case.

**Lemma E.4.** *Let As. 3.1 and 3.2 hold, $b \le 0.1(n-m)$, $m \le (n-2b)/2$. Assume that $\Delta_{\max}^k = \frac{(1+\sqrt{3})\sqrt{2}\sigma}{\sqrt{n_k-m}}$ and $\delta = 0$ is used to compute clipping parameter $\tau_l$ for CenteredClip. If CHECKAVERAGING is not triggered for any worker, then*

$$\mathbb{E}\left[\|\widehat{g}^k - \overline{g}^k\|^2 \mid x^k\right] \le C\sigma^2\mathbb{1}_{k,v}, \tag{27}$$

$$\mathbb{E}\left[\|\widehat{g}^k\|^2 \mid x^k\right] \le 2C\sigma^2 \mathbb{1}_{k,v} + 2\|\nabla f(x^k)\|^2 + \frac{2\sigma^2}{n - 2b - m}, \tag{28}$$

where $\overline{g}^k = \frac{1}{|\mathcal{G}_k \setminus \mathcal{C}_k|} \sum_{j \in \mathcal{G}_k \setminus \mathcal{C}_k} g_j^k$, $C = 4\left((1 + \sqrt{3})^2 + 4\right)$, and $\mathbb{1}_{k,v}$ is an indicator function of the

event that at least 1 Byzantine peer violates the protocol at iteration $k$. Moreover, if $\widehat{b}_k = 0$ and $n_k - m \ge 170$, then $\widehat{g}^k(i) = \overline{g}^k(i)$ and

$$\mathbb{P}\left\{\text{CHECKAVERAGING is triggered for } \ge 1 \text{ peer} \mid x^k\right\} \le \frac{149}{49(n_k - m)}. \tag{29}$$

*Proof.* If CHECKAVERAGING is not triggered for any worker, then $\|\widehat{g}^k(i) - \overline{g}^k(i)\|^2 \le (\Delta_{\max}^k)^2 \mathbb{1}_{k,v}$ for all $i \in (\mathcal{G}_k \cup \mathcal{C}_k) \setminus \mathcal{C}_k$ implying

$$
\begin{aligned}
\mathbb{E}\left[\|\widehat{g}^k - \overline{g}^k\|^2 \mid x^k\right] &= \sum_{i \in (\mathcal{G}_k \cup \mathcal{C}_k) \setminus \mathcal{C}_k} \mathbb{E}\left[\|\widehat{g}^k(i) - \overline{g}^k(i)\|^2 \mid x^k\right] \\
&\le (n_k - m) \cdot \frac{2\left((1 + \sqrt{3})^2 + 8\right)\sigma^2}{n_k - m} \mathbb{1}_{k,v} \le C\sigma^2 \mathbb{1}_{k,v}.
\end{aligned}
$$

Next, using the independence of $g_j^k$ for $j \in \mathcal{G}_k \setminus \mathcal{C}_k$ and fixed $x^k$ we derive

$$
\begin{aligned}
\mathbb{E}\left[\|\widehat{g}^k\|^2 \mid x^k\right] &\le 2\mathbb{E}\left[\|\widehat{g}^k - \overline{g}^k\|^2 \mid x^k\right] + 2\mathbb{E}\left[\|\overline{g}^k\|^2 \mid x^k\right] \\
&\overset{(27)}{\le} 2C\sigma^2 \mathbb{1}_{k,v} + 2\|\nabla f(x^k)\|^2 + 2\mathbb{E}\left[\|\overline{g}^k - \nabla f(x^k)\|^2 \mid x^k\right] \\
&\le 2C\sigma^2 \mathbb{1}_{k,v} + 2\|\nabla f(x^k)\|^2 + \frac{2\sigma^2}{|\mathcal{G}_k \setminus \mathcal{C}_k|} \\
&\le 2C\sigma^2 \mathbb{1}_{k,v} + 2\|\nabla f(x^k)\|^2 + \frac{2\sigma^2}{n - 2b - m}.
\end{aligned}
$$

The proof of the final part of the lemma is identical to the proof of the same result from Lemma E.2. □

### E.3.3 NON-CONVEX CASE

In this section, we provide the complete statements and the full proofs of the convergence results for BTARD-SGD when the objective function $f$ is smooth, but can be non-convex. We start with the case when the number of attacking Byzantine workers is known at each iteration.

**Theorem E.2.** *Let As. 3.1 and As. 3.2 hold, $Q = \mathbb{R}^d$, and $f$ be $L$-smooth (see Def. E.1) and uniformly lower bounded by $f_*$. Moreover, assume that $b \le 0.1(n - m)$, $m \le (n-2b)/2$, and the exact number of attacking Byzantine peers is known to all good peers at each iteration. Next, assume that*

$$\gamma = \min\left\{\frac{1}{4L}, \sqrt{\frac{\Delta_0 n}{L\sigma^2 K}}\right\}, \quad \Delta_{\max}^k = \frac{(1 + \sqrt{3})\sqrt{2}\sigma}{\sqrt{n_k - m}}, \tag{30}$$

*where $\Delta_0 = f(x^0) - f_*$ and $\Delta_{\max}^k$ is the parameter for verification 3 at iteration $k$ of BTARD-SGD. Then, we have $\mathbb{E}[\|\nabla f(\overline{x}^K)\|^2] \le \varepsilon^2$ after $K$ iterations of BTARD-SGD, where*

$$K = \mathcal{O}\left(\frac{L\Delta_0}{\varepsilon^2} + \frac{L\Delta_0 \sigma^2}{n\varepsilon^4} + \frac{n\delta\sigma^2}{m\varepsilon^2}\right) \tag{31}$$

*and $\overline{x}^K$ is picked uniformly at random from $\{x^0, x^1, \ldots, x^{K-1}\}$.*

*Proof.* From $L$-smoothness of $f$ we have

$$
\begin{aligned}
f(x^{k+1}) &\overset{(14)}{\le} f(x^k) + \langle \nabla f(x^k), x^{k+1} - x^k \rangle + \frac{L}{2}\|x^{k+1} - x^k\|^2 \\
&= f(x^k) - \gamma \langle \nabla f(x^k), \widehat{g}^k \rangle + \frac{L\gamma^2}{2}\|\widehat{g}^k\|^2.
\end{aligned}
$$

Taking the conditional expectation $\mathbb{E}[\cdot \mid x^k]$ from the both sides of the previous inequality we obtain

$$
\begin{aligned}
\mathbb{E}\left[f(x^{k+1}) \mid x^k\right] &\leq f(x^k) - \gamma\|\nabla f(x^k)\|^2 - \gamma\left\langle\nabla f(x^k), \mathbb{E}\left[\widehat{g}^k - \overline{g}^k \mid x^k\right]\right\rangle \\
&\quad + \frac{L\gamma^2}{2}\mathbb{E}\left[\|\widehat{g}^k\|^2 \mid x^k\right] \\
&\overset{(26)}{\leq} f(x^k) - \frac{\gamma}{2}\|\nabla f(x^k)\|^2 + \frac{\gamma}{2}\left\|\mathbb{E}\left[\widehat{g}^k - \overline{g}^k \mid x^k\right]\right\|^2 \\
&\quad + CL\gamma^2\widehat{\delta}_k\sigma^2 + L\gamma^2\|\nabla f(x^k)\| + \frac{L\gamma^2\sigma^2}{n-2b-m} \\
&\leq f(x^k) - \frac{\gamma}{2}(1-2L\gamma)\|\nabla f(x^k)\|^2 + \frac{\gamma}{2}\mathbb{E}\left[\|\widehat{g}^k - \overline{g}^k\|^2 \mid x^k\right] \\
&\quad + CL\gamma^2\widehat{\delta}_k\sigma^2 + \frac{L\gamma^2\sigma^2}{n-2b-m}.
\end{aligned}
$$

Since $\gamma \leq \frac{1}{4L}$ we continue our derivations as

$$
\begin{aligned}
\mathbb{E}\left[f(x^{k+1}) \mid x^k\right] &\leq f(x^k) - \frac{\gamma}{4}\|\nabla f(x^k)\|^2 + \gamma C\sigma^2(1+L\gamma)\widehat{\delta}_k + \frac{L\gamma^2\sigma^2}{n-2b-m} \\
&\leq f(x^k) - \frac{\gamma}{4}\|\nabla f(x^k)\|^2 + 2\gamma C\sigma^2\widehat{\delta}_k + \frac{L\gamma^2\sigma^2}{n-2b-m}.
\end{aligned}
$$

Taking the full expectation from the both sides of the obtained inequality and summing up the results for $k = 0, 1, \ldots, K-1$ we get

$$
\begin{aligned}
\frac{1}{K}\sum_{k=0}^{K-1}\mathbb{E}\left[\|\nabla f(x^k)\|^2\right] &\leq \frac{4}{\gamma K}\sum_{k=0}^{K-1}\mathbb{E}\left[f(x^k) - f(x^{k+1})\right] + \frac{8C\sigma^2}{K}\mathbb{E}\left[\sum_{k=0}^{K-1}\widehat{\delta}_k\right] + \frac{4L\gamma\sigma^2}{n-2b-m} \\
&= \frac{4\left(f(x^0) - \mathbb{E}[f(x^K)]\right)}{\gamma K} + \frac{8C\sigma^2}{K}\mathbb{E}\left[\sum_{k=0}^{K-1}\frac{\widehat{b}_k}{n_k-m}\right] + \frac{4L\gamma\sigma^2}{n-2b-m} \\
&\leq \frac{4(f(x^0) - f_*)}{\gamma K} + \frac{8C\sigma^2}{K(n-2b-m)}\mathbb{E}\left[\sum_{k=0}^{K-1}\widehat{b}_k\right] + \frac{4L\gamma\sigma^2}{n-2b-m}.
\end{aligned}
$$

If a Byzantine peer deviates from the protocol at iteration $k$, it will be detected with some probability $p_k$ during the next iteration. One can lower bound this probability as

$$
p_k \geq m \cdot \frac{|\mathcal{G}_k|}{n_k} \cdot \frac{1}{n_k} = \frac{m(1-\delta_k)}{n_k} \geq \frac{m}{n}.
$$

Therefore, each individual Byzantine worker can violate the protocol no more than $1/p$ times on average implying that

$$
\begin{aligned}
\frac{1}{K}\sum_{k=0}^{K-1}\mathbb{E}\left[\|\nabla f(x^k)\|^2\right] &\leq \frac{4(f(x^0) - f_*)}{\gamma K} + \frac{8Cnb\sigma^2}{Km(n-2b-m)} + \frac{4L\gamma\sigma^2}{n-2b-m} \\
&\leq \frac{4(f(x^0) - f_*)}{\gamma K} + \frac{16Cnb\sigma^2}{Km(n-2b)} + \frac{8L\gamma\sigma^2}{n-2b} \\
&\leq \frac{4(f(x^0) - f_*)}{\gamma K} + \frac{160Cn\delta\sigma^2}{7Km} + \frac{80L\gamma\sigma^2}{7n}.
\end{aligned}
$$

Since $\overline{x}^K$ is picked uniformly at random from $\{x^0, x^1, \ldots, x^{K-1}\}$ we have

$$
\mathbb{E}\left[\|\nabla f(\overline{x}^K)\|^2\right] \leq \frac{4(f(x^0) - f_*)}{\gamma K} + \frac{160Cn\delta\sigma^2}{7Km} + \frac{80L\gamma\sigma^2}{7n}.
$$

Using the stepsize rule

$$
\gamma = \min\left\{\frac{1}{4L}, \sqrt{\frac{\Delta_0 n}{L\sigma^2 K}}\right\}
$$

we derive

$$\mathbb{E}\left[\|\nabla f(\overline{x}^K)\|^2\right] = \mathcal{O}\left(\frac{L\Delta_0}{K} + \frac{\sqrt{L\Delta_0}\sigma}{\sqrt{nK}} + \frac{n\delta\sigma^2}{mK}\right)$$

meaning that after

$$K = \mathcal{O}\left(\frac{L\Delta_0}{\varepsilon^2} + \frac{L\Delta_0\sigma^2}{n\varepsilon^4} + \frac{n\delta\sigma^2}{m\varepsilon^2}\right)$$

iterations BTARD-SGD guarantees $\mathbb{E}\left[\|\nabla f(\overline{x}^K)\|^2\right] \leq \varepsilon^2$. $\qquad\square$

In the main part of the paper, we notice that the rate of BTARD-SGD in the presence of bad workers is asymptotically the same as for SGD without Byzantine peers when $\varepsilon$ is sufficiently small[7]. This phenomenon has a clear intuition. When the target accuracy $\varepsilon$ is small, the stepsize $\gamma$ is also needed to be small enough. However, as we show in Lemmas E.3 and E.4, Byzantine workers can produce only a bounded shift independent of the stepsize. Moreover, they can violate the protocol at only $\sim n/m$ iterations on average. Therefore, the overall impact of Byzantine workers on the convergence of BTARD-SGD decreases when the stepsize $\gamma$ decreases.

Next, we derive the result without assuming that $\widehat{b}^k$ is known to all peers at each iteration.

**Theorem E.3.** *Let As. 3.1 and 3.2 hold, $Q = \mathbb{R}^d$, and $f$ be $L$-smooth (see Def. E.1) and uniformly lower bounded by $f_*$. Moreover, assume that $b \leq 0.1(n-m)$, $m \leq (n-2b)/2$, and $\delta = 0$ is used to compute clipping parameter $\tau_l$ for CenteredClip. Next, assume that*

$$\gamma = \min\left\{\frac{1}{4L}, \sqrt{\frac{\Delta_0 n}{L\sigma^2 K}}\right\}, \quad \Delta_{\max}^k = \frac{(1+\sqrt{3})\sqrt{2}\sigma}{\sqrt{n_k - m}}, \tag{32}$$

*where $\Delta_0 = f(x^0) - f_*$ and $\Delta_{\max}^k$ is the parameter for verification 3 at iteration $k$ of BTARD-SGD. Then, we have $\mathbb{E}[\|\nabla f(\overline{x}^K)\|^2] \leq \varepsilon^2$ after $K$ iterations of BTARD-SGD, where*

$$K = \mathcal{O}\left(\frac{L\Delta_0}{\varepsilon^2} + \frac{L\Delta_0\sigma^2}{n\varepsilon^4} + \frac{nb\sigma^2}{m\varepsilon^2}\right) \tag{33}$$

*and $\overline{x}^K$ is picked uniformly at random from $\{x^0, x^1, \ldots, x^{K-1}\}$.*

*Proof.* The proof is almost identical to the proof of Theorem E.2. Following the same steps and using (27) and (28) instead of (25) and (26) respectively we obtain the same sequence of inequalities up to the following change: instead of $\widehat{\delta}_k$ we should use $\mathbb{1}_{k,v}$. Therefore, we have

$$\frac{1}{K}\sum_{k=0}^{K-1}\mathbb{E}\left[\|\nabla f(x^k)\|^2\right] \leq \frac{4(f(x^0) - f_*)}{\gamma K} + \frac{8C\sigma^2}{K}\mathbb{E}\left[\sum_{k=0}^{K-1}\mathbb{1}_{k,v}\right] + \frac{4L\gamma\sigma^2}{n - 2b - m}.$$

If a Byzantine peer deviates from the protocol at iteration $k$, it will be detected with some probability $p_k$ during the next iteration. One can lower bound this probability as

$$p_k \geq m \cdot \frac{|\mathcal{G}_k|}{n_k} \cdot \frac{1}{n_k} = \frac{m(1-\delta_k)}{n_k} \geq \frac{m}{n}.$$

That is, each individual Byzantine worker can violate the protocol no more than $1/p$ times on average. However, even one Byzantine peer can create a shift of the order $\Delta_{\max}^k$ at each part of the resulting vector. Therefore, all Byzantine peers can violate the protocol no more than $b/p$ times on average implying that

$$\begin{aligned}
\frac{1}{K}\sum_{k=0}^{K-1}\mathbb{E}\left[\|\nabla f(x^k)\|^2\right] &\leq \frac{4(f(x^0) - f_*)}{\gamma K} + \frac{8Cnb\sigma^2}{Km} + \frac{4L\gamma\sigma^2}{n - 2b - m} \\
&\leq \frac{4(f(x^0) - f_*)}{\gamma K} + \frac{8Cnb\sigma^2}{Km} + \frac{8L\gamma\sigma^2}{n - 2b} \\
&\leq \frac{4(f(x^0) - f_*)}{\gamma K} + \frac{8Cnb\sigma^2}{Km} + \frac{80L\gamma\sigma^2}{7n}.
\end{aligned}$$

---

[7]This is true for convex and strongly convex cases as well.

Since $\overline{x}^K$ is picked uniformly at random from $\{x^0, x^1, \ldots, x^{K-1}\}$ we have

$$\mathbb{E}\left[\|\nabla f(\overline{x}^K)\|^2\right] \leq \frac{4(f(x^0) - f_*)}{\gamma K} + \frac{8Cnb\sigma^2}{Km} + \frac{80L\gamma\sigma^2}{7n}.$$

Using the stepsize rule

$$\gamma = \min\left\{\frac{1}{4L}, \sqrt{\frac{\Delta_0 n}{L\sigma^2 K}}\right\}$$

we derive

$$\mathbb{E}\left[\|\nabla f(\overline{x}^K)\|^2\right] = \mathcal{O}\left(\frac{L\Delta_0}{K} + \frac{\sqrt{L\Delta_0}\sigma}{\sqrt{nK}} + \frac{nb\sigma^2}{mK}\right)$$

meaning that after

$$K = \mathcal{O}\left(\frac{L\Delta_0}{\varepsilon^2} + \frac{L\Delta_0\sigma^2}{n\varepsilon^4} + \frac{nb\sigma^2}{m\varepsilon^2}\right)$$

iterations BTARD-SGD guarantees $\mathbb{E}\left[\|\nabla f(\overline{x}^K)\|^2\right] \leq \varepsilon^2$. $\qquad\square$

As we notice in the main part of the paper, the third term of the obtained complexity result is significantly worse than in (31): it is proportional to $b$ instead of $\delta = b/n$. However, (33) is derived without assuming that $\widehat{b}_k$ is known for all workers at each iteration. Moreover, as in (31), the third term in (33) has better dependence on $\varepsilon$ than the second term implying that for small enough $\varepsilon$ the rate of BTARD-SGD in the presence of bad workers without assuming that $\widehat{b}_k$ is known at each iteration is asymptotically the same as for SGD without Byzantine peers[8].

### E.3.4 CONVEX CASE

In this section, we provide the complete statements and the full proofs of the convergence results for BTARD-SGD when the objective function $f$ is smooth and convex. We start with the case when the number of attacking Byzantine workers is known at each iteration.

**Theorem E.4.** *Let As. 3.1 and 3.2 hold, $Q = \mathbb{R}^d$, $f$ be $L$-smooth (see Def. E.1), convex, and $x^*$ be some optimum of $f$. Moreover, assume that $b \leq 0.1(n - m)$, $m \leq (n-2b)/2$, and the exact number of attacking Byzantine peers is known to all good peers at each iteration. Next, assume that*

$$\gamma = \min\left\{\frac{1}{4L}, \sqrt{\frac{7nR_0^2}{120\sigma^2 K}}, \sqrt{\frac{m^2 R_0^2}{1440C\sigma^2 n^2\delta}}\right\}, \quad \Delta_{\max}^k = \frac{(1+\sqrt{3})\sqrt{2}\sigma}{\sqrt{n_k - m}}, \qquad (34)$$

*where $R_0 \geq \|x^0 - x^*\|$ and $\Delta_{\max}^k$ is the parameter for verification 3 at iteration $k$ of BTARD-SGD. Then, we have $\mathbb{E}[f(\overline{x}^K) - f(x^*)] \leq \varepsilon$ after $K$ iterations of BTARD-SGD, where*

$$K = \mathcal{O}\left(\frac{LR_0^2}{\varepsilon} + \frac{\sigma^2 R_0^2}{n\varepsilon^2} + \frac{n\sqrt{\delta}\sigma R_0}{m\varepsilon}\right) \qquad (35)$$

*and $\overline{x}^K = \frac{1}{K}\sum_{k=0}^{K-1}$.*

*Proof.* Lemma E.3 implies

$$
\begin{aligned}
\mathbb{E}\left[\|x^{k+1} - x^*\|^2 \mid x^k\right] &= \mathbb{E}\left[\|x^k - x^* - \gamma\widehat{g}^k\|^2 \mid x^k\right] \\
&= \|x^k - x^*\|^2 - 2\gamma\mathbb{E}\left[\langle x^k - x^*, \widehat{g}^k\rangle \mid x^k\right] + \gamma^2\mathbb{E}\left[\|\widehat{g}^k\|^2 \mid x^k\right] \\
&\overset{(26)}{\leq} \|x^k - x^*\|^2 - 2\gamma\langle x^k - x^*, \nabla f(x^k)\rangle + 2\gamma^2\|\nabla f(x^k)\|^2 \\
&\quad - 2\gamma\mathbb{E}\left[\langle x^k - x^*, \widehat{g}^k - \overline{g}^k\rangle \mid x^k\right] + 2\gamma^2 C\widehat{\delta}_k\sigma^2 + \frac{2\gamma^2\sigma^2}{n - 2b - m}.
\end{aligned}
$$

---

[8]This is true for convex and strongly convex cases as well.

Next, we use convexity (see (16) with $\mu = 0$) and $L$-smoothness of $f$:

$$\mathbb{E}\left[\|x^{k+1} - x^*\|^2 \mid x^k\right] \overset{(15),(16)}{\leq} \|x^k - x^*\|^2 - 2\gamma(1 - 2L\gamma)\left(f(x^k) - f(x^*)\right)$$
$$-2\gamma\mathbb{E}\left[\langle x^k - x^*, \widehat{g}^k - \overline{g}^k\rangle \mid x^k\right] + 2\gamma^2 C\sigma^2 \frac{\widehat{b}_k}{n_k - m} + \frac{2\gamma^2\sigma^2}{n - 2b - m}.$$

To estimate the inner product in the right-hand side we apply Cauchy-Schwarz inequality:

$$-2\gamma\mathbb{E}\left[\langle x^k - x^*, \widehat{g}^k - \overline{g}^k\rangle \mid x^k\right] \leq 2\gamma\|x^k - x^*\|\mathbb{E}\left[\|\widehat{g}^k - \overline{g}^k\| \mid x^k\right]$$
$$\leq 2\gamma\|x^k - x^*\|\sqrt{\mathbb{E}\left[\|\widehat{g}^k - \overline{g}^k\|^2 \mid x^k\right]}$$
$$\overset{(25)}{\leq} 2\gamma\sqrt{C}\sigma\|x^k - x^*\|\sqrt{\widehat{\delta}_k} \leq \frac{2\gamma\sqrt{C}\sigma}{\sqrt{n_k - m}}\|x^k - x^*\|\sqrt{\widehat{b}_k}$$
$$\leq \frac{2\gamma\sqrt{C}\sigma}{\sqrt{n - 2b - m}}\|x^k - x^*\|\sqrt{\widehat{b}_k}.$$

Putting all together and using $b \leq 0.1(n - m)$, $m \leq (n-2b)/2$, $\gamma \leq 1/4L$, $n_k - m \geq n - 2b - m$, we obtain

$$\mathbb{E}\left[\|x^{k+1} - x^*\|^2 \mid x^k\right] \leq \|x^k - x^*\|^2 - \gamma\left(f(x^k) - f(x^*)\right)$$
$$+ \frac{4\gamma\sqrt{5C}\sigma}{\sqrt{n}}\|x^k - x^*\|\sqrt{\widehat{b}_k} + \frac{40\gamma^2 C\sigma^2}{7n}\widehat{b}_k + \frac{40\gamma^2\sigma^2}{7n}.$$

Taking the full expectation from the both sides of the above inequality and summing up the results for $k = 0, 1, \ldots, K - 1$ we derive

$$\frac{\gamma}{K}\sum_{k=0}^{K-1}\mathbb{E}[f(x^k) - f(x^*)] \leq \frac{1}{K}\sum_{k=0}^{K-1}\left(\mathbb{E}\left[\|x^k - x^*\|^2\right] - \mathbb{E}\left[\|x^{k+1} - x^*\|^2\right]\right) + \frac{40\gamma^2\sigma^2}{7n}$$
$$+ \frac{4\gamma\sqrt{5C}\sigma}{\sqrt{n}K}\sum_{k=0}^{K-1}\mathbb{E}\left[\|x^k - x^*\|\sqrt{\widehat{b}_k}\right] + \frac{40\gamma^2 C\sigma^2}{7nK}\sum_{k=0}^{K-1}\mathbb{E}[\widehat{b}_k]$$
$$\leq \frac{\|x^0 - x^*\|^2 - \mathbb{E}[\|x^K - x^*\|^2]}{K} + \frac{40\gamma^2\sigma^2}{7n}$$
$$+ \frac{4\gamma\sqrt{5C}\sigma}{\sqrt{n}K}\sum_{k=0}^{K-1}\sqrt{\mathbb{E}\left[\|x^k - x^*\|^2\right]\mathbb{E}[\widehat{b}_k]} + \frac{40\gamma^2 C\sigma^2}{7nK}\sum_{k=0}^{K-1}\mathbb{E}[\widehat{b}_k].$$

From Jensen's inequality we have $f(\overline{x}^K) \leq \frac{1}{K}\sum_{k=0}^{K-1}f(x^k)$, where $\overline{x}^K = \frac{1}{K}\sum_{k=0}^{K-1}x^k$. Using this and new notation $R_k = \|x^k - x^*\|$, $k > 0$, $R_0 \geq \|x^0 - x^*\|$ we get

$$0 \leq \gamma\mathbb{E}\left[f(\overline{x}^K) - f(x^*)\right] \leq \frac{R_0^2 - \mathbb{E}[R_K^2]}{K} + \frac{40\gamma^2\sigma^2}{7n}$$
$$+ \frac{4\gamma\sqrt{5C}\sigma}{\sqrt{n}K}\sum_{k=0}^{K-1}\sqrt{\mathbb{E}\left[R_k^2\right]\mathbb{E}[\widehat{b}_k]} + \frac{40\gamma^2 C\sigma^2}{7nK}\sum_{k=0}^{K-1}\mathbb{E}[\widehat{b}_k] \quad (36)$$

implying (after changing the indices) that

$$\mathbb{E}[R_k^2] \leq R_0^2 + \frac{40\gamma^2\sigma^2 k}{7n} + \frac{4\gamma\sqrt{5C}\sigma}{\sqrt{n}}\sum_{l=0}^{k-1}\sqrt{\mathbb{E}\left[R_l^2\right]\mathbb{E}[\widehat{b}_l]} + \frac{40\gamma^2 C\sigma^2}{7n}\sum_{l=0}^{k-1}\mathbb{E}[\widehat{b}_l] \quad (37)$$

holds for all $k \geq 0$. In the remaining part of the proof we derive by induction that

$$R_0^2 + \frac{40\gamma^2\sigma^2 k}{7n} + \frac{4\gamma\sqrt{5C}\sigma}{\sqrt{n}}\sum_{l=0}^{k-1}\sqrt{\mathbb{E}\left[R_l^2\right]\mathbb{E}[\widehat{b}_l]} + \frac{40\gamma^2 C\sigma^2}{7n}\sum_{l=0}^{k-1}\mathbb{E}[\widehat{b}_l] \leq 2R_0^2 \quad (38)$$

for all $k = 0, \ldots, K$. For $k = 0$ this inequality trivially holds. Next, assume that it holds for all $k = 0, 1, \ldots, T-1, T \leq K-1$. Let us show that it holds for $k = T$ as well. From (37) and (38) we have that $\mathbb{E}[R_k^2] \leq 2R_0^2$ for all $k = 0, 1, \ldots, T-1$. Therefore,

$$
\begin{aligned}
\mathbb{E}[R_T^2] &\leq R_0^2 + \frac{40\gamma^2\sigma^2 T}{7n} + \frac{4\gamma\sqrt{5C}\sigma}{\sqrt{n}} \sum_{l=0}^{T-1} \sqrt{\mathbb{E}\left[R_l^2\right]\mathbb{E}[\widehat{b}_l]} + \frac{40\gamma^2 C\sigma^2}{7n} \sum_{l=0}^{T-1} \mathbb{E}[\widehat{b}_l] \\
&\leq R_0^2 + \frac{40\gamma^2\sigma^2 T}{7n} + \frac{4\gamma\sqrt{10C}\sigma R_0}{\sqrt{n}} \sum_{l=0}^{T-1} \sqrt{\mathbb{E}[\widehat{b}_l]} + \frac{40\gamma^2 C\sigma^2}{7n} \sum_{l=0}^{T-1} \mathbb{E}[\widehat{b}_l].
\end{aligned}
$$

If a Byzantine peer deviates from the protocol at iteration $k$, it will be detected with some probability $p_k$ during the next iteration. One can lower bound this probability as

$$
p_k \geq m \cdot \frac{|\mathcal{G}_k|}{n_k} \cdot \frac{1}{n_k} = \frac{m(1-\delta_k)}{n_k} \geq \frac{m}{n}.
$$

Therefore, each individual Byzantine worker can violate the protocol no more than $1/p$ times on average implying that

$$
\begin{aligned}
\mathbb{E}[R_T^2] &\leq R_0^2 + \frac{40\gamma^2\sigma^2 T}{7n} + \frac{4n\gamma\sqrt{10Cb}\sigma R_0}{m\sqrt{n}} + \frac{40\gamma^2 C\sigma^2 nb}{7nm} \\
&= R_0^2 + \frac{40\gamma^2\sigma^2 T}{7n} + \frac{4n\gamma\sqrt{10C\delta}\sigma R_0}{m} + \frac{40\gamma^2 C\sigma^2 n\delta}{7m}.
\end{aligned}
$$

Taking

$$
\gamma = \min\left\{ \frac{1}{4L}, \sqrt{\frac{7nR_0^2}{120\sigma^2 K}}, \sqrt{\frac{m^2 R_0^2}{1440 C\sigma^2 n^2 \delta}} \right\}
$$

we ensure that

$$
\frac{40\gamma^2\sigma^2 T}{7n} + \frac{4n\gamma\sqrt{10C\delta}\sigma R_0}{m} + \frac{40\gamma^2 C\sigma^2 n\delta}{7m} \leq \frac{R_0^2}{3} + \frac{R_0^2}{3} + \frac{R_0^2}{3} = R_0^2,
$$

and, as a result, we get $\mathbb{E}[R_T^2] \leq 2R_0^2$. Therefore, (38) holds for all $k = 0, 1, \ldots, K$. Together with (36) it implies

$$
\mathbb{E}\left[f(\overline{x}^K) - f(x^*)\right] \leq \frac{2R_0^2}{\gamma K}.
$$

Next, from our stepsize rule (34) it follows that

$$
\mathbb{E}\left[f(\overline{x}^K) - f(x^*)\right] = \mathcal{O}\left( \frac{LR_0^2}{K} + \frac{\sigma R_0}{\sqrt{nK}} + \frac{n\sqrt{\delta}\sigma R_0}{mK} \right)
$$

meaning that after

$$
K = \mathcal{O}\left( \frac{LR_0^2}{\varepsilon} + \frac{\sigma^2 R_0^2}{n\varepsilon^2} + \frac{n\sqrt{\delta}\sigma R_0}{m\varepsilon} \right)
$$

iterations BTARD-SGD guarantees $\mathbb{E}[f(\overline{x}^K) - f(x^*)] \leq \varepsilon$. $\qquad \square$

In the convex case, similar observations hold as in the non-convex case. Next, we derive the result without assuming that $\widehat{b}^k$ is known to all peers at each iteration.

**Theorem E.5.** *Let As. 3.1 and 3.2 hold, $Q = \mathbb{R}^d$, $f$ be $L$-smooth (see Def. E.1), convex, and $x^*$ be some optimum of $f$. Moreover, assume that $b \leq 0.1(n-m)$, $m \leq (n-2b)/2$, and $\delta = 0$ is used to compute clipping parameter $\tau_l$ for CenteredClip. Next, assume that*

$$
\gamma = \min\left\{ \frac{1}{4L}, \sqrt{\frac{7nR_0^2}{120\sigma^2 K}}, \sqrt{\frac{m^2 R_0^2}{72 C\sigma^2 n^2 b^2}} \right\}, \quad \Delta_{\max}^k = \frac{(1+\sqrt{3})\sqrt{2}\sigma}{\sqrt{n_k - m}}, \tag{39}
$$

*where $R_0 \geq \|x^0 - x^*\|$ and $\Delta_{\max}^k$ is the parameter for verification 3 at iteration $k$ of BTARD-SGD. Then, we have $\mathbb{E}[f(\overline{x}^K) - f(x^*)] \leq \varepsilon$ after $K$ iterations of BTARD-SGD, where*

$$
K = \mathcal{O}\left( \frac{LR_0^2}{\varepsilon} + \frac{\sigma^2 R_0^2}{n\varepsilon^2} + \frac{nb\sigma R_0}{m\varepsilon} \right) \tag{40}
$$

*and $\overline{x}^K = \frac{1}{K}\sum_{k=0}^{K-1}$.*

*Proof.* The proof is almost identical to the proof of Theorem E.4. Following the same steps and using (27) and (28) instead of (25) and (26) respectively we obtain the same sequence of inequalities up to the following change: instead of $\widehat{\delta}_k$ we should use $\mathbb{1}_{k,v}$. Therefore, we have

$$
\begin{aligned}
\mathbb{E}\left[\|x^{k+1} - x^*\|^2 \mid x^k\right] \leq & \; \|x^k - x^*\|^2 - 2\gamma\left(1 - 2L\gamma\right)\left(f(x^k) - f(x^*)\right) \\
& -2\gamma\mathbb{E}\left[\langle x^k - x^*, \widehat{g}^k - \overline{g}^k\rangle \mid x^k\right] + 2\gamma^2 C\sigma^2 \mathbb{1}_{k,v} + \frac{2\gamma^2\sigma^2}{n - 2b - m},
\end{aligned}
$$

$$
-2\gamma\mathbb{E}\left[\langle x^k - x^*, \widehat{g}^k - \overline{g}^k\rangle \mid x^k\right] \leq 2\gamma\sqrt{C}\sigma\|x^k - x^*\|\mathbb{1}_{k,v},
$$

that result in

$$
\begin{aligned}
\mathbb{E}\left[\|x^{k+1} - x^*\|^2 \mid x^k\right] \leq & \; \|x^k - x^*\|^2 - \gamma\left(f(x^k) - f(x^*)\right) \\
& +2\gamma\sqrt{C}\sigma\|x^k - x^*\|\mathbb{1}_{k,v} + 2\gamma^2 C\sigma^2 \mathbb{1}_{k,v} + \frac{40\gamma^2\sigma^2}{7n}.
\end{aligned}
$$

Taking the full expectation from the both sides of the above inequality and summing up the results for $k = 0, 1, \ldots, K - 1$ we derive

$$
\begin{aligned}
\frac{\gamma}{K}\sum_{k=0}^{K-1}\mathbb{E}[f(x^k) - f(x^*)] \leq & \; \frac{1}{K}\sum_{k=0}^{K-1}\left(\mathbb{E}\left[\|x^k - x^*\|^2\right] - \mathbb{E}\left[\|x^{k+1} - x^*\|^2\right]\right) + \frac{40\gamma^2\sigma^2}{7n} \\
& +\frac{2\gamma\sqrt{C}\sigma}{K}\sum_{k=0}^{K-1}\mathbb{E}\left[\|x^k - x^*\|\mathbb{1}_{k,v}\right] + \frac{2\gamma^2 C\sigma^2}{K}\sum_{k=0}^{K-1}\mathbb{E}[\mathbb{1}_{k,v}] \\
\leq & \; \frac{\|x^0 - x^*\|^2 - \mathbb{E}[\|x^K - x^*\|^2]}{K} + \frac{40\gamma^2\sigma^2}{7n} \\
& +\frac{2\gamma\sqrt{C}\sigma}{K}\sum_{k=0}^{K-1}\sqrt{\mathbb{E}\left[\|x^k - x^*\|^2\right]\mathbb{E}[\mathbb{1}_{k,v}]} + \frac{2\gamma^2 C\sigma^2}{K}\sum_{k=0}^{K-1}\mathbb{E}[\mathbb{1}_{k,v}].
\end{aligned}
$$

From Jensen's inequality we have $f(\overline{x}^K) \leq \frac{1}{K}\sum_{k=0}^{K-1} f(x^k)$, where $\overline{x}^K = \frac{1}{K}\sum_{k=0}^{K-1} x^k$. Using this and new notation $R_k = \|x^k - x^*\|$, $k \geq 0$ we get

$$
\begin{aligned}
0 \leq \gamma\mathbb{E}\left[f(\overline{x}^K) - f(x^*)\right] \leq & \; \frac{R_0^2 - \mathbb{E}[R_K^2]}{K} + \frac{40\gamma^2\sigma^2}{7n} \\
& +\frac{2\gamma\sqrt{C}\sigma}{K}\sum_{k=0}^{K-1}\sqrt{\mathbb{E}\left[R_k^2\right]\mathbb{E}[\mathbb{1}_{k,v}]} + \frac{2\gamma^2 C\sigma^2}{K}\sum_{k=0}^{K-1}\mathbb{E}[\mathbb{1}_{k,v}] \quad (41)
\end{aligned}
$$

implying (after changing the indices) that

$$
\mathbb{E}[R_k^2] \leq R_0^2 + \frac{40\gamma^2\sigma^2 k}{7n} + 2\gamma\sqrt{C}\sigma\sum_{l=0}^{k-1}\sqrt{\mathbb{E}\left[R_l^2\right]\mathbb{E}[\mathbb{1}_{l,v}]} + 2\gamma^2 C\sigma^2\sum_{l=0}^{k-1}\mathbb{E}[\mathbb{1}_{l,v}] \quad (42)
$$

holds for all $k \geq 0$. In the remaining part of the proof we derive by induction that

$$
R_0^2 + \frac{40\gamma^2\sigma^2 k}{7n} + 2\gamma\sqrt{C}\sigma\sum_{l=0}^{k-1}\sqrt{\mathbb{E}\left[R_l^2\right]\mathbb{E}[\mathbb{1}_{l,v}]} + 2\gamma^2 C\sigma^2\sum_{l=0}^{k-1}\mathbb{E}[\mathbb{1}_{l,v}] \leq 2R_0^2 \quad (43)
$$

for all $k = 0, \ldots, K$. For $k = 0$ this inequality trivially holds. Next, assume that it holds for all $k = 0, 1, \ldots, T - 1$, $T \leq K - 1$. Let us show that it holds for $k = T$ as well. From (42) and (43) we have that $\mathbb{E}[R_k^2] \leq 2R_0^2$ for all $k = 0, 1, \ldots, T - 1$. Therefore,

$$
\begin{aligned}
\mathbb{E}[R_T^2] \leq & \; R_0^2 + \frac{40\gamma^2\sigma^2 k}{7n} + 2\gamma\sqrt{C}\sigma\sum_{l=0}^{T-1}\sqrt{\mathbb{E}\left[R_l^2\right]\mathbb{E}[\mathbb{1}_{l,v}]} + 2\gamma^2 C\sigma^2\sum_{l=0}^{T-1}\mathbb{E}[\mathbb{1}_{l,v}] \\
\leq & \; R_0^2 + \frac{40\gamma^2\sigma^2 k}{7n} + 2\gamma\sqrt{2C}\sigma R_0\sum_{l=0}^{T-1}\sqrt{\mathbb{E}[\mathbb{1}_{l,v}]} + 2\gamma^2 C\sigma^2\sum_{l=0}^{T-1}\mathbb{E}[\mathbb{1}_{l,v}].
\end{aligned}
$$

If a Byzantine peer deviates from the protocol at iteration $k$, it will be detected with some probability $p_k$ during the next iteration. One can lower bound this probability as

$$p_k \geq m \cdot \frac{|\mathcal{G}_k|}{n_k} \cdot \frac{1}{n_k} = \frac{m(1-\delta_k)}{n_k} \geq \frac{m}{n}.$$

That is, each individual Byzantine worker can violate the protocol no more than $1/p$ times on average. However, even one Byzantine peer can create a shift of the order $\Delta_{\max}^k$ at each part of the resulting vector. Therefore, all Byzantine peers can violate the protocol no more than $b/p$ times on average implying that

$$\mathbb{E}[R_T^2] \quad \leq \quad R_0^2 + \frac{40\gamma^2\sigma^2 T}{7n} + \frac{2\gamma n b\sqrt{2C}\sigma R_0}{m} + \frac{2\gamma^2 nbC\sigma^2}{m}.$$

Taking

$$\gamma = \min\left\{\frac{1}{4L}, \sqrt{\frac{7nR_0^2}{120\sigma^2 K}}, \sqrt{\frac{m^2 R_0^2}{72C\sigma^2 n^2 b^2}}\right\}$$

we ensure that

$$\frac{40\gamma^2\sigma^2 T}{7n} + \frac{2\gamma n b\sqrt{2C}\sigma R_0}{m} + \frac{2\gamma^2 nbC\sigma^2}{m} \leq \frac{R_0^2}{3} + \frac{R_0^2}{3} + \frac{R_0^2}{3} = R_0^2,$$

and, as a result, we get $\mathbb{E}[R_T^2] \leq 2R_0^2$. Therefore, (43) holds for all $k = 0, 1, \ldots, K$. Together with (41) it implies

$$\mathbb{E}\left[f(\overline{x}^K) - f(x^*)\right] \leq \frac{2R_0^2}{\gamma K}.$$

Next, from our stepsize rule (39) it follows that

$$\mathbb{E}\left[f(\overline{x}^K) - f(x^*)\right] = \mathcal{O}\left(\frac{LR_0^2}{K} + \frac{\sigma R_0}{\sqrt{nK}} + \frac{nb\sigma R_0}{mK}\right)$$

meaning that after

$$K = \mathcal{O}\left(\frac{LR_0^2}{\varepsilon} + \frac{\sigma^2 R_0^2}{n\varepsilon^2} + \frac{nb\sigma R_0}{m\varepsilon}\right)$$

iterations BTARD-SGD guarantees $\mathbb{E}[f(\overline{x}^K) - f(x^*)] \leq \varepsilon$. $\qquad\square$

### E.3.5 STRONGLY CONVEX CASE: RESTARTED-BTARD-SGD

In this section, we provide the complete statements and the full proofs of the convergence results for the restarted version of BTARD-SGD (RESTARTED-BTARD-SGD, Alg. 7) when the objective function $f$ is smooth and strongly convex.

---

**Algorithm 7** RESTARTED-BTARD-SGD

**Input:** $x^0$ – starting point, $r$ – number of restarts, $\{\gamma_t\}_{t=1}^r$ – stepsizes for BTARD-SGD, $\{K_t\}_{t=1}^r$ – number of iterations for BTARD-SGD, $\{s_{i,k,t}\}_{i,k,t=0,0,0}^{n,K-1,r}$ – seeds for batches computations
1: $\widehat{x}^0 = x^0$
2: **for** $t = 1, 2, \ldots, r$ **do**
3:      Run BTARD-SGD (Alg. 6) for $K_t$ iterations with stepsize $\gamma_t$, starting point $\widehat{x}^{t-1}$, and seeds for batches computations $\{s_{i,k,t}\}_{i,k=0,0}^{n,K-1}$. Define $\widehat{x}^t$ as $\widehat{x}^t = \frac{1}{K_t}\sum_{k=0}^{K_t} x^{k,t}$, where $x^{0,t}, x^{1,t}, \ldots, x^{K_t,t}$ are the iterates produced by BTARD-SGD.
**Output:** $\widehat{x}^r$

---

We start with the case when the number of attacking Byzantine workers is known at each iteration.

**Theorem E.6.** *Let As. 3.1 and 3.2 hold, $Q = \mathbb{R}^d$, $f$ be $L$-smooth (see Def. E.1), $\mu$-strongly convex (see Def. E.2), and $x^*$ be some optimum of $f$. Moreover, assume that $b \leq 0.1(n-m)$, $m \leq (n-2b)/2$, and the exact number of attacking Byzantine peers is known to all good peers at each iteration. Next, assume that*

$$\gamma_t = \min\left\{\frac{1}{4L}, \sqrt{\frac{7nR_0^2}{120 \cdot 2^t\sigma^2 K_t}}, \sqrt{\frac{m^2R_0^2}{1440 \cdot 2^t C\sigma^2 n^2\delta}}\right\}, \quad \Delta_{\max}^{k,t} = \frac{(1+\sqrt{3})\sqrt{2}\sigma}{\sqrt{n_k^t - m}}, \quad (44)$$

$$K_t = \left\lceil\max\left\{\frac{16L}{\mu}, \frac{32\sigma^2 2^t}{\mu^2 R_0^2}, \frac{48\sqrt{10C}n\sqrt{\delta}\sigma 2^{\frac{t}{2}}}{m\mu R_0}\right\}\right\rceil, \quad r = \left\lceil\log_2\frac{\mu R_0^2}{\varepsilon}\right\rceil - 1 \quad (45)$$

*where $R_0 \geq \|x^0 - x^*\|$, $\Delta_{\max}^{k,t}$ is the parameter for verification 3 at iteration $k$ of* BTARD-SGD *during the $t$-th restart, $n_k^t$ is the total number of workers at iteration $k$ of $t$-th restart. Then, we have $\mathbb{E}[f(\widehat{x}^r) - f(x^*)] \leq \varepsilon$ after $r$ restarts of* BTARD-SGD *and the total number of executed iterations of* BTARD-SGD *is*

$$\sum_{t=1}^{r} K_t = \mathcal{O}\left(\frac{L}{\mu}\log\frac{\mu R_0^2}{\varepsilon} + \frac{\sigma^2}{n\mu\varepsilon} + \frac{n\sqrt{\delta}\sigma}{m\sqrt{\mu\varepsilon}}\right). \quad (46)$$

*Proof.* Theorem E.4 implies that BTARD-SGD with

$$\gamma = \min\left\{\frac{1}{4L}, \sqrt{\frac{7nR_0^2}{120\sigma^2 K}}, \sqrt{\frac{m^2R_0^2}{1440C\sigma^2 n^2\delta}}\right\}$$

guarantees

$$\mathbb{E}\left[f(\overline{x}^K) - f(x^*)\right] \leq \frac{2R_0^2}{\gamma K}$$

after $K$ iterations. Therefore, after the first restart we have

$$\mathbb{E}[f(\widehat{x}^1) - f(x^*)] \leq \frac{2R_0^2}{\gamma_1 K_1} \leq \frac{\mu R_0^2}{4}.$$

From $\mu$-strong convexity of $f$ and $\nabla f(x^*) = 0$ we have

$$\frac{\mu}{2}\|\widehat{x}^1 - x^*\|^2 \leq f(\widehat{x}^1) - f(x^*) \implies \mathbb{E}[\|\widehat{x}^1 - x^*\|^2] \leq \frac{R_0^2}{2}.$$

Next, assume that we have $\mathbb{E}[f(\widehat{x}^t) - f(x^*)] \leq \frac{\mu R_0^2}{2^{t+1}}$, $\mathbb{E}[\|\widehat{x}^t - x^*\|^2] \leq \frac{R_0^2}{2^t}$ for some $t \leq r - 1$. Then, Theorem E.4 implies that

$$\mathbb{E}[f(\widehat{x}^{t+1}) - f(x^*) \mid x^t] \leq \frac{2\|\widehat{x}^t - x^*\|^2}{\gamma_t K_t}.$$

Taking the full expectation from the both sides of previous inequality we get

$$\mathbb{E}[f(\widehat{x}^{t+1}) - f(x^*)] \leq \frac{2\mathbb{E}[\|\widehat{x}^t - x^*\|^2]}{\gamma_t K_t} \leq \frac{2R_0^2}{2^t\gamma_t K_t} \leq \frac{\mu R_0^2}{2^{t+2}}.$$

From $\mu$-strong convexity of $f$ and $\nabla f(x^*) = 0$ we have

$$\frac{\mu}{2}\|\widehat{x}^{t+1} - x^*\|^2 \leq f(\widehat{x}^{t+1}) - f(x^*) \implies \mathbb{E}[\|\widehat{x}^{t+1} - x^*\|^2] \leq \frac{R_0^2}{2^{t+1}}.$$

Therefore, by mathematical induction we have that for all $t = 1, \ldots, r$

$$\mathbb{E}[f(\widehat{x}^t) - f(x^*)] \leq \frac{\mu R_0^2}{2^{t+1}}, \quad \mathbb{E}\left[\|\widehat{x}^t - x^*\|^2\right] \leq \frac{R_0^2}{2^t}.$$

Then, after $r = \left\lceil \log_2 \frac{\mu R_0^2}{\varepsilon} \right\rceil - 1$ restarts of BTARD-SGD we have $\mathbb{E}[f(\widehat{x}^r) - f(x^*)] \leq \varepsilon$. The total number of iterations executed by BTARD-SGD is

$$
\begin{aligned}
\sum_{t=1}^{r} K_t &= \mathcal{O}\left( \sum_{t=1}^{r} \max\left\{ \frac{L}{\mu}, \frac{\sigma^2 2^t}{\mu^2 R_0^2}, \frac{n\sqrt{\delta}\sigma 2^{\frac{t}{2}}}{m\mu R_0} \right\} \right) \\
&= \mathcal{O}\left( \frac{L}{\mu}r + \frac{\sigma^2 2^r}{\mu^2 R_0^2} + \frac{n\sqrt{\delta}\sigma 2^{\frac{r}{2}}}{m\mu R_0} \right) \\
&= \mathcal{O}\left( \frac{L}{\mu}\log\frac{\mu R_0^2}{\varepsilon} + \frac{\sigma^2}{\mu^2 R_0^2}\cdot\frac{\mu R_0^2}{\varepsilon} + \frac{n\sqrt{\delta}\sigma}{m\mu R_0}\cdot\sqrt{\frac{\mu R_0^2}{\varepsilon}} \right) \\
&= \mathcal{O}\left( \frac{L}{\mu}\log\frac{\mu R_0^2}{\varepsilon} + \frac{\sigma^2}{n\mu\varepsilon} + \frac{n\sqrt{\delta}\sigma}{m\sqrt{\mu\varepsilon}} \right).
\end{aligned}
$$

$\square$

In the strongly convex case, similar observations hold as in the non-convex case. Next, we derive the result without assuming that $\widehat{b}^k$ is known to all peers at each iteration.

**Theorem E.7.** *Let As. 3.1 and 3.2 hold, $Q = \mathbb{R}^d$, $f$ be L-smooth (see Def. E.1), $\mu$-strongly convex (see Def. E.2), and $x^*$ be some optimum of $f$. Moreover, assume that $b \leq 0.1(n - m)$, $m \leq (n-2b)/2$, and $\delta = 0$ is used to compute clipping parameter $\tau_l$ for CenteredClip. Next, assume that*

$$
\gamma_t = \min\left\{ \frac{1}{4L}, \sqrt{\frac{7nR_0^2}{120\cdot 2^t\sigma^2 K_t}}, \sqrt{\frac{m^2 R_0^2}{72\cdot 2^t C\sigma^2 n^2 b^2}} \right\}, \quad \Delta_{\max}^{k,t} = \frac{(1+\sqrt{3})\sqrt{2}\sigma}{\sqrt{n_k^t - m}}, \quad (47)
$$

$$
K_t = \left\lceil \max\left\{ \frac{16L}{\mu}, \frac{32\sigma^2 2^t}{\mu^2 R_0^2}, \frac{24\sqrt{2C}nb\sigma 2^{\frac{t}{2}}}{m\mu R_0} \right\} \right\rceil, \quad r = \left\lceil \log_2\frac{\mu R_0^2}{\varepsilon} \right\rceil - 1 \quad (48)
$$

*where $R_0 \geq \|x^0 - x^*\|$, $\Delta_{\max}^{k,t}$ is the parameter for verification 3 at iteration $k$ of BTARD-SGD during the t-th restart, $n_k^t$ is the total number of workers at iteration $k$ of t-th restart. Then, we have $\mathbb{E}[f(\widehat{x}^r) - f(x^*)] \leq \varepsilon$ after $r$ restarts of BTARD-SGD and the total number of executed iterations of BTARD-SGD is*

$$
\sum_{t=1}^{r} K_t = \mathcal{O}\left( \frac{L}{\mu}\log\frac{\mu R_0^2}{\varepsilon} + \frac{\sigma^2}{n\mu\varepsilon} + \frac{nb\sigma}{m\sqrt{\mu\varepsilon}} \right). \quad (49)
$$

*Proof.* Theorem E.5 implies that BTARD-SGD with

$$
\gamma = \min\left\{ \frac{1}{4L}, \sqrt{\frac{7nR_0^2}{120\sigma^2 K}}, \sqrt{\frac{m^2 R_0^2}{72C\sigma^2 n^2 b^2}} \right\}
$$

guarantees

$$
\mathbb{E}\left[ f(\overline{x}^K) - f(x^*) \right] \leq \frac{2R_0^2}{\gamma K}
$$

after $K$ iterations. Therefore, after the first restart we have

$$
\mathbb{E}[f(\widehat{x}^1) - f(x^*)] \leq \frac{2R_0^2}{\gamma_1 K_1} \leq \frac{\mu R_0^2}{4}.
$$

From $\mu$-strong convexity of $f$ and $\nabla f(x^*) = 0$ we have

$$
\frac{\mu}{2}\|\widehat{x}^1 - x^*\|^2 \leq f(\widehat{x}^1) - f(x^*) \implies \mathbb{E}[\|\widehat{x}^1 - x^*\|^2] \leq \frac{R_0^2}{2}.
$$

Next, assume that we have $\mathbb{E}[f(\widehat{x}^t) - f(x^*)] \leq \frac{\mu R_0^2}{2^{t+1}}$, $\mathbb{E}[\|\widehat{x}^t - x^*\|^2] \leq \frac{R_0^2}{2^t}$ for some $t \leq r - 1$. Then, Theorem E.5 implies that

$$
\mathbb{E}[f(\widehat{x}^{t+1}) - f(x^*) \mid x^t] \leq \frac{2\|\widehat{x}^t - x^*\|^2}{\gamma_t K_t}.
$$

Taking the full expectation from the both sides of previous inequality we get

$$\mathbb{E}[f(\widehat{x}^{t+1}) - f(x^*)] \leq \frac{2\mathbb{E}[\|\widehat{x}^t - x^*\|^2]}{\gamma_t K_t} \leq \frac{2R_0^2}{2^t \gamma_t K_t} \leq \frac{\mu R_0^2}{2^{t+2}}.$$

From $\mu$-strong convexity of $f$ and $\nabla f(x^*) = 0$ we have

$$\frac{\mu}{2}\|\widehat{x}^{t+1} - x^*\|^2 \leq f(\widehat{x}^{t+1}) - f(x^*) \implies \mathbb{E}[\|\widehat{x}^{t+1} - x^*\|^2] \leq \frac{R_0^2}{2^{t+1}}.$$

Therefore, by mathematical induction we have that for all $t = 1, \ldots, r$

$$\mathbb{E}[f(\widehat{x}^t) - f(x^*)] \leq \frac{\mu R_0^2}{2^{t+1}}, \quad \mathbb{E}\left[\|\widehat{x}^t - x^*\|^2\right] \leq \frac{R_0^2}{2^t}.$$

Then, after $r = \left\lceil \log_2 \frac{\mu R_0^2}{\varepsilon} \right\rceil - 1$ restarts of BTARD-SGD we have $\mathbb{E}[f(\widehat{x}^r) - f(x^*)] \leq \varepsilon$. The total number of iterations executed by BTARD-SGD is

$$
\begin{aligned}
\sum_{t=1}^r K_t &= \mathcal{O}\left(\sum_{t=1}^r \max\left\{\frac{L}{\mu}, \frac{\sigma^2 2^t}{\mu^2 R_0^2}, \frac{nb\sigma 2^{\frac{t}{2}}}{m\mu R_0}\right\}\right) \\
&= \mathcal{O}\left(\frac{L}{\mu}r + \frac{\sigma^2 2^r}{\mu^2 R_0^2} + \frac{nb\sigma 2^{\frac{r}{2}}}{m\mu R_0}\right) \\
&= \mathcal{O}\left(\frac{L}{\mu}\log\frac{\mu R_0^2}{\varepsilon} + \frac{\sigma^2}{\mu^2 R_0^2}\cdot\frac{\mu R_0^2}{\varepsilon} + \frac{nb\sigma}{m\mu R_0}\cdot\sqrt{\frac{\mu R_0^2}{\varepsilon}}\right) \\
&= \mathcal{O}\left(\frac{L}{\mu}\log\frac{\mu R_0^2}{\varepsilon} + \frac{\sigma^2}{n\mu\varepsilon} + \frac{nb\sigma}{m\sqrt{\mu\varepsilon}}\right).
\end{aligned}
$$

$\square$

### E.4 CONVERGENCE GUARANTEES FOR BTARD-CLIPPED-SGD

The results for BTARD-SGD and RESTARTED-BTARD-SGD rely on As. 3.2 that the stochastic gradients have not too heavy tails, i.e., sub-quadratically decreasing tails. The main reason why it is needed in the analysis is to prevent too often extra computations because of **Verification 3** from BTARD when all workers honestly follow the protocol. However, in many important NLP tasks such as BERT training (Zhang et al., 2020), the noise in the stochastic gradient has such a heavy noise that As. 3.2 becomes unnatural.

---

**Algorithm 8** BTARD-CLIPPED-SGD

**Input:** $x^0$ – starting point, $\gamma$ – stepsize, $K$ – number of iterations, $\{s_{i,k}\}_{i,k=0,0}^{n,K-1}$ – seeds for batches computations, $\{\lambda_k\}_{k=0}^{K-1}$ – gradient clipping parameter

1: $C_0 = \text{Banned}_{-1} = \varnothing$
2: **for** $k = 0, 1, \ldots, K-1$ **do**
3:      Worker $i$ computes $\widetilde{g}_i^k = \begin{cases} \min\left\{1, \frac{\lambda_k}{\|\nabla f(x^k, \xi_{i,k})\|}\right\}\nabla f(x^k, \xi_{i,k}), & \text{if } i \in \mathcal{G}_k \setminus \mathcal{C}_k, \\ *, & \text{if } i \in \mathcal{B}_k \setminus \mathcal{C}_k, \end{cases}$ , where
     $\xi_{i,k}$ is generated via seed $s_{i,k}$ available to every worker
4:      $(\widehat{g}^k, \text{public\_info}_k) = \text{BTARD}(\widetilde{g}_{i_1^k}^k, g_{i_1^k}^k, \ldots, \widetilde{g}_{i_{a_k}^k}^k)$, where $\{i_1^k, \ldots, i_{a_k}^k\} = (\mathcal{G}_k \cup \mathcal{B}_k) \setminus \mathcal{C}_k$
5:      Choose $2m$ workers $c_1^{k+1}, \ldots, c_m^{k+1}, u_1^{k+1}, \ldots, u_m^{k+1}$ uniformly at random without replacement, $\mathcal{C}_{k+1} = \{c_1^{k+1}, \ldots, c_m^{k+1}\}, \mathcal{U}_{k+1} = \{u_1^{k+1}, \ldots, u_m^{k+1}\}$
6:      $\text{Banned}_k = \text{CHECKCOMPUTATIONS}(\mathcal{C}_{k+1}, \mathcal{U}_{k+1}, \text{public\_info}_k)$
7:      $x^{k+1} = \text{proj}_Q(x^k - \gamma\widehat{g}^k) := \text{argmin}_{x \in Q}\|x - (x^k - \gamma\widehat{g}^k)\|$
8:      $\mathcal{G}_{k+1} = \mathcal{G}_k \setminus \text{Banned}_{k-1}$
9:      $\mathcal{B}_{k+1} = \mathcal{B}_k \setminus \text{Banned}_{k-1}$

---

To handle the problems with heavy-tailed noise distributions we consider BTARD-CLIPPED-SGD (see Alg. 8 in Appendix) applied to solve (3) such that $Q$ is bounded. Essentially, this algorithm

coincides with BTARD-SGD up to the following change: all good peers $i \in G_k \setminus C_k$ use clipped stochastic gradients $\widetilde{g}_i^k = (\widetilde{g}_i^k(1)^\top, \ldots, \widetilde{g}_i^k(n_k - m)^\top)^\top$, where $\widetilde{g}_i^k(l) = \min\left\{1, \frac{\lambda_k}{\|g_i^k(l)\|}\right\} g_i^k(l)$, $l = 1, \ldots, n_k - m$, and $g_i^k$ is the stochastic gradient. Next, we introduce the following assumption.

**Assumption E.1.** *There exist such constant $G > 0$, $s_0 \in [d]$, and $\alpha \in (1, 2]$ that for any set of indices $S = (i_1, \ldots, i_d)$, $1 \le i_1 < i_2 < \ldots < i_s \le d$, $s \ge s_0$ and arbitrary $x \in Q$ stochastic gradient $\nabla f(x, \xi)$ satisfy*

$$\mathbb{E}[\nabla f(x, \xi)] = \nabla f(x), \quad \mathbb{E}\left[\|\nabla_{[S]} f(x, \xi)\|^\alpha\right] \le \left(\frac{\sqrt{s}G}{\sqrt{d}}\right)^\alpha, \tag{50}$$

*where $\nabla_{[S]} f(x, \xi)$ is defined in As. 3.1.*

This is a modified version of the assumption used in Zhang et al. (2020). When $\alpha < 2$ the variance of the stochastic gradient can be unbounded. One can show that in such a regime vanilla SGD can diverge (Zhang et al., 2020).

Under As. E.1 we derive the convergence results for convex and strongly convex problems.

### E.4.1 QUALITY OF THE AGGREGATION

Since now we have As. E.1 instead of As. 3.1 and 3.2 it is needed to derive new guarantees for the quality of the aggregation. We start with the following useful lemma about the properties of clipped stochastic gradeints.

**Lemma E.5** (See also Lemma 9 from Zhang et al. (2020)). *Let As. E.1 holds and $i, j \in \mathcal{G}_k \setminus \mathcal{C}_k$. Then, for all $l = 1, 2, \ldots, n_k - m$ we have*

$$\sqrt{\mathbb{E}\left[\|\widetilde{g}_i^k(l) - \widetilde{g}_j^k(l)\|^4 \mid x^k\right]} \le 4\lambda_k^{\frac{4-\alpha}{2}} \left(\frac{G}{\sqrt{n_k - m}}\right)^{\frac{\alpha}{2}}, \tag{51}$$

$$\mathbb{E}\left[\|\overline{g}^k(l)\|^2 \mid x^k\right] \le \frac{G^\alpha \lambda_k^{2-\alpha}}{(n_k - m)^{\frac{\alpha}{2}}}, \tag{52}$$

$$\left\|\mathbb{E}[\overline{g}^k(l) \mid x^k] - \nabla_{(l)} f(x^k)\right\|^2 \le \frac{G^{2\alpha}}{(n_k - m)^\alpha \lambda_k^{2(\alpha-1)}}, \tag{53}$$

*where $\overline{g}^k(l) = \frac{1}{|\mathcal{G}_k \setminus \mathcal{C}_k|} \sum\limits_{i \in \mathcal{G}_k \setminus \mathcal{C}_k} \widetilde{g}_i^k(l)$ for all $l = 1, \ldots, n_k - m$.*

*Proof.* First of all, we derive

$$
\begin{aligned}
\mathbb{E}\left[\|\widetilde{g}_i^k(l) - \widetilde{g}_j^k(l)\|^4 \mid x^k\right] &= \mathbb{E}\left[\|\widetilde{g}_i^k(l) - \widetilde{g}_j^k(l)\|^\alpha \|\widetilde{g}_i^k(l) - \widetilde{g}_j^k(l)\|^{4-\alpha} \mid x^k\right] \\
&\le 8\lambda_k^{4-\alpha} \mathbb{E}\left[\|\nabla_{(l)} f(x^k, \xi_{i,k})\|^\alpha + \|\nabla_{(l)} f(x^k, \xi_{j,k})\|^\alpha \mid x^k\right] \\
&\overset{(50)}{\le} 16\lambda_k^{4-\alpha} \left(\frac{G}{\sqrt{n_k - m}}\right)^\alpha
\end{aligned}
$$

implying (51). Next, for all $i \in \mathcal{G}_k \setminus \mathcal{C}_k$ we have

$$
\begin{aligned}
\mathbb{E}\left[\|\widetilde{g}_i^k(l)\|^2 \mid x^k\right] &= \mathbb{E}\left[\|\widetilde{g}_i^k(l)\|^\alpha \|\widetilde{g}_i^k(l)\|^{2-\alpha} \mid x^k\right] \le \lambda_k^{2-\alpha} \mathbb{E}\left[\|\nabla_{(l)} f(x^k, \xi_{i,k})\|^\alpha \mid x^k\right] \\
&\overset{(50)}{\le} \frac{G^\alpha \lambda_k^{2-\alpha}}{(n_k - m)^{\frac{\alpha}{2}}}
\end{aligned}
$$

implying

$$\mathbb{E}\left[\|\overline{g}^k(l)\|^2 \mid x^k\right] \le \frac{1}{|\mathcal{G}_k \setminus \mathcal{C}_k|} \sum_{i \in \mathcal{G}_k \setminus \mathcal{C}_k} \mathbb{E}\left[\|\widetilde{g}_i^k(l)\|^2 \mid x^k\right] \le \frac{G^\alpha \lambda_k^{2-\alpha}}{(n_k - m)^{\frac{\alpha}{2}}}.$$

Finally, for all $i \in \mathcal{G}_k \setminus \mathcal{C}_k$ we derive

$$
\begin{aligned}
\left\| \mathbb{E}[\widetilde{g}_i^k(l) \mid x^k] - \nabla_{(l)} f(x^k) \right\| &= \left\| \mathbb{E}[\widetilde{g}_i^k(l) - \nabla_{(l)} f(x^k, \xi_{i,k}) \mid x^k] \right\| \\
&\leq \mathbb{E}\left[ \left\| \widetilde{g}_i^k(l) - \nabla_{(l)} f(x^k, \xi_{i,k}) \right\| \mid x^k \right] \\
&= \mathbb{E}\left[ \left\| \widetilde{g}_i^k(l) - \nabla_{(l)} f(x^k, \xi_{i,k}) \right\| \mathbb{1}_{\{\|\nabla_{(l)} f(x^k, \xi_{i,k})\| \geq \lambda_k\}} \mid x^k \right] \\
&\leq \mathbb{E}\left[ \left\| \nabla_{(l)} f(x^k, \xi_{i,k}) \right\| \mathbb{1}_{\{\|\nabla_{(l)} f(x^k, \xi_{i,k})\| \geq \lambda_k\}} \mid x^k \right] \\
&\leq \frac{\mathbb{E}\left[ \left\| \nabla_{(l)} f(x^k, \xi_{i,k}) \right\|^\alpha \mathbb{1}_{\{\|\nabla_{(l)} f(x^k, \xi_{i,k})\| \geq \lambda_k\}} \mid x^k \right]}{\lambda_k^{\alpha-1}} \\
&\overset{(50)}{\leq} \frac{G^\alpha}{(n_k - m)^{\frac{\alpha}{2}} \lambda_k^{\alpha-1}}
\end{aligned}
$$

implying

$$
\begin{aligned}
\left\| \mathbb{E}[\overline{g}^k(l) \mid x^k] - \nabla_{(l)} f(x^k) \right\|^2 &\leq \frac{1}{|\mathcal{G}_k \setminus \mathcal{C}_k|} \sum_{i \in \mathcal{G}_k \setminus \mathcal{C}_k} \mathbb{E}\left[ \|\widetilde{g}_i^k(l) - \nabla_{(l)} f(x^k)\|^2 \mid x^k \right] \\
&\leq \frac{G^{2\alpha}}{(n_k - m)^\alpha \lambda_k^{2(\alpha-1)}}.
\end{aligned}
$$

$\square$

Next, we derive the guarantees for the quality of the aggregation in the case when the number of Byzantine peers violating the protocol $\widehat{b}_k$ is known at each iteration.

**Lemma E.6.** *Let As. E.1 hold and $b \leq 0.15(n - m)$. Assume that $\widehat{b}_k$ is known for each worker at iteration $k$, $\Delta_{\max}^k = 2\lambda_k = \frac{2\lambda}{\sqrt{n_k - m}}$ and $\delta = \widehat{\delta}_k$ is used to compute clipping parameter $\tau_l$ for* CenteredClip. *If the total number of iterations $T$ of* CenteredClip *satisfies $T \geq \log_{0.94} \frac{2\delta\sigma^2}{\mathbb{E}[\|v^0 - \overline{g}^k\|^2]}$ and* CHECKAVERAGING *is not triggered for any worker, then*

$$
\mathbb{E}\left[ \|\widehat{g}^k - \overline{g}^k\|^2 \mid x^k \right] \leq \widehat{\delta}_k (C_1 \lambda^{\frac{4-\alpha}{2}} G^{\frac{\alpha}{2}} + C_2 \lambda^2), \tag{54}
$$

$$
\mathbb{E}\left[ \|\widehat{g}^k\|^2 \mid x^k \right] \leq 2\widehat{\delta}_k (C_1 \lambda^{\frac{4-\alpha}{2}} G^{\frac{\alpha}{2}} + C_2 \lambda^2) + 2 G^\alpha \lambda^{2-\alpha}, \tag{55}
$$

*where $\overline{g}^k = \frac{1}{|\mathcal{G}_k \setminus \mathcal{C}_k|} \sum_{j \in \mathcal{G}_k \setminus \mathcal{C}_k} g_j^k$, $C_1 = 384$, and $C_2 = 4$.*

*Proof.* Consider the $i$-th part of $\widehat{g}^k$, i.e., consider $\widehat{g}^k(i)$. If $i \in \mathcal{G}_k \setminus \mathcal{C}_k$, then, in view of (51), we can directly apply Lemma E.1 and get

$$
\mathbb{E}\left[ \|\widehat{g}^k(i) - \overline{g}^k(i)\|^2 \mid x^k \right] \leq 384 \widehat{\delta}_k \lambda_k^{\frac{4-\alpha}{2}} \frac{G^{\frac{\alpha}{2}}}{(n_k - m)^{\frac{\alpha}{4}}} = \frac{384 \widehat{\delta}_k \lambda^{\frac{4-\alpha}{2}} G^{\frac{\alpha}{2}}}{n_k - m}.
$$

Next, if $i \in \mathcal{B}_k \setminus \mathcal{C}_k$, then

$$
\mathbb{E}\left[ \|\widehat{g}^k(i) - \overline{g}^k(i)\|^2 \mid x^k \right] \leq (\Delta_{\max}^k)^2 = 4\lambda_k^2 = \frac{4\lambda^2}{n_k - m}.
$$

Putting all together, we derive

$$
\begin{aligned}
\mathbb{E}\left[ \|\widehat{g}^k - \overline{g}^k\|^2 \mid x^k \right] &= \sum_{i \in \mathcal{G}_k \setminus \mathcal{C}_k} \mathbb{E}\left[ \|\widehat{g}^k(i) - \overline{g}^k(i)\|^2 \mid x^k \right] + \sum_{i \in \mathcal{B}_k \setminus \mathcal{C}_k} \mathbb{E}\left[ \|\widehat{g}^k(i) - \overline{g}^k(i)\|^2 \mid x^k \right] \\
&\leq (1 - \widehat{\delta}_k)(n_k - m) \cdot \frac{384 \widehat{\delta}_k \lambda^{\frac{4-\alpha}{2}} G^{\frac{\alpha}{2}}}{n_k - m} + \widehat{\delta}_k (n_k - m) \cdot \frac{4\lambda^2}{n_k - m} \\
&\leq \widehat{\delta}_k (C_1 \lambda^{\frac{4-\alpha}{2}} G^{\frac{\alpha}{2}} + C_2 \lambda^2).
\end{aligned}
$$

Using (52) we obtain

$$
\begin{aligned}
\mathbb{E}\left[\|\widehat{g}^k\|^2 \mid x^k\right] &\leq 2\mathbb{E}\left[\|\widehat{g}^k - \overline{g}^k\|^2 \mid x^k\right] + 2\mathbb{E}\left[\|\overline{g}^k\|^2 \mid x^k\right] \\
&\overset{(54)}{\leq} 2\widehat{\delta}_k(C_1\lambda^{\frac{4-\alpha}{2}}G^{\frac{\alpha}{2}} + C_2\lambda^2) + 2\sum_{i \in (\mathcal{G}_k \cup \mathcal{B}_k)\setminus\mathcal{C}_k} \frac{G^\alpha \lambda_k^{2-\alpha}}{(n_k - m)^{\frac{\alpha}{2}}} \\
&= 2\widehat{\delta}_k(C_1\lambda^{\frac{4-\alpha}{2}}G^{\frac{\alpha}{2}} + C_2\lambda^2) + 2G^\alpha \lambda^{2-\alpha}.
\end{aligned}
$$

$\square$

We notice that **Verification 3** can be simplified in the following way: if at least on good peer $i$ notices that $\|\widetilde{g}_i^k(j) - \widehat{g}^k(j)\| > \Delta_{\max}^k = 2\lambda_k$, then peer $i$ should accuse $j$-th peer and both are removed from the training process. In this scenario, there is no sense for Byzantine workers in triggering to deviate significantly from the clipped stochastic gradients of the good peers.

As for BTARD-SGD, when $\widehat{b}_k$ is unknown we always use CENTEREDCLIP with $\tau_l = \infty$ for all $l \geq 0$, i.e., good peers compute an exact average. In this settings, even 1 Byzantine worker can significantly shift the average in all parts of the vector. The next lemma quantifies the negative effect of Byzantine workers in this case.

**Lemma E.7.** *Let As. E.1 hold and $b \leq 0.15(n-m)$. Assume that $\widehat{b}_k$ is known for each worker at iteration $k$, $\Delta_{\max}^k = 2\lambda_k = \frac{2\lambda}{\sqrt{n_k - m}}$ and $\delta = \widehat{\delta}_k$ is used to compute clipping parameter $\tau_l$ for CenteredClip. If the total number of iterations $T$ of CenteredClip satisfies $T \geq \log_{0.94}\frac{2\delta\sigma^2}{\mathbb{E}[\|v^0 - \overline{g}^k\|^2]}$ and* CHECKAVERAGING *is not triggered for any worker, then*

$$
\mathbb{E}\left[\|\widehat{g}^k - \overline{g}^k\|^2 \mid x^k\right] \leq C_2\lambda^2 \mathbb{1}_{k,v}, \tag{56}
$$

$$
\mathbb{E}\left[\|\widehat{g}^k\|^2 \mid x^k\right] \leq 2C_2\lambda^2 \mathbb{1}_{k,v} + 2G^\alpha \lambda^{2-\alpha}, \tag{57}
$$

*where $\overline{g}^k = \frac{1}{|\mathcal{G}_k \setminus \mathcal{C}_k|}\sum_{j \in \mathcal{G}_k \setminus \mathcal{C}_k} g_j^k$, $C_2 = 4$, and $\mathbb{1}_{k,v}$ is an indicator function of the event that at least $1$ Byzantine peer violates the protocol at iteration $k$.*

*Proof.* For all $i \in (\mathcal{G}_k \cup \mathcal{B}_k)\setminus\mathcal{C}_k$ we have

$$
\mathbb{E}\left[\|\widehat{g}^k(i) - \overline{g}^k(i)\|^2 \mid x^k\right] \leq (\Delta_{\max}^k)^2 \mathbb{1}_{k,v} = 4\lambda_k^2 \mathbb{1}_{k,v} = \frac{4\lambda^2}{n_k - m}\mathbb{1}_{k,v}
$$

implying

$$
\begin{aligned}
\mathbb{E}\left[\|\widehat{g}^k - \overline{g}^k\|^2 \mid x^k\right] &= \sum_{i \in (\mathcal{G}_k \cup \mathcal{B}_k)\setminus\mathcal{C}_k} \mathbb{E}\left[\|\widehat{g}^k(i) - \overline{g}^k(i)\|^2 \mid x^k\right] \\
&\leq (n_k - m)\cdot \frac{4\lambda^2}{n_k - m}\mathbb{1}_{k,v} = C_2\lambda^2 \mathbb{1}_{k,v}.
\end{aligned}
$$

Using (52) we obtain

$$
\begin{aligned}
\mathbb{E}\left[\|\widehat{g}^k\|^2 \mid x^k\right] &\leq 2\mathbb{E}\left[\|\widehat{g}^k - \overline{g}^k\|^2 \mid x^k\right] + 2\mathbb{E}\left[\|\overline{g}^k\|^2 \mid x^k\right] \\
&\overset{(54)}{\leq} 2C_2\lambda^2 \mathbb{1}_{k,v} + 2\sum_{i \in (\mathcal{G}_k \cup \mathcal{B}_k)\setminus\mathcal{C}_k} \frac{G^\alpha \lambda_k^{2-\alpha}}{(n_k - m)^{\frac{\alpha}{2}}} = 2C_2\lambda^2 \mathbb{1}_{k,v} + 2G^\alpha \lambda^{2-\alpha}.
\end{aligned}
$$

$\square$

### E.4.2 CONVEX CASE

In this section, we provide the complete statements and the full proofs of the convergence results for BTARD-CLIPPED-SGD when the objective function $f$ is smooth and convex. We start with the case when the number of Byzantine peers violating the protocol $\widehat{b}_k$ is known at each iteration.

**Theorem E.8.** *Let As. E.1 hold, $Q$ is bounded, $f$ be convex, $x^*$ be some optimum of $f$, and $\nabla f(x^*) = 0$. Moreover, assume that $b \le 0.15(n-m)$, $m \le {}^{(n-2b)}/_2$, and the exact number of attacking Byzantine peers is known to all good peers at each iteration. Next, assume that*

$$\gamma = \min\left\{\frac{R_0}{\sqrt{6}GK^{\frac{1}{\alpha}}}, \frac{mR_0}{12Gn\sqrt{10\delta(C_1K^{\frac{4-\alpha}{2\alpha}} + C_2K^{\frac{2}{\alpha}})}}\right\}, \quad \Delta_{\max}^k = 2\lambda_k = \frac{2\lambda}{\sqrt{n_k - m}}, \quad (58)$$

$$\lambda = GK^{\frac{1}{\alpha}}, \tag{59}$$

*where $R_0 \ge \|x^0 - x^*\|$ and $\Delta_{\max}^k$ is the parameter for verification 3 at iteration $k$ of BTARD-CLIPPED-SGD. Then, we have $\mathbb{E}[f(\overline{x}^K) - f(x^*)] \le \varepsilon$ after $K$ iterations of BTARD-CLIPPED-SGD, where*

$$K = \mathcal{O}\left(\left(\frac{GR_0}{\varepsilon}\right)^{\frac{\alpha}{\alpha-1}} + \left(\frac{n\sqrt{\delta}GR_0}{m\varepsilon}\right)^{\frac{\alpha}{\alpha-1}}\right) \tag{60}$$

*and $\overline{x}^K = \frac{1}{K}\sum_{k=0}^{K-1}.$*

*Proof.* Non-expansiveness of the projection operator and convexity of $f$ imply

$$
\begin{aligned}
\|x^{k+1} - x^*\|^2 &= \left\|\mathrm{proj}_Q(x^k - \gamma\widehat{g}^k) - \mathrm{proj}_Q(x^*)\right\|^2 \\
&\le \|x^k - x^* - \gamma\widehat{g}^k\|^2 \\
&= \|x^k - x^*\|^2 - 2\gamma\langle x^k - x^*, \widehat{g}^k\rangle + \gamma^2\|\widehat{g}^k\|^2 \\
&= \|x^k - x^*\|^2 - 2\gamma\langle x^k - x^*, \nabla f(x^k)\rangle - 2\gamma\langle x^k - x^*, \widehat{g}^k - \nabla f(x^k)\rangle + \gamma^2\|\widehat{g}^k\|^2 \\
&\le \|x^k - x^*\|^2 - 2\gamma\left(f(x^k) - f(x^*)\right) - 2\gamma\langle x^k - x^*, \widehat{g}^k - \nabla f(x^k)\rangle + \gamma^2\|\widehat{g}^k\|^2.
\end{aligned}
$$

Taking conditional expectation $\mathbb{E}[\cdot \mid x^k]$ from the both sides of previous inequality we derive

$$
\begin{aligned}
\mathbb{E}\left[\|x^{k+1} - x^*\|^2 \mid x^k\right] &\le \|x^k - x^*\|^2 - 2\gamma\left(f(x^k) - f(x^*)\right) \\
&\quad - 2\gamma\mathbb{E}\left[\langle x^k - x^*, \widehat{g}^k - \nabla f(x^k)\rangle \mid x^k\right] + \gamma^2\mathbb{E}\left[\|\widehat{g}^k\|^2 \mid x^k\right] \\
&\overset{(55)}{\le} \|x^k - x^*\|^2 - 2\gamma\left(f(x^k) - f(x^*)\right) + 2\gamma^2 G^\alpha\lambda^{2-\alpha} \\
&\quad - 2\gamma\left\langle x^k - x^*, \mathbb{E}\left[\widehat{g}^k - \overline{g}^k \mid x^k\right]\right\rangle + 2\gamma^2\widehat{\delta}_k(C_1\lambda^{\frac{4-\alpha}{2}}G^{\frac{\alpha}{2}} + C_2\lambda^2) \\
&= \|x^k - x^*\|^2 - 2\gamma\left(f(x^k) - f(x^*)\right) + 2\gamma^2 G^2 K^{\frac{2-\alpha}{\alpha}} \\
&\quad - 2\gamma\left\langle x^k - x^*, \mathbb{E}\left[\widehat{g}^k - \overline{g}^k \mid x^k\right]\right\rangle + \frac{2\gamma^2 G^2(C_1 K^{\frac{4-\alpha}{2\alpha}} + C_2 K^{\frac{2}{\alpha}})}{n_k - m}\widehat{b}_k.
\end{aligned}
$$

To estimate the inner product in the right-hand side we apply Cauchy-Schwarz inequality:

$$
\begin{aligned}
-2\gamma\left\langle x^k - x^*, \mathbb{E}\left[\widehat{g}^k - \overline{g}^k \mid x^k\right]\right\rangle &\le 2\gamma\|x^k - x^*\| \cdot \left\|\mathbb{E}\left[\widehat{g}^k - \overline{g}^k \mid x^k\right]\right\| \\
&\le 2\gamma\|x^k - x^*\|\mathbb{E}\left[\|\widehat{g}^k - \overline{g}^k\| \mid x^k\right] \\
&\le 2\gamma\|x^k - x^*\|\sqrt{\mathbb{E}\left[\|\widehat{g}^k - \overline{g}^k\|^2 \mid x^k\right]} \\
&\overset{(54)}{\le} 2\gamma\|x^k - x^*\|\sqrt{\widehat{\delta}_k(C_1\lambda^{\frac{4-\alpha}{2}}G^{\frac{\alpha}{2}} + C_2\lambda^2)} \\
&= \frac{2\gamma G\|x^k - x^*\|\sqrt{C_1 K^{\frac{4-\alpha}{2\alpha}} + C_2 K^{\frac{2}{\alpha}}}}{\sqrt{n_k - m}}\sqrt{\widehat{b}_k} \\
&\le \frac{2\gamma G\|x^k - x^*\|\sqrt{20(C_1 K^{\frac{4-\alpha}{2\alpha}} + C_2 K^{\frac{2}{\alpha}})}}{\sqrt{7n}}\sqrt{\widehat{b}_k},
\end{aligned}
$$

where in the last inequality we use $b \leq 0.15(n-m)$, $m \leq (n-2b)/2$, $\gamma \leq 1/4L$, $n_k - m \geq n - 2b - m \geq \frac{7}{20}n$. Putting all together we obtain

$$
\begin{aligned}
\mathbb{E}\left[\|x^{k+1} - x^*\|^2 \mid x^k\right] &\leq \|x^k - x^*\|^2 - 2\gamma\left(f(x^k) - f(x^*)\right) + 2\gamma^2 G^2 K^{\frac{2-\alpha}{\alpha}} \\
&\quad + \frac{2\gamma G\|x^k - x^*\|\sqrt{20(C_1 K^{\frac{4-\alpha}{2\alpha}} + C_2 K^{\frac{2}{\alpha}})}}{\sqrt{7n}}\sqrt{\widehat{b_k}} \\
&\quad + \frac{40\gamma^2 G^2(C_1 K^{\frac{4-\alpha}{2\alpha}} + C_2 K^{\frac{2}{\alpha}})}{7n}\widehat{b_k}.
\end{aligned}
$$

Taking the full expectation from the both sides of the above inequality and summing up the results for $k = 0, 1, \ldots, T-1$ we derive

$$
\begin{aligned}
\frac{2\gamma}{T}\sum_{k=0}^{T-1}\mathbb{E}[f(x^k) - f(x^*)] &\leq \frac{1}{T}\sum_{k=0}^{T-1}\left(\mathbb{E}\left[\|x^k - x^*\|^2\right] - \mathbb{E}\left[\|x^{k+1} - x^*\|^2\right]\right) + 2\gamma^2 G^2 K^{\frac{2-\alpha}{\alpha}} \\
&\quad + \frac{4\gamma G\sqrt{5(C_1 K^{\frac{4-\alpha}{2\alpha}} + C_2 K^{\frac{2}{\alpha}})}}{\sqrt{n}T}\sum_{k=0}^{T-1}\mathbb{E}\left[\|x^k - x^*\|\sqrt{\widehat{b_k}}\right] \\
&\quad + \frac{40\gamma^2 G^2(C_1 K^{\frac{4-\alpha}{2\alpha}} + C_2 K^{\frac{2}{\alpha}})}{7nT}\sum_{k=0}^{T-1}\mathbb{E}[\widehat{b_k}] \\
&\leq \frac{\|x^0 - x^*\|^2 - \mathbb{E}[\|x^K - x^*\|^2]}{K} + 2\gamma^2 G^2 K^{\frac{2-\alpha}{\alpha}} \\
&\quad + \frac{4\gamma G\sqrt{5(C_1 K^{\frac{4-\alpha}{2\alpha}} + C_2 K^{\frac{2}{\alpha}})}}{\sqrt{n}T}\sum_{k=0}^{T-1}\sqrt{\mathbb{E}\left[\|x^k - x^*\|^2\right]\mathbb{E}\left[\widehat{b_k}\right]} \\
&\quad + \frac{40\gamma^2 G^2(C_1 K^{\frac{4-\alpha}{2\alpha}} + C_2 K^{\frac{2}{\alpha}})}{7nT}\sum_{k=0}^{T-1}\mathbb{E}[\widehat{b_k}].
\end{aligned}
$$

From Jensen's inequality we have $f(\overline{x}^T) \leq \frac{1}{T}\sum_{k=0}^{T-1}f(x^k)$, where $\overline{x}^T = \frac{1}{T}\sum_{k=0}^{T-1}x^k$. Using this and new notation $R_k = \|x^k - x^*\|$, $k > 0$, $R_0 \geq \|x^0 - x^*\|$ we get

$$
\begin{aligned}
0 \leq 2\gamma\mathbb{E}\left[f(\overline{x}^T) - f(x^*)\right] &\leq \frac{R_0^2 - \mathbb{E}[R_T^2]}{T} + 2\gamma^2 G^2 K^{\frac{2-\alpha}{\alpha}} \\
&\quad + \frac{4\gamma G\sqrt{5(C_1 K^{\frac{4-\alpha}{2\alpha}} + C_2 K^{\frac{2}{\alpha}})}}{\sqrt{n}T}\sum_{k=0}^{T-1}\sqrt{\mathbb{E}\left[R_k^2\right]\mathbb{E}\left[\widehat{b_k}\right]} \\
&\quad + \frac{40\gamma^2 G^2(C_1 K^{\frac{4-\alpha}{2\alpha}} + C_2 K^{\frac{2}{\alpha}})}{7nT}\sum_{k=0}^{T-1}\mathbb{E}[\widehat{b_k}] \quad (61)
\end{aligned}
$$

implying (after changing the indices) that

$$
\begin{aligned}
\mathbb{E}[R_k^2] &\leq R_0^2 + 2\gamma^2 G^2 k K^{\frac{2-\alpha}{\alpha}} + \frac{4\gamma G\sqrt{5(C_1 K^{\frac{4-\alpha}{2\alpha}} + C_2 K^{\frac{2}{\alpha}})}}{\sqrt{n}}\sum_{l=0}^{k-1}\sqrt{\mathbb{E}\left[R_l^2\right]\mathbb{E}\left[\widehat{b_l}\right]} \\
&\quad + \frac{40\gamma^2 G^2(C_1 K^{\frac{4-\alpha}{2\alpha}} + C_2 K^{\frac{2}{\alpha}})}{7n}\sum_{l=0}^{k-1}\mathbb{E}[\widehat{b_l}] \quad (62)
\end{aligned}
$$

holds for all $k \geq 0$. In the remaining part of the proof we derive by induction that

$$
\begin{aligned}
R_0^2 + 2\gamma^2 G^2 k K^{\frac{2-\alpha}{\alpha}} &+ \frac{4\gamma G\sqrt{5(C_1 K^{\frac{4-\alpha}{2\alpha}} + C_2 K^{\frac{2}{\alpha}})}}{\sqrt{n}}\sum_{l=0}^{k-1}\sqrt{\mathbb{E}\left[R_l^2\right]\mathbb{E}\left[\widehat{b_l}\right]} \\
&+ \frac{40\gamma^2 G^2(C_1 K^{\frac{4-\alpha}{2\alpha}} + C_2 K^{\frac{2}{\alpha}})}{7n}\sum_{l=0}^{k-1}\mathbb{E}[\widehat{b_l}] \leq 2R_0^2 \quad (63)
\end{aligned}
$$

for all $k = 0, \ldots, K$. For $k = 0$ this inequality trivially holds. Next, assume that it holds for all $k = 0, 1, \ldots, T - 1, T \le K - 1$. Let us show that it holds for $k = T$ as well. From (37) and (38) we have that $\mathbb{E}[R_k^2] \le 2R_0^2$ for all $k = 0, 1, \ldots, T - 1$. Therefore,

$$
\begin{aligned}
\mathbb{E}[R_T^2] \;\le\;& R_0^2 + 2\gamma^2 G^2 T K^{\frac{2-\alpha}{\alpha}} + \frac{4\gamma G \sqrt{5(C_1 K^{\frac{4-\alpha}{2\alpha}} + C_2 K^{\frac{2}{\alpha}})}}{\sqrt{n}} \sum_{l=0}^{T-1} \sqrt{\mathbb{E}\left[R_l^2\right] \mathbb{E}\left[\widehat{b}_l\right]} \\
&+ \frac{40\gamma^2 G^2 (C_1 K^{\frac{4-\alpha}{2\alpha}} + C_2 K^{\frac{2}{\alpha}})}{7n} \sum_{l=0}^{T-1} \mathbb{E}[\widehat{b}_l] \\
\le\;& R_0^2 + 2\gamma^2 G^2 T K^{\frac{2-\alpha}{\alpha}} + \frac{4\gamma G R_0 \sqrt{10(C_1 K^{\frac{4-\alpha}{2\alpha}} + C_2 K^{\frac{2}{\alpha}})}}{\sqrt{n}} \sum_{l=0}^{T-1} \sqrt{\mathbb{E}\left[\widehat{b}_l\right]} \\
&+ \frac{40\gamma^2 G^2 (C_1 K^{\frac{4-\alpha}{2\alpha}} + C_2 K^{\frac{2}{\alpha}})}{7n} \sum_{l=0}^{T-1} \mathbb{E}[\widehat{b}_l]
\end{aligned}
$$

If a Byzantine peer deviates from the protocol at iteration $k$, it will be detected with some probability $p_k$ during the next iteration. One can lower bound this probability as

$$
p_k \ge m \cdot \frac{|\mathcal{G}_k|}{n_k} \cdot \frac{1}{n_k} = \frac{m(1 - \delta_k)}{n_k} \ge \frac{m}{n}.
$$

Therefore, each individual Byzantine worker can violate the protocol no more than $1/p$ times on average implying that

$$
\begin{aligned}
\mathbb{E}[R_T^2] \;\le\;& R_0^2 + 2\gamma^2 G^2 T K^{\frac{2-\alpha}{\alpha}} + \frac{4\gamma G R_0 n \sqrt{10(C_1 K^{\frac{4-\alpha}{2\alpha}} + C_2 K^{\frac{2}{\alpha}})b}}{m\sqrt{n}} \\
&+ \frac{40\gamma^2 G^2 (C_1 K^{\frac{4-\alpha}{2\alpha}} + C_2 K^{\frac{2}{\alpha}})nb}{7nm} \\
\overset{T \le K}{\le}\;& R_0^2 + 2\gamma^2 G^2 K^{\frac{2}{\alpha}} + \frac{4\gamma G R_0 n \sqrt{10(C_1 K^{\frac{4-\alpha}{2\alpha}} + C_2 K^{\frac{2}{\alpha}})\delta}}{m} \\
&+ \frac{40\gamma^2 G^2 (C_1 K^{\frac{4-\alpha}{2\alpha}} + C_2 K^{\frac{2}{\alpha}})n\delta}{7m}.
\end{aligned}
$$

Taking

$$
\gamma = \min\left\{ \frac{R_0}{\sqrt{6}GK^{\frac{1}{\alpha}}}, \frac{mR_0}{12Gn\sqrt{10\delta(C_1 K^{\frac{4-\alpha}{2\alpha}} + C_2 K^{\frac{2}{\alpha}})}} \right\}
$$

we ensure that

$$
\begin{aligned}
2\gamma^2 G^2 K^{\frac{2}{\alpha}} + \frac{4\gamma G R_0 n \sqrt{10(C_1 K^{\frac{4-\alpha}{2\alpha}} + C_2 K^{\frac{2}{\alpha}})\delta}}{m} & \\
+ \frac{40\gamma^2 G^2 (C_1 K^{\frac{4-\alpha}{2\alpha}} + C_2 K^{\frac{2}{\alpha}})n\delta}{7m} \;\le\;& \frac{R_0^2}{3} + \frac{R_0^2}{3} + \frac{R_0^2}{3} = R_0^2
\end{aligned}
$$

and, as a result, we get $\mathbb{E}[R_T^2] \le 2R_0^2$. Therefore, (63) holds for all $k = 0, 1, \ldots, K$. Together with (61) it implies

$$
\mathbb{E}\left[f(\overline{x}^K) - f(x^*)\right] \le \frac{R_0^2}{\gamma K}.
$$

Next, from our stepsize rule (58) it follows that

$$
\mathbb{E}\left[f(\overline{x}^K) - f(x^*)\right] = \mathcal{O}\left( \frac{GR_0}{K^{\frac{1-\alpha}{\alpha}}} + \frac{n\sqrt{\delta}GR_0}{mK^{\frac{1-\alpha}{\alpha}}} \right)
$$

meaning that after

$$K = \mathcal{O}\left(\left(\frac{GR_0}{\varepsilon}\right)^{\frac{\alpha}{\alpha-1}} + \left(\frac{n\sqrt{\delta}GR_0}{m\varepsilon}\right)^{\frac{\alpha}{\alpha-1}}\right)$$

iterations BTARD-CLIPPED-SGD guarantees $\mathbb{E}[f(\overline{x}^K) - f(x^*)] \le \varepsilon$. $\qquad\square$

If there are no Byzantine peers ($\delta = 0$), the theorem establishes new result for the convergence of CLIPPED-SGD for convex objectives. In the strongly convex case, the theorem recovers the rates that are optimal in this setting as shown in Zhang et al. (2020). Next, when the number of attacking Byzantines is known at each iteration and $n\sqrt{\delta}/m = \mathcal{O}(1)$, the complexity bound is the same as in the case when $\delta = 0$. This means that the negative impact of Byzantine workers is negligible. Finally, the derived theoretical guarantees do not benefit from the increase of the total number of peers $n$. However, the result holds even for non-smooth problems and it is known that parallelization does not help to improve the complexity bounds in such generality. Nevertheless, our results show that BTARD-CLIPPED-SGD provably converges to any predefined accuracy $\varepsilon > 0$. This is a property that the majority of previous methods does not have (Karimireddy et al., 2020).

Next, we derive the result without assuming that $\widehat{b}^k$ is known to all peers at each iteration.

**Theorem E.9.** *Let As. E.1 hold, $Q$ is bounded, $f$ be convex, $x^*$ be some optimum of $f$, and $\nabla f(x^*) = 0$. Moreover, assume that $b \le 0.15(n-m)$, $m \le (n-2b)/2$, and $\delta = 0$ is used to compute clipping parameter $\tau_l$ for CenteredClip. Next, assume that*

$$\gamma = \min\left\{\frac{R_0}{\sqrt{6}GK^{\frac{1}{\alpha}}}, \frac{mR_0}{12\sqrt{2C_2}GnbK^{\frac{1}{\alpha}}}\right\}, \quad \Delta_{\max}^k = 2\lambda_k = \frac{2\lambda}{\sqrt{n_k - m}}, \tag{64}$$

$$\lambda = GK^{\frac{1}{\alpha}}, \tag{65}$$

*where $R_0 \ge \|x^0 - x^*\|$ and $\Delta_{\max}^k$ is the parameter for verification 3 at iteration $k$ of BTARD-CLIPPED-SGD. Then, we have $\mathbb{E}[f(\overline{x}^K) - f(x^*)] \le \varepsilon$ after $K$ iterations of BTARD-CLIPPED-SGD, where*

$$K = \mathcal{O}\left(\left(\frac{GR_0}{\varepsilon}\right)^{\frac{\alpha}{\alpha-1}} + \left(\frac{nbGR_0}{m\varepsilon}\right)^{\frac{\alpha}{\alpha-1}}\right) \tag{66}$$

*and $\overline{x}^K = \frac{1}{K}\sum_{k=0}^{K-1}$.*

*Proof.* The proof is almost identical to the proof of Theorem E.8. Following the same steps and using (56) and (57) instead of (54) and (55) respectively we obtain the same sequence of inequalities up to the following change: instead of $\widehat{\delta}_k$ we should use $\mathbb{1}_{k,v}$. Therefore, we have

$$\begin{aligned}
\mathbb{E}\left[\|x^{k+1} - x^*\|^2 \mid x^k\right] &\le \|x^k - x^*\|^2 - 2\gamma\left(f(x^k) - f(x^*)\right) + 2\gamma^2G^2K^{\frac{2-\alpha}{\alpha}} \\
&\quad - 2\gamma\left\langle x^k - x^*, \mathbb{E}\left[\widehat{g}^k - \overline{g}^k \mid x^k\right]\right\rangle + 2\gamma^2C_2G^2K^{\frac{2}{\alpha}}\mathbb{1}_{k,v}.
\end{aligned}$$

$$-2\gamma\left\langle x^k - x^*, \mathbb{E}\left[\widehat{g}^k - \overline{g}^k \mid x^k\right]\right\rangle \le 2\gamma G\|x^k - x^*\|\sqrt{C_2}K^{\frac{1}{\alpha}}\mathbb{1}_{k,v},$$

and

$$\begin{aligned}
\mathbb{E}\left[\|x^{k+1} - x^*\|^2 \mid x^k\right] &\le \|x^k - x^*\|^2 - 2\gamma\left(f(x^k) - f(x^*)\right) + 2\gamma^2G^2K^{\frac{2-\alpha}{\alpha}} \\
&\quad + 2\gamma G\sqrt{C_2}K^{\frac{1}{\alpha}}\|x^k - x^*\|\mathbb{1}_{k,v} + 2\gamma^2C_2G^2K^{\frac{2}{\alpha}}\mathbb{1}_{k,v}.
\end{aligned}$$

Taking the full expectation from the both sides of the above inequality and summing up the results for $k = 0, 1, \ldots, T-1$ we derive

$$
\begin{aligned}
\frac{2\gamma}{T} \sum_{k=0}^{T-1} \mathbb{E}[f(x^k) - f(x^*)] \quad &\leq \quad \frac{1}{T} \sum_{k=0}^{T-1} \left( \mathbb{E}\left[\|x^k - x^*\|^2\right] - \mathbb{E}\left[\|x^{k+1} - x^*\|^2\right] \right) + 2\gamma^2 G^2 K^{\frac{2-\alpha}{\alpha}} \\
&\quad + \frac{2\gamma G \sqrt{C_2} K^{\frac{1}{\alpha}}}{T} \sum_{k=0}^{T-1} \mathbb{E}\left[\|x^k - x^*\| \mathbb{1}_{k,v}\right] + \frac{2\gamma^2 C_2 G^2 K^{\frac{2}{\alpha}}}{T} \sum_{k=0}^{T-1} \mathbb{E}[\mathbb{1}_{k,v}] \\
&\leq \quad \frac{\|x^0 - x^*\|^2 - \mathbb{E}[\|x^K - x^*\|^2]}{K} + 2\gamma^2 G^2 K^{\frac{2-\alpha}{\alpha}} \\
&\quad + \frac{2\gamma G \sqrt{C_2} K^{\frac{1}{\alpha}}}{T} \sum_{k=0}^{T-1} \sqrt{\mathbb{E}\left[\|x^k - x^*\|^2\right] \mathbb{E}\left[\mathbb{1}_{k,v}\right]} \\
&\quad + \frac{2\gamma^2 C_2 G^2 K^{\frac{2}{\alpha}}}{T} \sum_{k=0}^{T-1} \mathbb{E}[\mathbb{1}_{k,v}].
\end{aligned}
$$

From Jensen's inequality we have $f(\overline{x}^T) \leq \frac{1}{T} \sum_{k=0}^{T-1} f(x^k)$, where $\overline{x}^T = \frac{1}{T} \sum_{k=0}^{T-1} x^k$. Using this and new notation $R_k = \|x^k - x^*\|$, $k > 0$, $R_0 \geq \|x^0 - x^*\|$ we get

$$
\begin{aligned}
0 \leq 2\gamma \mathbb{E}\left[f(\overline{x}^T) - f(x^*)\right] \quad &\leq \quad \frac{R_0^2 - \mathbb{E}[R_T^2]}{T} + 2\gamma^2 G^2 K^{\frac{2-\alpha}{\alpha}} \\
&\quad + \frac{2\gamma G \sqrt{C_2} K^{\frac{1}{\alpha}}}{T} \sum_{k=0}^{T-1} \sqrt{\mathbb{E}\left[R_k^2\right] \mathbb{E}\left[\mathbb{1}_{k,v}\right]} \\
&\quad + \frac{2\gamma^2 C_2 G^2 K^{\frac{2}{\alpha}}}{T} \sum_{k=0}^{T-1} \mathbb{E}[\mathbb{1}_{k,v}]
\end{aligned} \tag{67}
$$

implying (after changing the indices) that

$$
\begin{aligned}
\mathbb{E}[R_k^2] \quad &\leq \quad R_0^2 + 2\gamma^2 G^2 k K^{\frac{2-\alpha}{\alpha}} + 2\gamma G \sqrt{C_2} K^{\frac{1}{\alpha}} \sum_{l=0}^{k-1} \sqrt{\mathbb{E}\left[R_l^2\right] \mathbb{E}\left[\mathbb{1}_{l,v}\right]} \\
&\quad + 2\gamma^2 C_2 G^2 K^{\frac{2}{\alpha}} \sum_{l=0}^{k-1} \mathbb{E}[\mathbb{1}_{l,v}]
\end{aligned} \tag{68}
$$

holds for all $k \geq 0$. In the remaining part of the proof we derive by induction that

$$
\begin{aligned}
R_0^2 + 2\gamma^2 G^2 k K^{\frac{2-\alpha}{\alpha}} + 2\gamma G \sqrt{C_2} K^{\frac{1}{\alpha}} \sum_{l=0}^{k-1} \sqrt{\mathbb{E}\left[R_l^2\right] \mathbb{E}\left[\mathbb{1}_{l,v}\right]} \\
+ 2\gamma^2 C_2 G^2 K^{\frac{2}{\alpha}} \sum_{l=0}^{k-1} \mathbb{E}[\mathbb{1}_{l,v}] \quad \leq \quad 2R_0^2
\end{aligned} \tag{69}
$$

for all $k = 0, \ldots, K$. For $k = 0$ this inequality trivially holds. Next, assume that it holds for all $k = 0, 1, \ldots, T-1$, $T \leq K-1$. Let us show that it holds for $k = T$ as well. From (42) and (43) we have that $\mathbb{E}[R_k^2] \leq 2R_0^2$ for all $k = 0, 1, \ldots, T-1$. Therefore,

$$
\begin{aligned}
\mathbb{E}[R_T^2] \quad &\leq \quad R_0^2 + 2\gamma^2 G^2 T K^{\frac{2-\alpha}{\alpha}} + 2\gamma G \sqrt{C_2} K^{\frac{1}{\alpha}} \sum_{l=0}^{T-1} \sqrt{\mathbb{E}\left[R_l^2\right] \mathbb{E}\left[\mathbb{1}_{l,v}\right]} \\
&\quad + 2\gamma^2 C_2 G^2 K^{\frac{2}{\alpha}} \sum_{l=0}^{T-1} \mathbb{E}[\mathbb{1}_{l,v}] \\
&\leq \quad R_0^2 + 2\gamma^2 G^2 T K^{\frac{2-\alpha}{\alpha}} + 2\gamma G R_0 \sqrt{2C_2} K^{\frac{1}{\alpha}} \sum_{l=0}^{T-1} \sqrt{\mathbb{E}\left[\mathbb{1}_{l,v}\right]} \\
&\quad + 2\gamma^2 C_2 G^2 K^{\frac{2}{\alpha}} \sum_{l=0}^{T-1} \mathbb{E}[\mathbb{1}_{l,v}]
\end{aligned}
$$

If a Byzantine peer deviates from the protocol at iteration $k$, it will be detected with some probability $p_k$ during the next iteration. One can lower bound this probability as

$$p_k \geq m \cdot \frac{|\mathcal{G}_k|}{n_k} \cdot \frac{1}{n_k} = \frac{m(1 - \delta_k)}{n_k} \geq \frac{m}{n}.$$

That is, each individual Byzantine worker can violate the protocol no more than $1/p$ times on average. However, even one Byzantine peer can create a shift of the order $\Delta_{\max}^k$ at each part of the resulting vector. Therefore, all Byzantine peers can violate the protocol no more than $b/p$ times on average implying that

$$\mathbb{E}[R_T^2] \quad \leq \quad R_0^2 + 2\gamma^2 G^2 T K^{\frac{2-\alpha}{\alpha}} + \frac{2\gamma G R_0 \sqrt{2C_2} K^{\frac{1}{\alpha}} nb}{m} + \frac{2\gamma^2 C_2 G^2 K^{\frac{2}{\alpha}} nb}{m}.$$

Taking

$$\gamma = \min\left\{ \frac{R_0}{\sqrt{6} G K^{\frac{1}{\alpha}}}, \frac{m R_0}{12\sqrt{2C_2} G n b K^{\frac{1}{\alpha}}} \right\}$$

we ensure that

$$2\gamma^2 G^2 T K^{\frac{2-\alpha}{\alpha}} + \frac{2\gamma G R_0 \sqrt{2C_2} K^{\frac{1}{\alpha}} nb}{m} + \frac{2\gamma^2 C_2 G^2 K^{\frac{2}{\alpha}} nb}{m} \quad \leq \quad \frac{R_0^2}{3} + \frac{R_0^2}{3} + \frac{R_0^2}{3} = R_0^2$$

and, as a result, we get $\mathbb{E}[R_T^2] \leq 2R_0^2$. Therefore, (69) holds for all $k = 0, 1, \ldots, K$. Together with (67) it implies

$$\mathbb{E}\left[ f(\overline{x}^K) - f(x^*) \right] \leq \frac{R_0^2}{\gamma K}.$$

Next, from our stepsize rule (64) it follows that

$$\mathbb{E}\left[ f(\overline{x}^K) - f(x^*) \right] = \mathcal{O}\left( \frac{G R_0}{K^{\frac{1-\alpha}{\alpha}}} + \frac{nb G R_0}{m K^{\frac{1-\alpha}{\alpha}}} \right)$$

meaning that after

$$K = \mathcal{O}\left( \left(\frac{G R_0}{\varepsilon}\right)^{\frac{\alpha}{\alpha-1}} + \left(\frac{nb G R_0}{m\varepsilon}\right)^{\frac{\alpha}{\alpha-1}} \right)$$

iterations BTARD-CLIPPED-SGD guarantees $\mathbb{E}[f(\overline{x}^K) - f(x^*)] \leq \varepsilon$. □

That is, when the number of attacking Byzantines is unknown the complexity bound becomes $(nb/m)^{\alpha/(\alpha-1)}$ times worse in comparison to (60).

### E.4.3 STRONGLY CONVEX CASE: RESTARTED-BTARD-CLIPPED-SGD

In this section, we provide the complete statements and the full proofs of the convergence results for the restarted version of BTARD-CLIPPED-SGD (RESTARTED-BTARD-CLIPPED-SGD, Alg. 7) when the objective function $f$ is smooth and strongly convex.

---

**Algorithm 9** RESTARTED-BTARD-CLIPPED-SGD

**Input:** $x^0$ – starting point, $r$ – number of restarts, $\{\gamma_t\}_{t=1}^r$ – stepsizes for BTARD-CLIPPED-SGD, $\{K_t\}_{t=1}^r$ – number of iterations for BTARD-CLIPPED-SGD, $\{s_{i,k,t}\}_{i,k,t=0,0,1}^{n,K-1,r}$ – seeds for batches computations, $\{\lambda_{k,t}\}_{k,t=0,1}^{K_t,r}$ – gradient clipping parameters
1: $\widehat{x}^0 = x^0$
2: **for** $t = 1, 2, \ldots, r$ **do**
3:    Run BTARD-CLIPPED-SGD (Alg. 8) for $K_t$ iterations with stepsize $\gamma_t$, starting point $\widehat{x}^{t-1}$, gradient clipping parameters $\{\lambda_{k,t}\}_{k=0}^{K-1}$, and seeds for batches computations $\{s_{i,k,t}\}_{i,k=0,0}^{n,K-1}$. Define $\widehat{x}^t$ as $\widehat{x}^t = \frac{1}{K_t} \sum_{k=0}^{K_t} x^{k,t}$, where $x^{0,t}, x^{1,t}, \ldots, x^{K_t,t}$ are the iterates produced by BTARD-CLIPPED-SGD.
**Output:** $\widehat{x}^r$

---

We start with the case when the number of attacking Byzantine workers is known at each iteration.

**Theorem E.10.** *Let As. E.1 hold, $Q$ is bounded, $f$ be $\mu$-strongly convex (see Def. E.2), $x^*$ be some optimum of $f$, and $\nabla f(x^*) = 0$. Moreover, assume that $b \leq 0.15(n - m)$, $m \leq {}^{(n-2b)}/_2$, and the exact number of attacking Byzantine peers is known to all good peers at each iteration. Next, assume that*

$$\gamma = \min\left\{\frac{R_0}{\sqrt{6} \cdot 2^{\frac{t}{2}}GK_t^{\frac{1}{\alpha}}}, \frac{mR_0}{12 \cdot 2^{\frac{t}{2}}Gn\sqrt{10\delta(C_1K_t^{\frac{4-\alpha}{2\alpha}} + C_2K_t^{\frac{2}{\alpha}})}}\right\}, \quad \Delta_{\max}^{k,t} = 2\lambda_{k,t} = \frac{2\lambda_t}{\sqrt{n_k^t - m}},$$
$$(70)$$

$$K_t = \max\left\{\left(\frac{2\sqrt{6}G \cdot 2^{\frac{t}{2}}}{\mu R_0}\right)^{\frac{\alpha}{\alpha-1}}, \left(\frac{24Gn\sqrt{10\delta(C_1+C_2)}2^{\frac{t}{2}}}{m\mu R_0}\right)^{\frac{\alpha}{\alpha-1}}\right\}, \quad \lambda_t = GK_t^{\frac{1}{\alpha}}, \quad (71)$$

$$r = \left\lceil\log_2\frac{\mu R_0^2}{\varepsilon}\right\rceil - 1, \tag{72}$$

*where $R_0 \geq \|x^0 - x^*\|$ and $\Delta_{\max}^{k,t}$ is the parameter for verification 3 at iteration $k$ of BTARD-CLIPPED-SGD, $n_k^t$ is the total number of workers at iteration $k$ of $t$-th restart. Then, we have $\mathbb{E}[f(\widehat{x}^r) - f(x^*)] \leq \varepsilon$ after $r$ restarts of BTARD-CLIPPED-SGD and the total number of executed iterations of BTARD-CLIPPED-SGD is*

$$\sum_{t=1}^{r} K_t = \mathcal{O}\left(\left(\frac{G^2}{\mu\varepsilon}\right)^{\frac{\alpha}{2(\alpha-1)}} + \left(\frac{n\sqrt{\delta}}{m}\right)^{\frac{\alpha}{\alpha-1}}\left(\frac{G^2}{\mu\varepsilon}\right)^{\frac{\alpha}{2(\alpha-1)}}\right) \tag{73}$$

*Proof.* Theorem E.8 implies that BTARD-CLIPPED-SGD with

$$\gamma = \min\left\{\frac{R_0}{\sqrt{6}GK^{\frac{1}{\alpha}}}, \frac{mR_0}{12Gn\sqrt{10\delta(C_1K^{\frac{4-\alpha}{2\alpha}} + C_2K^{\frac{2}{\alpha}})}}\right\}$$

guarantees

$$\mathbb{E}\left[f(\overline{x}^K) - f(x^*)\right] \leq \frac{R_0^2}{\gamma K}$$

after $K$ iterations. Therefore, after the first restart we have

$$\mathbb{E}[f(\widehat{x}^1) - f(x^*)] \leq \frac{R_0^2}{\gamma_1 K_1} \leq \frac{\mu R_0^2}{4}.$$

From $\mu$-strong convexity of $f$ and $\nabla f(x^*) = 0$ we have

$$\frac{\mu}{2}\|\widehat{x}^1 - x^*\|^2 \leq f(\widehat{x}^1) - f(x^*) \Longrightarrow \mathbb{E}[\|\widehat{x}^1 - x^*\|^2] \leq \frac{R_0^2}{2}.$$

Next, assume that we have $\mathbb{E}[f(\widehat{x}^t) - f(x^*)] \leq \frac{\mu R_0^2}{2^{t+1}}$, $\mathbb{E}[\|\widehat{x}^t - x^*\|^2] \leq \frac{R_0^2}{2^t}$ for some $t \leq r - 1$. Then, Theorem E.8 implies that

$$\mathbb{E}[f(\widehat{x}^{t+1}) - f(x^*) \mid x^t] \leq \frac{\|\widehat{x}^t - x^*\|^2}{\gamma_t K_t}.$$

Taking the full expectation from the both sides of previous inequality we get

$$\mathbb{E}[f(\widehat{x}^{t+1}) - f(x^*)] \leq \frac{\mathbb{E}[\|\widehat{x}^t - x^*\|^2]}{\gamma_t K_t} \leq \frac{R_0^2}{2^t \gamma_t K_t} \leq \frac{\mu R_0^2}{2^{t+2}}.$$

From $\mu$-strong convexity of $f$ and $\nabla f(x^*) = 0$ we have

$$\frac{\mu}{2}\|\widehat{x}^{t+1} - x^*\|^2 \leq f(\widehat{x}^{t+1}) - f(x^*) \Longrightarrow \mathbb{E}[\|\widehat{x}^{t+1} - x^*\|^2] \leq \frac{R_0^2}{2^{t+1}}.$$

Therefore, by mathematical induction we have that for all $t = 1, \ldots, r$

$$\mathbb{E}[f(\widehat{x}^t) - f(x^*)] \leq \frac{\mu R_0^2}{2^{t+1}}, \quad \mathbb{E}\left[\|\widehat{x}^t - x^*\|^2\right] \leq \frac{R_0^2}{2^t}.$$

Then, after $r = \left\lceil \log_2 \frac{\mu R_0^2}{\varepsilon} \right\rceil - 1$ restarts of BTARD-CLIPPED-SGD we have $\mathbb{E}[f(\widehat{x}^r) - f(x^*)] \leq \varepsilon$. The total number of iterations executed by BTARD-CLIPPED-SGD is

$$
\begin{aligned}
\sum_{t=1}^{r} K_t &= \mathcal{O}\left( \sum_{t=1}^{r} \max\left\{ \left( \frac{G \cdot 2^{\frac{t}{2}}}{\mu R_0} \right)^{\frac{\alpha}{\alpha-1}}, \left( \frac{Gn\sqrt{\delta}2^{\frac{t}{2}}}{m\mu R_0} \right)^{\frac{\alpha}{\alpha-1}} \right\} \right) \\
&= \mathcal{O}\left( \max\left\{ \left( \frac{G}{\mu R_0} \right)^{\frac{\alpha}{\alpha-1}} \cdot 2^{\frac{r\alpha}{2(\alpha-1)}}, \left( \frac{Gn\sqrt{\delta}}{m\mu R_0} \right)^{\frac{\alpha}{\alpha-1}} \cdot 2^{\frac{r\alpha}{2(\alpha-1)}} \right\} \right) \\
&= \mathcal{O}\left( \max\left\{ \left( \frac{G}{\mu R_0} \right)^{\frac{\alpha}{\alpha-1}} \cdot \left( \frac{\mu R_0^2}{\varepsilon} \right)^{\frac{\alpha}{2(\alpha-1)}}, \left( \frac{Gn\sqrt{\delta}}{m\mu R_0} \right)^{\frac{\alpha}{\alpha-1}} \cdot \left( \frac{\mu R_0^2}{\varepsilon} \right)^{\frac{\alpha}{2(\alpha-1)}} \right\} \right) \\
&= \mathcal{O}\left( \left( \frac{G^2}{\mu\varepsilon} \right)^{\frac{\alpha}{2(\alpha-1)}} + \left( \frac{n\sqrt{\delta}}{m} \right)^{\frac{\alpha}{\alpha-1}} \left( \frac{G^2}{\mu\varepsilon} \right)^{\frac{\alpha}{2(\alpha-1)}} \right).
\end{aligned}
$$

$\square$

In the strongly convex case, similar observations hold as in the convex case. Next, we derive the result without assuming that $\widehat{b}^k$ is known to all peers at each iteration.

**Theorem E.11.** *Let As. E.1 hold, $Q$ is bounded, $f$ be $\mu$-strongly convex (see Def. E.2), $x^*$ be some optimum of $f$, and $\nabla f(x^*) = 0$. Moreover, assume that $b \leq 0.15(n-m)$, $m \leq {(n-2b)}/{2}$, and $\delta = 0$ is used to compute clipping parameter $\tau_l$ for CenteredClip. Next, assume that*

$$
\gamma = \min\left\{ \frac{R_0}{\sqrt{6} \cdot 2^{\frac{t}{2}} G K_t^{\frac{1}{\alpha}}}, \frac{mR_0}{12 \cdot 2^{\frac{t}{2}} Gnb\sqrt{2C_2} K_t^{\frac{1}{\alpha}}} \right\}, \quad \Delta_{\max}^{k,t} = 2\lambda_{k,t} = \frac{2\lambda_t}{\sqrt{n_k^t - m}}, \tag{74}
$$

$$
K_t = \max\left\{ \left( \frac{2\sqrt{6}G \cdot 2^{\frac{t}{2}}}{\mu R_0} \right)^{\frac{\alpha}{\alpha-1}}, \left( \frac{24Gnb\sqrt{2C_2}2^{\frac{t}{2}}}{m\mu R_0} \right)^{\frac{\alpha}{\alpha-1}} \right\}, \quad \lambda_t = GK_t^{\frac{1}{\alpha}}, \tag{75}
$$

$$
r = \left\lceil \log_2 \frac{\mu R_0^2}{\varepsilon} \right\rceil - 1, \tag{76}
$$

*where $R_0 \geq \|x^0 - x^*\|$ and $\Delta_{\max}^{k,t}$ is the parameter for verification 3 at iteration $k$ of BTARD-CLIPPED-SGD, $n_k^t$ is the total number of workers at iteration $k$ of $t$-th restart. Then, we have $\mathbb{E}[f(\widehat{x}^r) - f(x^*)] \leq \varepsilon$ after $r$ restarts of BTARD-CLIPPED-SGD and the total number of executed iterations of BTARD-CLIPPED-SGD is*

$$
\sum_{t=1}^{r} K_t = \mathcal{O}\left( \left( \frac{G^2}{\mu\varepsilon} \right)^{\frac{\alpha}{2(\alpha-1)}} + \left( \frac{nb}{m} \right)^{\frac{\alpha}{\alpha-1}} \left( \frac{G^2}{\mu\varepsilon} \right)^{\frac{\alpha}{2(\alpha-1)}} \right) \tag{77}
$$

*Proof.* Theorem E.9 implies that BTARD-CLIPPED-SGD with

$$
\gamma = \min\left\{ \frac{R_0}{\sqrt{6}GK^{\frac{1}{\alpha}}}, \frac{mR_0}{12\sqrt{2C_2}GnbK^{\frac{1}{\alpha}}} \right\}
$$

guarantees

$$
\mathbb{E}\left[ f(\overline{x}^K) - f(x^*) \right] \leq \frac{R_0^2}{\gamma K}
$$

after $K$ iterations. Therefore, after the first restart we have

$$
\mathbb{E}[f(\widehat{x}^1) - f(x^*)] \leq \frac{R_0^2}{\gamma_1 K_1} \leq \frac{\mu R_0^2}{4}.
$$

From $\mu$-strong convexity of $f$ and $\nabla f(x^*) = 0$ we have

$$
\frac{\mu}{2}\|\widehat{x}^1 - x^*\|^2 \leq f(\widehat{x}^1) - f(x^*) \implies \mathbb{E}[\|\widehat{x}^1 - x^*\|^2] \leq \frac{R_0^2}{2}.
$$

Next, assume that we have $\mathbb{E}[f(\widehat{x}^t) - f(x^*)] \leq \frac{\mu R_0^2}{2^{t+1}}$, $\mathbb{E}[\|\widehat{x}^t - x^*\|^2] \leq \frac{R_0^2}{2^t}$ for some $t \leq r - 1$. Then, Theorem E.9 implies that

$$\mathbb{E}[f(\widehat{x}^{t+1}) - f(x^*) \mid x^t] \leq \frac{\|\widehat{x}^t - x^*\|^2}{\gamma_t K_t}.$$

Taking the full expectation from the both sides of previous inequality we get

$$\mathbb{E}[f(\widehat{x}^{t+1}) - f(x^*)] \leq \frac{\mathbb{E}[\|\widehat{x}^t - x^*\|^2]}{\gamma_t K_t} \leq \frac{R_0^2}{2^t \gamma_t K_t} \leq \frac{\mu R_0^2}{2^{t+2}}.$$

From $\mu$-strong convexity of $f$ and $\nabla f(x^*) = 0$ we have

$$\frac{\mu}{2}\|\widehat{x}^{t+1} - x^*\|^2 \leq f(\widehat{x}^{t+1}) - f(x^*) \implies \mathbb{E}[\|\widehat{x}^{t+1} - x^*\|^2] \leq \frac{R_0^2}{2^{t+1}}.$$

Therefore, by mathematical induction we have that for all $t = 1, \ldots, r$

$$\mathbb{E}[f(\widehat{x}^t) - f(x^*)] \leq \frac{\mu R_0^2}{2^{t+1}}, \quad \mathbb{E}\left[\|\widehat{x}^t - x^*\|^2\right] \leq \frac{R_0^2}{2^t}.$$

Then, after $r = \left\lceil \log_2 \frac{\mu R_0^2}{\varepsilon} \right\rceil - 1$ restarts of BTARD-CLIPPED-SGD we have $\mathbb{E}[f(\widehat{x}^r) - f(x^*)] \leq \varepsilon$. The total number of iterations executed by BTARD-CLIPPED-SGD is

$$
\begin{aligned}
\sum_{t=1}^{r} K_t &= \mathcal{O}\left(\sum_{t=1}^{r} \max\left\{\left(\frac{G \cdot 2^{\frac{t}{2}}}{\mu R_0}\right)^{\frac{\alpha}{\alpha-1}}, \left(\frac{Gn\sqrt{\delta}2^{\frac{t}{2}}}{m\mu R_0}\right)^{\frac{\alpha}{\alpha-1}}\right\}\right) \\
&= \mathcal{O}\left(\max\left\{\left(\frac{G}{\mu R_0}\right)^{\frac{\alpha}{\alpha-1}} \cdot 2^{\frac{r\alpha}{2(\alpha-1)}}, \left(\frac{Gnb}{m\mu R_0}\right)^{\frac{\alpha}{\alpha-1}} \cdot 2^{\frac{r\alpha}{2(\alpha-1)}}\right\}\right) \\
&= \mathcal{O}\left(\max\left\{\left(\frac{G}{\mu R_0}\right)^{\frac{\alpha}{\alpha-1}} \cdot \left(\frac{\mu R_0^2}{\varepsilon}\right)^{\frac{\alpha}{2(\alpha-1)}}, \left(\frac{Gnb}{m\mu R_0}\right)^{\frac{\alpha}{\alpha-1}} \cdot \left(\frac{\mu R_0^2}{\varepsilon}\right)^{\frac{\alpha}{2(\alpha-1)}}\right\}\right) \\
&= \mathcal{O}\left(\left(\frac{G^2}{\mu\varepsilon}\right)^{\frac{\alpha}{2(\alpha-1)}} + \left(\frac{nb}{m}\right)^{\frac{\alpha}{\alpha-1}}\left(\frac{G^2}{\mu\varepsilon}\right)^{\frac{\alpha}{2(\alpha-1)}}\right).
\end{aligned}
$$

$\square$

## F    REPUTATION SYSTEM FOR PUBLIC COLLABORATIONS

In this section, we address Byzantine-tolerant training in a setup where new participants can join or leave collaboration midway through training. This requirement arises naturally if a given training run relies on volunteers or an open pool of paid participants (Kijsipongse et al., 2018; Ryabinin & Gusev, 2020; Atre et al., 2021; Diskin et al., 2021). In addition to all existing concerns from Section 3, this new setup allows Byzantine attackers to assume new identity each time they are blocked. Further yet, Byzantine participants can simultaneously use multiple identities in order to obtain majority in the voting procedure, which is known as Sybil attacks (Douceur, 2002; Trifa & Khemakhem, 2014; Wang & Kangasharju, 2012).

In this analysis[9], we consider a training run where Byzantine peers collectively possess $\delta < \delta_{max}$ of all compute resources (we explore the role of $\delta_{max} < 1/2$ later in this section). Intuitively, one can think of this setting as distributed training with $n$ identical computers, $\lfloor \delta \cdot n \rceil$ of which are controlled by Byzantines. The "Byzantine GPUs" can be allocated between an arbitrary number of identities. For instance, one accelerator can run full BTARD-SGD protocol for one peer or drop some of the computation and use the freed "compute cycles" to run computation for another participant. Theoretically, a device can run computation for an arbitrarily large number of peers, as long as it actually computes as many gradients as one benign participant does in the same time-frame.

To protect against this new attack type, we augment BTARD-SGD with a reputation system designed to limit the impact of pseudonymous identities with the actual underlying compute. We base this system on the following three assumptions:

1. **Unique and optimal computations:** the gradients computed by peer $i$ at step $k$ cannot be circumvented or reused from other peers and/or previous steps.
2. **Public key infrastructure:** peers have unique public/private key pairs and know each other's public keys.
3. **Cryptographic hash:** peers have access to a hash function such that finding a vector $x$ satisfying $\text{hash}(x) = y$ is infeasible for $\lfloor \delta \cdot n \rceil$ compute over the entire training duration.

We associate each participant with a public record that is used to verify that peer's legitimacy. These records can be securely stored in a Distributed Hash Table (see Appendix G). When a new peer joins the network, it begins with an empty record and is therefore "untrusted". Untrusted peers compute gradients normally, but cannot aggregate vectors from others and cannot serve as validators. More importantly, other peers exclude untrusted gradients from aggregation, using them only for the purpose of validating those peers.

Each time a peer computes gradients $g_i^k$ over publicly known batch $\xi_i^k$, it must write $\text{hash}(g_i^k)$ to its own public record and sign it with its private key. As in the original BTARD-SGD, some of those entries will be validated by other peers chosen by MPRNG. In turn, the chosen validators will either approve their entry or invoke ACCUSE to ban the peer.

In order to become trusted, a given peer must report consecutive gradients until it accumulates $T$ entries approved by (provably) random peers. Here, $T$ is a hyperparameter that should be large enough for the training to recover from any previous attacks and make some progress before previously banned malicious peers can earn trust again. In practice, $T$ may be chosen experimentally by observing the number of iterations it takes to improve the loss upon its pre-attack value in case of the most effective attacks, as reported in Section 4.

While $T$ may be application-dependent, we note that its minimal value is small in terms of the relative training time in all our experiments. $T$ corresponding to the 10% of total training time is more than 3 times larger than the worst "recovery time" for both setups considered in Section 4, where almost a half of the peers are Byzantine. Moreover, Appendix I.1 suggests that recovery from the worst-case attack may happen even faster in case of a smaller share of Byzantines. In that setup (with $\approx 20\%$ of peers being Byzantine), $T$ corresponding to the 1% of training time is already enough.

Once a peer becomes trusted, it must continue reporting gradient hashes to maintain trust. Even a single missing or invalidated hash breaks the chain and results in the corresponding peer being

---

[9]Note that we only provide rigorous convergence guarantees for the case of the Byzantine attacks. However, a heuristic described in this section helps with resisting the Sybil attacks in practice.

banned. To maintain this invariant, peers chosen as a validators add the recalculated hashes into their own record instead of the skipped iteration.

To protect against dilution attacks, a cooperative training run can simultaneously consider at most as many "untrusted" peers as there are trusted ones: all subsequent peers wait in a queue until one of the untrusted peers becomes either trusted or banned.

**Analysis.** Under this formalism, a Sybil attacker will attempt to maximize the number of trusted identities it can control with a limited amount of compute. In the simplest case, an attacker has exactly one GPU that can be used to either run all computations for identity or partial computation for multiple identities.

In the latter case, an attacker can honestly compute gradients for identity A with probability $p \in [0, 1]$ and for identity B with probability $1 - p$. To breaking the chain, the identity that does **not** compute gradients at a given step can report arbitrary (e.g. random) entries instead of hash($g_i^k$).

Consider the expected number of "trusted" identities after enough steps for T validations by *honest* validators (on average, $T \cdot \frac{n}{k \cdot (1-\delta)}$ steps). Identity A becomes trusted with probability $p^T$, otherwise it is banned. Similarly, identitiy B survives with probability $(1 - p)^T$. Thus, the expected number of trusted identities after T steps is $p^T + (1 - p)^T$.

For $T > 1$, this expectation is maximal iff $p \in \{0, 1\}$. Thus, if a peer needs more than one validation to become trusted, the "optimal strategy" for a Sybil attacker is to fully support one identity instead of spreading the resources between multiple ones. This observation can be generalized for distributing $\lfloor \delta \cdot n \rfloor$ over an $m \geq \lfloor \delta \cdot n \rfloor$ pseudonymous identities, where maximizing the expected number of trusted identities requires fully supporting any $\lfloor \delta \cdot n \rfloor$ identities and disregarding the rest (for $T > 1$, as before).

**Overhead computation.** When training without Byzantine participants, this modified version of BTARD-SGD requires, on average, $T \cdot \frac{n}{k}$ additional gradient computations per participant at the very beginning. However, once all peers become trusted, the algorithm computes exactly the same number of gradients as regular BTARD-SGD, effectively training at $\frac{n-k}{n}$ efficiency of AR-SGD, plus the same communication overhead.

**Remark 1: Temporary majority.** Despite the fact that spreading 1 "compute unit" across multiple identities reduces the *expected* number of trusted identities, it may still be useful to establish a temporary majority, albeit with a small probability. For instance, splitting one compute unit evenly among $m$ identities (each with $p=1/m$) may result in both $m$ identities temporarily gaining trust with probability:

$$P(\text{peer}_1 \wedge \cdots \wedge \text{peer}_m) = \prod_{i=1}^{m} \frac{1}{m^T} = m^{-Tm} \tag{78}$$

A Sybil attacker can simply repeat this procedure on every step until it can establish a temporary majority and use this majority to harm training (e.g. ban non-malicious peers). A natural way to remedy this is to increase $T$ to such an extent that (78) becomes negligibly small.

**Remark 2: Extra compute for Byzantine nodes.** Unlike benign peers, Byzantine attackers do not need to honestly validate each other. When a Byzantine peer is chosen as validator, it can approve its target without actually computing the gradients. In turn, the freed compute resources can be used to support additional Byzantine identities.

Thus, if a given training run has $n$ trusted peers and chooses $k$ validators on each step, Sybil attackers can control slightly more than $\lfloor \delta \cdot n \rfloor$ of all identities by using the free compute cycles from validation to support additional peers. Thus, the proposed reputation system requires that the total computational power $B_{max}$ available to Byzantines is less than $\frac{1}{2}$ by a (typically small) margin that depends on $n$, $k$, and $T$.

**Remark 3: Perpetual attacks.** When training in open collaborations, one cannot ban the Byzantine peers entirely: a Byzantine attacker will always be able to assume a new identity at the cost of running

honestly for $T \cdot \frac{n}{k \cdot (1-\delta)}$ gradient steps. Thus, unlike in Appendix E, we cannot make BTARD-SGD unbiased by increasing $\tau$. However, as we demonstrated in Section 4, the biased variant of BTARD-SGD with constant $\tau$ can still train real-world deep learning models with the same or virtually the same learning curves as regular SGD.

## G  SECURE DISTRIBUTED HASH TABLES

Distributed Hash Tables (DHT) are protocols that establish a decentralized key-value storage over decentralized unreliable participants (Maymounkov & Mazieres, 2002; Balakrishnan et al., 2003; Zhao et al., 2003; Rowstron & Druschel, 2001). To determine which DHT peers are responsible for a given key-value pair, each participant samples a unique binary identifier (ID) sampled uniformly from the space of hash function outputs. When "storing a $(key,\ value)$" on the DHT, one finds $k$ peers whose IDs are nearest to $\mathrm{hash}(key)$ and sends the data to each one of those peers. In turn, a peer that wants to read the value or a given key will also search for neighbors whose IDs are close to $\mathrm{hash}(key)$ and request the data from those peers. Thus, the data can be accessed as long as at least one o $k$ chosen peers remains active, with some DHT variants introducing additional replication protocols.

Our specific implementation is based on Kademlia (Maymounkov & Mazieres, 2002), a popular DHT variant that determines nearest neighbors based on XOR distance function or their IDs: $d(x, y) = \mathrm{int}(x \oplus y)$. More importantly, Kademlia protocol organizes nodes in such a way that each individual peer only "knows" a small subset of $O(\log_2 n)$ direct neighbors, however, it is possible to navigate the neighborhood graph to find the globally nearest neighbors in $O(\log_2 N)$ network requests.

DHT protocols were originally designed for large-scale distributed systems such as BitTorrent, IPFS and several cryptocurrencies. To maintain integrity in these applications, modern DHT protocols also employ security measures that make them resistant to Byzantine and Sybil attacks (Urdaneta et al., 2011).

In our specific scenario, the most sensitive DHT entries are personal records that determine whether or not a given peer is trusted. We protect thee records by enforcing that every value stored in the DHT must be signed by their author's digital signature (Rivest et al., 1978). Thus, if a malicious peer attempts to modify a record it was not supposed to, all other peers will be able to detect that and eliminate such peers from the collective.

However, digital signature are known to be vulnerable to replay attacks: every time a non-Byzantine peer stores an given key-value pair signed with its private key, a Byzantine eavesdropper can record the signed entry and replay it in future. For ordinary DHTs, this would allow an attacker to revert any key-value pair to its previous state by replaying such pre-recorded messages.

Our algorithm protects against replay attacks by associating each key-value pair with a third value denoted as **expiration time**. Given two entries for the same key, DHT nodes will now prioritize the ones with the latest expiration time and consider it valid up to that time. Furthermore, in order to store a new entry to the DHT, a peer must now sign the entire key-value-expiration tuple. Thus, if a Byzantine peer replays a pre-recorded message, it will not be able to overwrite newer DHT entries that were signed for a more recent expiration time.

## H  ALBERT EXPERIMENT SETUP

In Section 4.2, we pretrain ALBERT (Lan et al., 2019) — a self-supervised Transformer model for learning representations of language data. We deliberately choose ALBERT instead of other models like BERT (Devlin et al., 2019) due to its high communication efficiency, which is caused by layerwise weight sharing and embedding layer factorization. In particular, we focus on a communication-efficient model, because the connection speed between the workers can become a noticeable constraint when averaging gradients of models with hundreds of millions of parameters. We train ALBERT-large on sequences of 512 tokens from the WikiText-103 (Merity et al., 2017) dataset. The training procedure starts from a random initialization, but the subword vocabulary (Sennrich et al., 2016) is the same as created by the authors of the original ALBERT models.

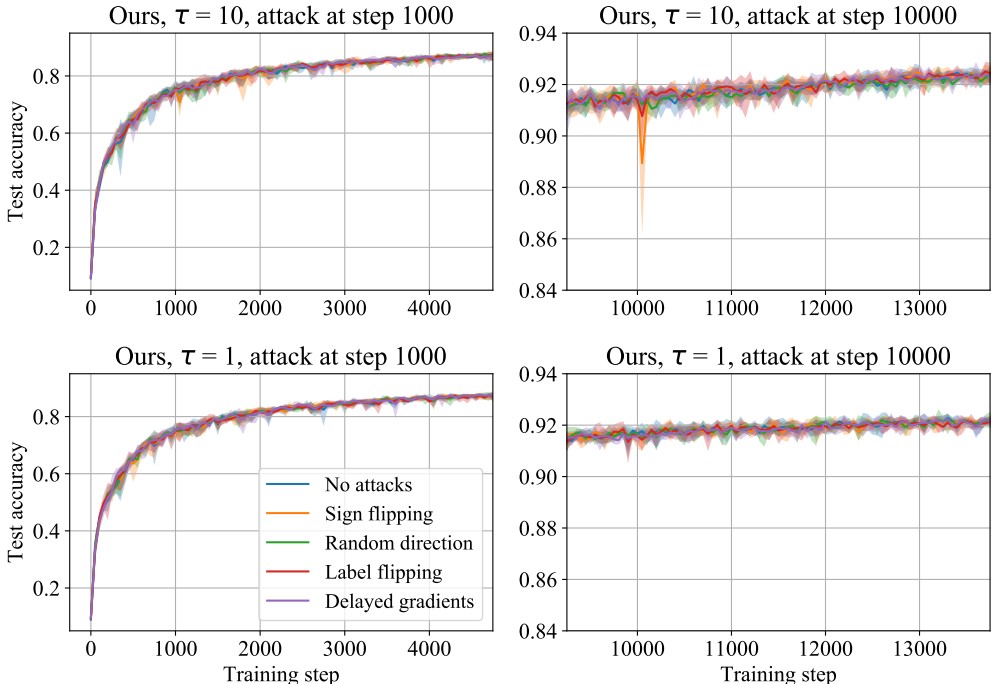

Figure 5: Effectiveness of attacks against BTARD-SGD for the case when 3 of 16 participants are Byzantine.

This model is trained with two objectives: masked language modeling (given a sentence with several masked tokens, predict the tokens that were masked) and sentence order prediction (given two segments from the same document, determine if they were swapped). We use LAMB optimizer (You et al., 2020) with batches that contain 4,096 examples, training with a peak learning rate equal to 0,00176 and a warmup of 5,000 gradient descent steps. In addition, we use gradient clipping with a maximum norm of 1 and weight decay regularization with the weight of 0,01. We run distributed training on 16 cloud instances, each equipped with a single Tesla T4 GPU. Each training run takes 2–3 days, depending on the instance availability.

# I  ADDITIONAL EXPERIMENTS

## I.1  EXTRA EVALUATIONS ON THE CIFAR10 CLASSIFICATION TASK

In this section, we perform several additional experiments with BTARD-SGD used to train the ResNet-18 model to solve the CIFAR10 classification task.

To better explore the space of possible attack vectors, we also evaluate two alternative settings. First, we consider a situation where Byzantine peers are less numerous. For this experiment, we use the same configuration as in Section 4.1, but with only 3 Byzantine peers out of 16 (just under 20%). Figure 5 demonstrates similar behavior to our original setup, but with significantly weaker in magnitude across all attacks.

Next, we explore a situation where Byzantine peers send incorrect gradients periodically, e.g. once per $T$ iterations. This reduces the attack intensity, but allows them to stay undetected for longer. In this setting, we consider 7 Byzantine peers and reuse all parameters from the original setup, except for the new attack period. We consider $T = 10$ for both scenarios (early and late attacks). The attacks are performed at steps $s + k \cdot T, k \in \mathbb{N}$ until the attacker is eventually banned. As expected, this setup increases the duration of each attack by a factor of $T$, but decreases the peak attack influence (see Figure 6).

Finally, we evaluate the convergence and the final test accuracy of the less computationally intensive variants of BTARD-SGD that limit the maximal number of iterations in the CenteredClip procedure

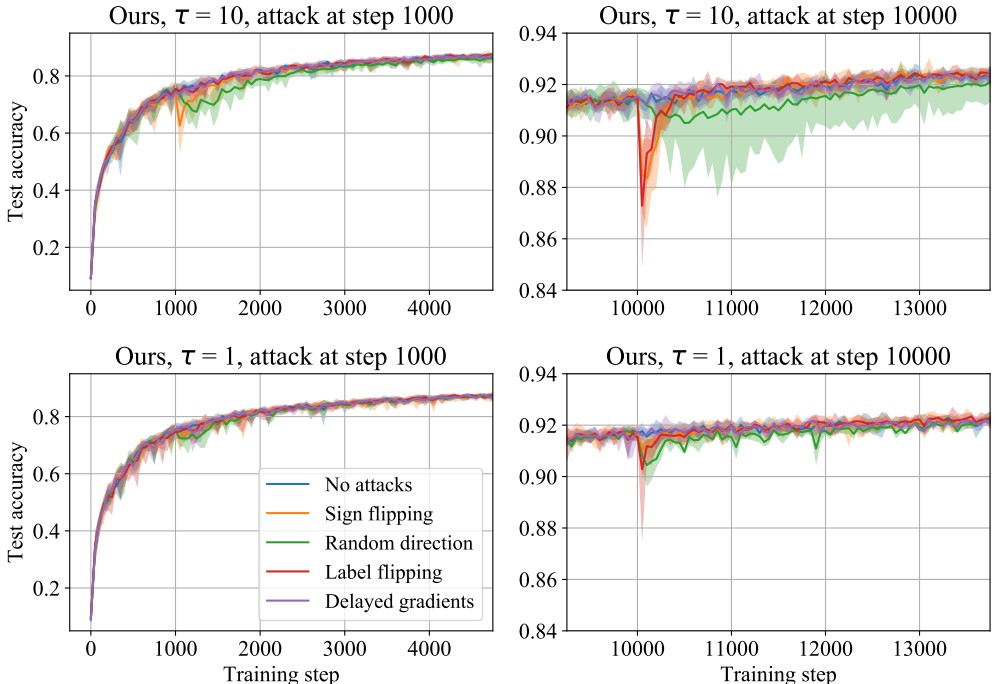

Figure 6: Effectiveness of attacks against BTARD-SGD for the case when Byzantines send incorrect gradients once per $T = 10$ steps.

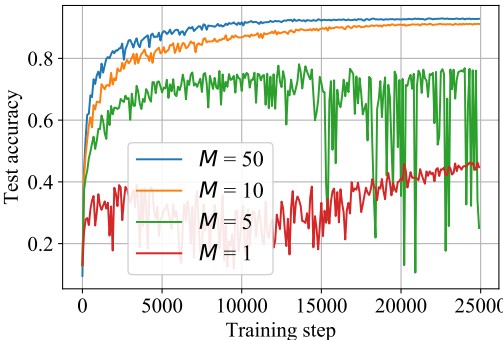

Figure 7: Convergence of BTARD-SGD with $\tau = 1$ depending on the maximal number of iterations $M$ in the CenteredClip procedure.

to $M$, where $M$ varies from 1 to 50. In the setup with $\tau = 1$, we observe that $M = 50$ iterations are always enough for CenteredClip to converge with $\epsilon = 10^{-6}$ in absence of the attacks. Figure 7 demonstrates that stopping the procedure earlier has negative effect on the final test accuracy. The effect becomes more significant for the smaller values of $M$.

### I.2 EVALUATING COMPUTATION OVERHEAD IN TERMS OF WALL TIME

For this analysis, we consider the ALBERT-large training setup from Section 4.2. Our training "swarm" contains 16 peers with T4 GPUs and 1 GiB/s network bandwidth. On average over 1000 training steps, the full training step for this model takes up 28.56 seconds. Of this, approximately 23.96 seconds were used up for communication and the remaining 4.60 seconds were spent for gradient aggregation CENTEREDCLIP.

Since MPRNG is running in the background, the only part of BTARD that affects the training time is Algorithm 2 (BUTTERFLYCLIP). Thus, we measure the time complexity of this algorithm with different numbers of internal iterations. During "normal" epochs where all Byzantines remained passive, the algorithm converged in 2–3 iterations for $\tau = 0.25$ and 5–10 iterations with $\tau = 0.125$.

Table 3: Computation overhead of BTARD in terms of wall time.

| No. of iterations | Wall time (CPU), sec | Wall time (GPU), sec |
|---|---|---|
| 3 | 0.362 ± 0.003 | 0.040 ± 0.002 |
| 5 | 0.430 ± 0.002 | 0.042 ± 0.002 |
| 10 | 0.601 ± 0.003 | 0.056 ± 0.005 |
| 20 | 0.943 ± 0.002 | 0.085 ± 0.009 |

We also noticed that this value has temporarily increased by 2–3 times while Byzantine peers were performing their attack.

In Table 3, we report the average wall time of our algorithm with a different number of iterations in two hardware setups: running on a 8-core VM with 3.1Ghz Intel Xeon 6148 CPU and on a single 1080 Ti GPU. We report the average wall time and the standard deviation over 10 runs.

Thus, even the worst case overhead ($\tau = 0.125$, CPU) is less than the 3% of the total step time without attacks and less than the 4% when the attack is active. One important consideration here is that the overhead is constant with respect to the number of peers due to the scaling properties of All-Reduce. Thus, if we train with hundreds of peers, the 0.3–0.6 second overhead can eventually become significant. However, it can be easily offset by moving the CENTEREDCLIP execution to GPU, which at this stage is waiting for the CENTEREDCLIP results anyway.

### I.3   EXPERIMENTS AT A LARGER SCALE

In this section, we evaluate the most effective attacks against BTARD-SGD in case of a larger number of peers to ensure that our algorithm scales well.

We still consider the ALBERT-large training setup from Section 4.2 and increase the number of peers to 64 (the largest hardware setup available to us), setting up 31 of them to be Byzantine. To balance for the increased number of peers, we divide the individual batch size of each peer by 4 and set the number of validators to be 4 as well.

Due to the large computation costs, we only evaluate the two most effective strategies for Byzantines based on Figure 3, making only one training run for each of them. We choose the random direction attack starting at the step 1000 and the sign flipping attack starting at the step 5000.

The results are shown in the Figure 8. Similarly to our previous experiments, the Byzantine peers managed to temporarily offset the training loss. As in the case with 16 peers, the sign flipping attack at the step 5000 obtains the "peak" distortion approximately 20 steps into the attack, and the random direction attack at the step 1000 has longer but less intensive effect. However, BTARD-SGD is able to quickly detect and ban the attackers, banning all 31 Byzantines in 100–150 steps and catching up with the original learning curve after approximately 150 steps (it is fair to take the original curves from Figure 3 since the aggregation results in the vanilla All-Reduce do not depend on the number of workers). We conclude that BTARD-SGD maintains its efficiency even at this scale.

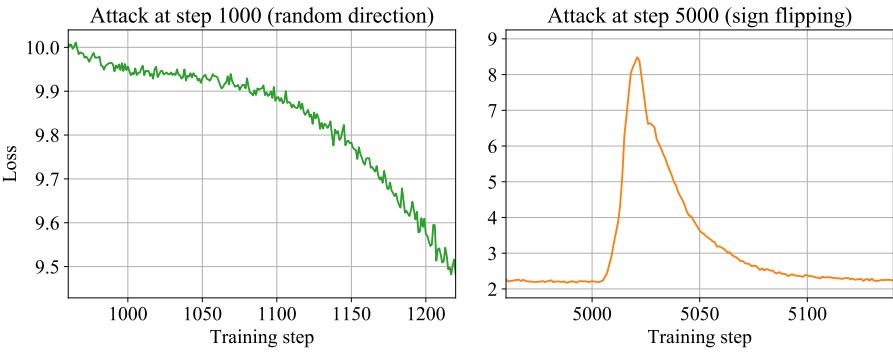

Figure 8: Effectiveness of attacks against BTARD-SGD for the case when 31 of 64 participants are Byzantine.

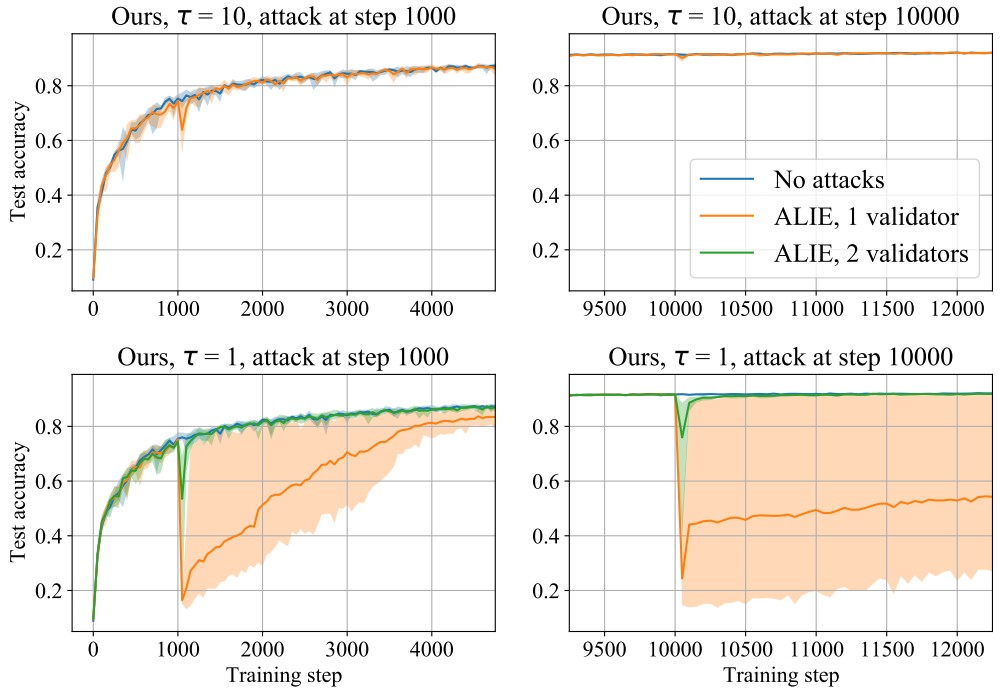

Figure 9: Effectiveness of the "A LITTLE IS ENOUGH" (ALIE) attack (Baruch et al., 2019) against BTARD-SGD.

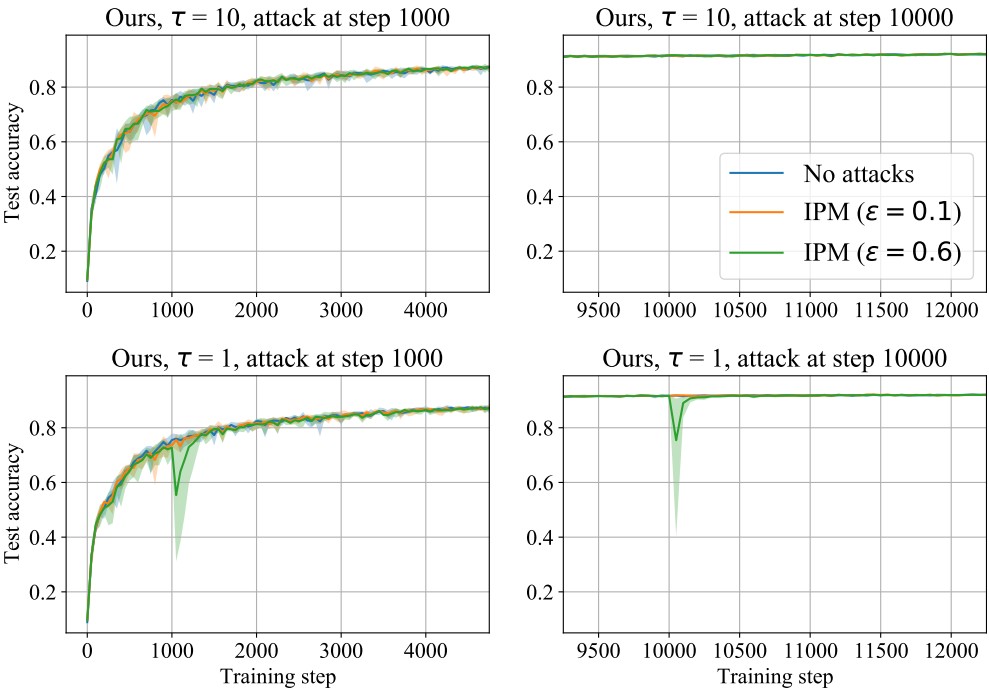

Figure 10: Effectiveness of the INNER PRODUCT MANIPULATION (IPM) attack (Xie et al., 2020; Allen-Zhu et al., 2021) against BTARD-SGD.

## I.4 EXPERIMENTS WITH LOW-MAGNITUDE ATTACKS

In this section, we evaluate several low-magnitude attacks against BTARD-SGD used for training the ResNet-18 model to solve the CIFAR10 classification task. We use the configuration from Section 4.1 and consider the following attacks:

- "A LITTLE IS ENOUGH" (ALIE): attackers collude to move the coordinate-wise median while still sending values inside the population variance. Baruch et al. (2019) show that this attack is effective against TrimmedMean (Yin et al., 2018) and Krum (Blanchard et al., 2017).

- INNER PRODUCT MANIPULATION (IPM): attackers send the average of all honest peers' gradients multiplied by $-\epsilon$. We test $\epsilon = 0.1$ (Xie et al. (2020) demonstrate its efficiency against the coordinate-wise median and Krum) and $\epsilon = 0.6$ (Allen-Zhu et al. (2021) state that it is the most efficient attack against their method among the reported ones).

As in Section 4.1, we report the mean and range of the test accuracy over 5 runs for $\tau \in \{1, 10\}$.

Regarding the ALIE attack (see Figure 9), we note that BTARD-SGD needs either two validators, or one validator and the weaker clipping ($\tau = 10$) to guarantee the fast recovery in the worst case (with 7 out of 16 peers being Byzantine).

This observation coincides with Baruch et al. (2019) demonstrating that the ALIE attack is more harmful against median-based and clipping approaches than to the usual mean aggregation without any defenses. Indeed, since $\tau$ in CENTEREDCLIP allows to smoothly interpolate between the geometric median ($\tau \to 0$) and the mean ($\tau \to \inf$) aggregation rules (as explained in Appendix D.1), it is natural to expect that the setups where $\tau$ is too small are more sensitive to the ALIE attack.

In turn, the IPM attack (see Figure 10) has a short negative effect for $\tau = 1$ (it increases for a larger $\epsilon$) and an almost unnoticeable effect for $\tau = 10$. One validator is enough to combat all variants of this attack.

We conclude that BTARD-SGD remains practical in case of the considered low-magnitude attacks. It is able to defend the training from the worst-case attacks (with almost half of the peers being Byzantine) while requiring no more than $\approx 13\%$ of the compute to be dedicated for validation.

