# OpenReview forum: "Secure Distributed Training at Scale"
_ICLR.cc/2022/Conference — ICLR 2022 Submitted_

### Official Review · Reviewer_38p5 · 2021-10-24

**Correctness:** 4
**Technical Novelty And Significance:** 3
**Empirical Novelty And Significance:** 3
**Recommendation:** 6
**Confidence:** 4

**Main Review:**

### Strengths

[Significance]

All reduce is a very popular architecture for large scale distributed training with the case of Byzantine tolerance being an open problem. The results of this paper are important.

[Novelty]

The proposal builds on multiple existing schemes that come from both the ML optimization literature (centered clip, butterfly all reduce) and distributed algorithms (e.g., DHT, MPRNG).

[Quality]

There is sufficient technical depth for the main claims of the paper (both for theoretical guarantees and for system design choices). The authors discuss their main assumptions and clarify cases in which they don't apply.

[Presentation / Clarity]

The paper is well written and easy to follow. Parameter choices are provided and the extensive supplementary material contain the necessary details for the proofs and complementary explanations of the method.

### Weaknesses

[Significance]

- The assumptions needed for the algorithm, prohibit an extension of the work to a setup with decentralized data (such as in federated learning). The problem of Byzantine tolerance is less motivated in the case where each node can access the entire training dataset as in this case it is often preferable to scale out in a more controlled environment (e.g., data centers) where crash failures are present but Byzantine ones are much less likely to occur. The strict IID assumptions for the paper are limiting. Convergence guarantees can still be achieved with some bound on the dissimilarity among the datasets as in practice the datasets are not arbitrarily different. Byzantine-resilient learning on decentralized data is a challenging yet open problem. For reference see the submission and discussion here: https://openreview.net/forum?id=7JSTDTZtn7-

- The reputation system for nodes that join and leave introduces another limitation for the proposal, as it requires an additional application-dependent hyperparameter (T).

[Evaluation]

- The paper targets the large-scale setup. However, there is no evaluation of the scalability of the proposal and only experiments with up to 16 participants are shown.

- Missing experimental results for the case of nodes joining.



**Summary Of The Paper:**

This paper introduces BTARD-SGD as a new algorithm for distributed training among untrusted Byzantine nodes. The underlying all reduce architecture makes the proposal appealing for large scale applications due to the reduction in the communication cost. The paper contains a convergence analysis for convex and non-convex loss functions and a proof for resilience against Byzantine and Sybil attacks. The authors verify the efficacy of their proposal on image classification and NLP tasks.

**Summary Of The Review:**

Although the paper has some significant weaknesses, I tend to recommend an acceptance given the technical depth and significance of the problem addressed.

---

> ### Author Response · Authors · 2021-11-11
> **Official Response to Reviewer 38p5**
>
> We thank the reviewer for their feedback. We appreciate the positive evaluation of our work: Reviewer 38p5 acknowledges the significance and novelty of our approach and states that our paper is "well written and easy to follow". We address the reviewer's concerns below.
>
> > The paper targets the large-scale setup. However, there is no evaluation of the scalability of the proposal and only experiments with up to 16 participants are shown.
>
> We provide experiments with 64 peers (including 31 Byzantines) in Appendix I.3. We refer to this appendix at the end of Section 4.2.
>
> > The strict IID assumptions for the paper are limiting. Convergence guarantees can still be achieved with some bound on the dissimilarity among the datasets as in practice the datasets are not arbitrarily different. Byzantine-resilient learning on decentralized data is a challenging yet open problem. For reference see the submission and discussion here: https://openreview.net/forum?id=7JSTDTZtn7-
>
> We refer to the updated version of this paper in footnote 4 (page 6) and discuss its results further in Appendix E.2. The referred paper proves that one cannot achieve any predefined accuracy of the solution of the optimization problem with heterogeneous functions available on the clients  **even if the dataset dissimilarity is bounded**. Although the authors of that paper show that under the additional assumption holding for some over-parameterized models one can circumvent the given lower bound, the corresponding result typically implies that $\delta$ should be too small.
>
> Therefore, we emphasize that the IID assumptions are not a weakness of our approach but rather an **inevitable limitation** that we explicitly mention in the paper.
>
> > The problem of Byzantine tolerance is less motivated in the case where each node can access the entire training dataset as in this case it is often preferable to scale out in a more controlled environment (e.g., data centers) where crash failures are present but Byzantine ones are much less likely to occur.
>
> We emphasize that training with public data is not limited to controlled environments such as data centers. Many recent works run distributed training over the Internet, using collaborations of volunteers [1, 2, 3, 4] (see references below) or cheap commodity instances [5, 6] to alleviate the compute costs of training large models. In many of these cases, a model is just too large to be trained by any of the participants alone.
>
> > The reputation system for nodes that join and leave introduces another limitation for the proposal, as it requires an additional application-dependent hyperparameter (T).
>
> First of all, we emphasize that the reputation system is a minor contribution of our paper needed to extend the applicability of our approach to the case of Sybil attacks. This lies beyond the scope of the *classical* setup of Byzantine-tolerant optimization. Please note that many previous works on Byzantine robustness do not consider such a general scenario at all. Moreover, as our Table 2 from Appendix A.1.2 shows, our convergence guarantees are strongly superior to the best known ones in decentralized settings and even better than SOTA rates derived for parameter-server-based approaches when the target accuracy $\varepsilon$ of the solution is small enough.
>
> Secondly, in the case of Sybil attacks, our reputation system indeed introduces a new application-dependent hyperparameter. However, we discuss how to choose it in Appendix F (the end of page 56) and observe that using even a conservative upper estimate of $T$ would not harm the method's efficiency in the setups considered in our experiments.
>
> References:
>
> [1] ​​Diskin, M., Bukhtiyarov, A., Ryabinin, M., Saulnier, L., Lhoest, Q., Sinitsin, A., Popov, D., Pyrkin, D., Kashirin, M., Borzunov, A. and del Moral, A.V., 2021. Distributed Deep Learning in Open Collaborations. arXiv preprint arXiv:2106.10207.
>
> [2] Kijsipongse, E., Piyatumrong, A. and Suriya, U., 2018. A hybrid GPU cluster and volunteer computing platform for scalable deep learning. The Journal of Supercomputing, 74(7), pp.3236-3263.
>
> [3] Atre, M., Jha, B. and Rao, A., 2021. Distributed Deep Learning Using Volunteer Computing-Like Paradigm. arXiv preprint arXiv:2103.08894.
>
> [4] Ryabinin, M. and Gusev, A., 2020. Towards crowdsourced training of large neural networks using decentralized mixture-of-experts. arXiv preprint arXiv:2002.04013.
>
> [5] Ligeng Zhu and Yao Lu and Yujun Lin and Song Han. 2020. Distributed Training Across the World.
>
> [6] Max Ryabinin, Eduard Gorbunov, Vsevolod Plokhotnyuk, and Gennady Pekhimenko. Moshpit SGD: Communication-efficient decentralized training on heterogeneous unreliable devices. arXiv preprint arXiv:2103.03239, 2021.

---

> > ### Comment · Reviewer_38p5 · 2021-11-29
> > **Comments on the response**
> >
> > I would like to thank the authors for their response. The response contains some valuable information. However, except from the concern regarding the large-scale setup, all my other concerns remain. I still believe this is a borderline work based on my review.

---

> > > ### Author Response · Authors · 2021-11-29
> > > **Response to the comment of Reviewer 38p5**
> > >
> > > We thank the reviewer for acknowledging that we have addressed the concern regarding large-scale experiments. However, in our previous response, we also did our best to address the two other reviewer’s concerns, and we would like to understand why it was not enough.
> > >
> > > We kindly ask the reviewer to clarify their position on the two remaining concerns: the IID assumption and the resistance to Sybil attacks. To help mutual understanding, we briefly summarize our position below.
> > >
> > > The first concern is related to the limitations of the IID assumption on the data and motivation of the setup when all workers access the entire dataset. As we explain in our response and the paper, the IID-data assumption is not a limitation of our work but rather the limitation of Byzantine-tolerance in general. Moreover, we provide several references to the papers where this setup is well-motivated and supported by particular examples of the problems in these settings.
> > >
> > > The second concern is about the limitations of BTARD’s resistance against Sybil attacks. We acknowledge that our solution has some limitations, but it is still superior to any existing distributed optimization algorithms. In fact, none of the SOTA papers on Byzantine-robust optimization provide any guarantees against Sybil attacks.
> > >
> > > To the best of our knowledge, the reviewer’s criticism of our solution is focused around the fact that it requires an additional hyperparameter. We strongly believe that **having an extra hyperparameter for Sybil resistance is better than having no Sybil resistance at all**.

---

### Official Review · Reviewer_ojVo · 2021-11-02

**Correctness:** 2
**Technical Novelty And Significance:** 2
**Empirical Novelty And Significance:** 2
**Recommendation:** 3
**Confidence:** 2

**Main Review:**

This paper addresses a very interesting task of machine learning under a decentralized setting. As the authors mention, ensuring Byzantine fault tolerance in the learning system is very hard under a general learning setting. Some of the requirements will need to be relaxed, as is the case in recent decentralized learning works (just a few examples: I know they do not discuss Bizantine):

- Tran et al., "An efficient approach for privacy preserving decentralized deep learning models based on secure multi-party computation," Neurocomputing 422(2021) 245-262
- Ide et al., "Efficient Protocol for Collaborative Dictionary Learning in Decentralized Networks," Proceedings of the 28th International Joint Conference on Artificial Intelligence (IJCAI 19,  August 10-16, Macao, China), pp.2585-2591

In this paper, the authors use leader selection schemes for "validator" and "accuse", which does not make the protocol entirely decentralized even if the selection is random. The authors need to clarify the main challenge of the existing methods and the key achievement of the paper. The first half of the paper does not look very successful in clarifying research motivation.

Questions
p.3
"that permutation-invariant algorithms cannot converge to any predefined accuracy of the solution" sounds unreasonable because the original aggregation is permutation symmetric. Please elaborate.

p.4
Is there any risk assessment of the validator selection step?

p.5
No definition of Split.


**Summary Of The Paper:**

This paper seems to propose a new use-case of an existing Byzantine fault-tolerant protocol called CenteredClip in a federated learning setting. Although the authors state that their approach fits the decentralized learning setting, the protocol relies on some sort of leader selection.

The authors bring CenterClip's equation without explaining the problem setting to a reasonable clarity. It seems the use of multi-party random number generation for Eq.(1) is the main novelty, but its context is not clear.

The authors make two assumptions on the objective function on which distributed members strive to compute the gradient, but the novelty and the context are unclear.

**Summary Of The Review:**

- unclear description
- limited novelty

---

> ### Author Response · Authors · 2021-11-11
> **Official Response to Reviewer ojVo**
>
> We thank the reviewer for their feedback. However, we are afraid that there may have been a misunderstanding about the main contributions of our work. We do our best to clarify this misunderstanding below and are open to further discussion if the reviewer deems it necessary.
>
> > This paper seems to propose a new use-case of an existing Byzantine fault-tolerant protocol called CenteredClip in a federated learning setting.
>
> In the very first sentence of Section 3 (Method), we explicitly state that we consider training on public datasets. In contrast, Federated Learning focuses on training with private data. Moreover, our contribution goes far beyond just the application of CenteredClip (see our response to Reviewer 63QJ, Part 4/5, reply to comment 10).
>
> > the protocol relies on some sort of leader selection
> > In this paper, the authors use leader selection schemes for "validator" and "accuse", which does not make the protocol entirely decentralized even if the selection is random.
>
> Our algorithm does not involve leader selection. The “accuse” procedure can be performed by any peer at any point. As for validators, BTARD peers elect **multiple** random peers to serve as validators. For instance, Figure 8 reports training with 4 validators. Being a validator lasts for a single optimizer step. Furthermore, the validators are not trusted and BTARD is tolerant to the case when the validators are Byzantines themselves. We outline this idea in Section 3 and elaborate in Appendices C and D.3. Furthermore, we report experiments with multiple validators in Figure 8.
>
> > The authors need to clarify the main challenge of the existing methods and the key achievement of the paper.
>
> The main challenges of the existing methods are summarized at the end of page 1, and the key achievements of the paper are highlighted at the beginning of page 2.
>
> > The authors make two assumptions on the objective function on which distributed members strive to compute the gradient, but the novelty and the context are unclear.
>
> We explain the assumptions on the objective function in Appendix E.3.1. In short, Assumption 3.1 is a modification of the standard uniformly bounded variance (UBV) assumption from literature (but can be easily replaced by UBV assumption if the blocks of the vectors are arranged uniformly at random in BTARD). In turn, Assumption 3.2 holds for a large class of practically important problems as shown in prior work.
>
> > Questions p.3 "that permutation-invariant algorithms cannot converge to any predefined accuracy of the solution" sounds unreasonable because the original aggregation is permutation symmetric. Please elaborate.
>
> CenteredClip is indeed permutation invariant. For this reason, both BTARD and prior works rely on additional techniques to break the symmetry. For instance, [1] (see the references below) maintains a “client momentum” that represents the “history” of a given peer’s past updates. Unfortunately, this technique relies on a central server to maintain each client’s momentum, which is impractical in our setup. In turn, BTARD is not permutation invariant because a peer that behaves maliciously at step $t$ will be banned for all subsequent steps if it is spotted by the validators.
>
> > p.4 Is there any risk assessment of the validator selection step?
>
> Yes, we outline it in Section 3.1 (paragraph 4), describe how we take this into account in the experiments at the beginning of page 8, and provide a detailed analysis in Appendix C.
>
> > p.5 No definition of Split.
>
> Thank you for pointing this out. While Split is a standard function in vector computing packages such as NumPy and PyTorch, we agree that it would be beneficial to define it for the sake of formalism. We will add the definition of Split and other auxiliary functions in the nearest update.
>
> > limited novelty
>
> We politely disagree with this assessment. BTARD-SGD is the first **decentralized** Byzantine-robust optimization method that has convergence guarantees that are not worse than for Parallel SGD without Byzantine attacks. The guarantees are based on solid theoretical arguments that hold under quite general assumptions. We provide an explicit comparison with previous SOTA results in Table 2 from Appendix A.1.2. In brief, our theoretical guarantees are strongly superior to the best known ones in decentralized settings and even better than SOTA rates derived for parameter-server-based approaches when the target accuracy $\varepsilon$ of the solution is small enough.
>
> References:
>
> [1] Sai Praneeth Karimireddy, Lie He, and Martin Jaggi. Learning from history for byzantine
> 518 robust optimization. arXiv preprint arXiv:2012.10333, 2020.

---

> > ### Comment · Reviewer_ojVo · 2021-11-29
> > **Thank you for your reply**
> >
> > Thank you for your reply. I am sorry for not responding timely.
> >
> > Some of the comments clarified my questions, but I am still not convinced with the novelty of the paper. I still have difficulties in getting what practical benefits you can get through the proposed algorithm. Perhaps the authors' motivation would be almost purely theoretical. But the semi-application-oriented way of introducing the problem in Introduction still confuses me.
> >
> > To get the rating their paper deserves, the authors need to make sure that all the readers immediately understand the research motivation you followed just by reading the first few sections. The paper is supposed to be self-contained. A paper is not a court document. Readability matters.
> >
> > I cannot change my recommendation, but I just lowered the confidence score because I do not have complete knowledge of some of the prior works.

---

> > > ### Author Response · Authors · 2021-11-30
> > > **Response to Reviewer ojVo**
> > >
> > > Thank you for the reply.
> > >
> > > > I still have difficulties in getting what practical benefits you can get through the proposed algorithm. Perhaps the authors' motivation would be almost purely theoretical.
> > >
> > > We emphasize that our motivation is not purely theoretical.
> > >
> > > One real-life example of where our algorithm could be useful is described in [1] (see references below). We cite this paper while motivating our work in Section 1. In this paper, the authors note that there are only a few pretrained language models available for the Bengali language (despite that it has over 230M native speakers who can benefit from the applications built using such models). The authors recruited Bengali-speaking volunteers and outside collaborators to pretrain a large language model in a distributed way using their hardware and cloud platform accounts. They used only openly available datasets and trained a model that beats state-of-the-art. It is unlikely that any of the participants could train such a model alone, without participating in this collaboration.
> > >
> > > However, in this experiment, any single participant could jeopardize the entire training run by sending incorrect updates, whether deliberately or due to a hardware error. This would not be possible in the case of using our algorithm.
> > >
> > > Other possible use cases are:
> > >
> > > - Several small companies making image editing apps can collaborate to train a large multimodal model (like OpenAI's DALL-E or CLIP) that can improve their products. They can use openly available large image-text datasets such as [2] and the combined computing power of the companies' clusters.
> > > - Several universities can collaborate to train a large language model with specific architecture for research purposes. They can use openly available large text datasets such as [3] and the combined computing power of their hardware.
> > >
> > > > authors need to make sure that all the readers immediately understand the research motivation you followed just by reading the first few sections.
> > >
> > > Our paper already starts with describing the research motivation.
> > >
> > > At the beginning of Section 1, we describe volunteer computing and scientific collaboration setups that motivated our work (our intended use-cases). In the 2nd paragraph of this section, we cite many related papers proving that these setups are practical for deep learning and studied in literature. Some of the papers cited there (such as [4], [1]) explicitly acknowledge the need for algorithms protecting the collaborations from the malicious workers.
> > >
> > > Moreover, the motivation is summarized in the first two sentences of the abstract.
> > >
> > > > I cannot change my recommendation
> > >
> > > We did our best to answer all questions and address all mentioned concerns, and we would like to understand why it is not enough. We kindly ask you to explain the remaining concerns motivating your evaluation of our work.
> > >
> > > References:
> > >
> > > [1] Diskin, Michael, et al. "Distributed Deep Learning in Open Collaborations." arXiv preprint arXiv:2106.10207 (2021).
> > >
> > > [2] ​​Schuhmann, Christoph, et al. "LAION-400M: Open Dataset of CLIP-Filtered 400 Million Image-Text Pairs." arXiv preprint arXiv:2111.02114 (2021).
> > >
> > > [3] Gao, Leo, et al. "The Pile: An 800GB Dataset of Diverse Text for Language Modeling." arXiv preprint arXiv:2101.00027 (2020).
> > >
> > > [4] Ryabinin, Max, and Anton Gusev. "Towards crowdsourced training of large neural networks using decentralized mixture-of-experts." arXiv preprint arXiv:2002.04013 (2020).

---

### Official Review · Reviewer_63QJ · 2021-11-02

**Correctness:** 2
**Technical Novelty And Significance:** 2
**Empirical Novelty And Significance:** 1
**Recommendation:** 3
**Confidence:** 4

**Main Review:**

Thanks for submitting your work to ICLR2022. The paper addresses a very important and timely problem. While there is a lot of work addressing Byzantine resilience in ML, we need works that focus on their practicality in terms of performance and assumptions about the environment. I also like the motivating example of crowdsourcing computational resources from (possibly) untrusted parties. I appreciate the convergence rates that the authors derive in different scenarios (strongly convex, convex, and non-convex functions), and I like the fact that the overhead of the proposed algorithm does not depend on the model size. We, as a community, need to always think of such solutions (whose overhead is decoupled from the model size).

Yet, I believe the paper cannot be accepted as-is as it requires to be revised in a few aspects. In summary, (1) the authors make several assumptions that seem unrealistic and unjustified. (2) The most important parts of the paper are either unclear or deferred to the appendices; this makes it hard to understand the algorithm and to see its benefits. (3) As the main aim of the paper is to provide a computationally efficient algorithm, the paper should give clearly the computation complexity of all parts of the algorithm.

The first area of improvement is to clarify and back up the assumptions. Specifically, I'd like to point out 3 assumptions: The first is the public dataset assumption. Doesn't that make the problem trivial? Can you give examples in which this assumption is practical? Furthermore, many parts of the main algorithm require synchronous networks. It's important that the authors clearly state this in their assumptions. Synchronous networks are easy to achieve in a controlled environment like a datacenter. Yet, in a more challenging environment that the authors consider, I'm not sure how synchronicity can be achieved. The third uncommon assumption is about adversary capabilities. I do not understand the need for the second paragraph on page 6 (which describes the possible attacks). Byzantines should be able to do whatever they want [1,2]. Putting them in categories in which they can do only limited things raises the question of whether the considered model is really for Byzantine attacks or not.
[1] Lamport, Leslie. "The weak Byzantine generals problem." Journal of the ACM (JACM) 30.3 (1983): 668-676.
[2] Castro, Miguel, and Barbara Liskov. "Practical byzantine fault tolerance." OSDI. Vol. 99. No. 1999. 1999.

There are essential parts in the paper that are not described clearly and left to the appendices. One example is how the broadcast channels work in Section 2.3. The authors seem to describe consensus. Is that it? If yes, this concept is orthogonal to multi-party computation (MPC). If not, then what is the difference between the suggested method and consensus? The same comment goes for the main algorithm in the paper. Essentially, there is no appropriate (even high-level, intuitive) explanation of why it works, with all the proofs deferred to the appendices. The most important example is the CenteredClip algorithm, which is the main tool to tolerate the Byzantines. For example (more examples below in the additional comments/questions), from the description, I understand that the input to this algorithm is two gradients: the local one and the one gathered from all peers (by merging a small part of the gradient from each peer). If my understanding is correct, I do not see how clipping (based on two inputs only) can be effective in tolerating Byzantine behavior. It's crucial to understand at least intuition without looking at complex mathematical proofs. I also expected to see even proof sketches in Sections 3.2 and 3.3. The absence of these parts makes it very hard to judge the effectiveness of the proposed work.

Parts of the algorithm are not clear and the computation complexity is not rigorously quantified. For example, what is the cost of the validation step? If we need a lot of validators, why do we need distributed learning in the first place? If peers can compute their gradients and the others' gradients (for validation), why bother distributing? In addition, I believe many parts of the algorithm rely on consensus (even if that was never explicitly mentioned by the authors). Consensus is costly, and a careful analysis of its complexity is required. Along the same direction, I have two issues with the "accuse" procedure: (1) it's very costly because all peers have to compute the gradient of the accused peer; a Byzantine can use this trick to make learning very slow. (2) The procedure relies on consensus, which is also costly. Another unclear part is the cost of sampling the random direction z among all peers.

Additional comments/questions:
1. The authors mentioned that existing solutions either require a trusted server or have prohibitive communication costs. I know at least one paper [3] that does not require either. Can the author comment on that?
[3] El-Mhamdi, E. M., Guerraoui, R., Guirguis, A., Hoang, L. N., & Rouault, S. (2020, July). Genuinely distributed byzantine machine learning. In Proceedings of the 39th Symposium on Principles of Distributed Computing (pp. 355-364).

2. The third paragraph on page 6 is problematic. The authors mentioned an attack in which the system removes one Byzantine and one benign node. In this case, the Byzantines can do this attack until 2f nodes are removed (with f being the original number of Byzantines). Since the algorithm requires n > 2f+1 then, in extreme cases, this attack leaves the system with only 1 node! In other words, the authors seem to acknowledge that a Byzantine node can defeat the distribution of the algorithm, making it behave as if there is only one central node.

3. I do not understand the last line of Section 2.1. Why does **each** peer require O(dn) in the PS architecture? I believe each one requires only 2d: 1d to upload the gradient and 1d to download the model. Can you please clarify this?

4. The first sentence of Section 3.1 ("The core design principle behind our algorithm is that all types of Byzantine faults must have limited effect and a chance of being discovered.") is a bit weird. Usually, Byzantine attacks are posed as the strongest attack to diverge learning. It is strong because Byzantine peers can do whatever they want (i.e., it's also called: behave arbitrarily).

5. It's not clear if clipping is effective against random attacks? I understand it will be robust against gradients with big magnitudes (like the attacks in the evaluation section) but what about the other attacks?

6. It's not clear how the butterfly method works? If you only broadcast a part of the gradient, then what is aggregated in this case? Is it only the local part of a peer and a corresponding part received from another peer?

7. In Algorithm 2, a Byzantine can equivocate as follows. It can send different hashes (for the same part of the gradient) to different peers. It can then send different values for that part to different peers so that the hashes match the received gradients on each peer. Is that a problem for your algorithm?

8. The validation step is not rigorous because you might be unlucky and get the Byzantine node to validate another Byzantine node in one iteration. Clearly, Byzantines can get away with their attack (without being caught) in this iteration. What are the practical/theoretical results of such a situation?

9. The solution to Sybil attacks (in Section 3.2) is not convincing. A Byzantine node can prove its honesty only to enter the system and then it can do whatever it wants. This problem (i.e., tolerating Sybil attacks) is interesting and I encourage the authors to give it more discussion in the main paper.

10. The authors seem to rely on (Karimireddy et al., 2020) a lot. What are the main differences between this work compared to this work? Is it only PS architecture vs. decentralized architecture? If so, what are the additional challenges that you solved?

11. The federated learning (FL) example in the motivation does not fit nicely. As mentioned in the paper, one important aspect of FL is data privacy yet, the solution in the paper relies on the assumption that the dataset is public and can be accessible by anybody. I advise the authors to remove FL from their motivating examples.

12. The authors chose the clipping value (\tau) based on out-of-band experiments. Can the authors add a discussion on how to choose a clipping value practically (without initial experiments)?

13. What are the coordinate-wise trimmed median and geometric median baselines? It's clear how to use these aggregators with a central server yet, they are not well-defined in the decentralized case. If the authors are referring to some work in the literature, please add the necessary citations.

14. As the authors mentioned, they use only specific parts of MPC rather than a fully-fledged MPC protocol. I suggest the authors do not use this term and use specific terms of the specific algorithms they use instead.

15. The expression "zero trust assumption" is a bit of a stretch. For example, you need at least 50% of honest peers.

16. What is the percentage of the validators (i.e., what is the inequality between m and n)? How does this percentage affect the correctness of the algorithm?

17. The authors mentioned that m validators will validate the work of m peers. It seems from the evaluation section that there is a one-to-one mapping, in this case, i.e., each validator validates one peer. Please make this clear while describing the algorithm.

18. How can one report a Byzantine node? This again requires consensus, right?

19. It will be great if the authors can add to the paper an explanation of how CenteredClip works.

20. I suggest you better explain \tau in Equation (2). Currently, it is not clear in the text; I understood what it means only in the evaluation section.

21. In Algorithm 2, Step 7: you either mean "broadcast" instead of "send" or you mean g_i[j] instead of g_i. The same fro Step 8. I believe you mean the latter though (given Step 9 is "MERGE" rather than "AGGREGATE").

22. A few important function definitions are missing: CenteredClip, ValidatePeer, VoteForBan, and Ban. Other functions are also missing, but they are less important.

23. Footnote 4 is a bit off because the paper does not consider the heterogeneous case.

24. I do not buy the omission of "common low-magnitude attacks (Baruch et al., 2019; Xie et al., 2020; Allen-Zhu et al., 2021)". These are the SOTA attacks, and it is important to show how your algorithm behaves under these attacks.

25. What is the attack in Figure 2 (first two figures)?

26. The upper-middle plot of Figure 2 shows that the geometric median is almost as good as your algorithm. What is the benefit of your algorithm in this case?

27. Why does the coordinate-wise median diverge in the upper-left figure in Fig. 2?

**Summary Of The Paper:**

The paper proposes BTARD-SGD, which is a Byzantine-tolerant training algorithm (via stochastic gradient descent) in decentralized environments (i.e., peer-to-peer architectures). BTARD-SGD does not require the presence of a trusted central server and introduces only a marginal overhead compared to a vanilla All-Reduce SGD algorithm. BTARD-SGD relies on an existing clipping algorithm in the literature called CenteredClip in addition to a series of checks to catch Byzantine nodes which might not follow the algorithm. The authors provide theoretical analysis in addition to a fair amount of experiments.

**Summary Of The Review:**

Though the paper addresses a very important and timely problem, I believe it cannot be accepted as-is as it requires to be revised in a few aspects. In summary, (1) the authors make several assumptions that seem unrealistic and unjustified. (2) The most important parts of the paper are deferred to the appendices; this makes it hard to understand the algorithm and to see its benefits. (3) As the main aim of the paper is to provide a computationally efficient algorithm, the paper should give clearly the computation complexity of all parts of the algorithm.

---

> ### Author Response · Authors · 2021-11-11
> **References for the Official Response to Reviewer 63QJ**
>
> [1] Chuan Li. Demystifying gpt-3 language model: A technical overview. "https://
> lambdalabs.com/blog/demystifying-gpt-3".
>
> [2] ​​Diskin, M., Bukhtiyarov, A., Ryabinin, M., Saulnier, L., Lhoest, Q., Sinitsin, A., Popov, D., Pyrkin, D., Kashirin, M., Borzunov, A. and del Moral, A.V., 2021. Distributed Deep Learning in Open Collaborations. arXiv preprint arXiv:2106.10207.
>
> [3] Kijsipongse, E., Piyatumrong, A. and Suriya, U., 2018. A hybrid GPU cluster and volunteer computing platform for scalable deep learning. The Journal of Supercomputing, 74(7), pp.3236-3263.
>
> [4] Atre, M., Jha, B. and Rao, A., 2021. Distributed Deep Learning Using Volunteer Computing-Like Paradigm. arXiv preprint arXiv:2103.08894.
>
> [5] Ryabinin, M. and Gusev, A., 2020. Towards crowdsourced training of large neural networks using decentralized mixture-of-experts. arXiv preprint arXiv:2002.04013.
>
> [6] Gao, L., Biderman, S., Black, S., Golding, L., Hoppe, T., Foster, C., Phang, J., He, H., Thite, A., Nabeshima, N. and Presser, S., 2020. The Pile: An 800GB Dataset of Diverse Text for Language Modeling. arXiv preprint arXiv:2101.00027.
>
> [7] Raffel, C., Shazeer, N., Roberts, A., Lee, K., Narang, S., Matena, M., Zhou, Y., Li, W. and Liu, P.J., 2019. Exploring the limits of transfer learning with a unified text-to-text transformer. arXiv preprint arXiv:1910.10683.
>
> [8] Narayanan, D., Shoeybi, M., Casper, J., LeGresley, P., Patwary, M., Korthikanti, V.A., Vainbrand, D., Kashinkunti, P., Bernauer, J., Catanzaro, B. and Phanishayee, A., 2021. Efficient large-scale language model training on gpu clusters. arXiv preprint arXiv:2104.04473.
>
> [9] You, Y., Li, J., Reddi, S., Hseu, J., Kumar, S., Bhojanapalli, S., Song, X., Demmel, J., Keutzer, K. and Hsieh, C.J., 2019. Large batch optimization for deep learning: Training bert in 76 minutes. arXiv preprint arXiv:1904.00962.
>
> [10] Brown, T.B., Mann, B., Ryder, N., Subbiah, M., Kaplan, J., Dhariwal, P., Neelakantan, A., Shyam, P., Sastry, G., Askell, A. and Agarwal, S., 2020. Language models are few-shot learners. arXiv preprint arXiv:2005.14165.
>
> [11] You, Y., Gitman, I. and Ginsburg, B., 2017. Large batch training of convolutional networks. arXiv preprint arXiv:1708.03888.
>
> [12] Das, S., Krishnan, V., Isaac, I.M. and Ren, L., 2021. SPURT: Scalable Distributed Randomness Beacon with Transparent Setup. IACR Cryptol. ePrint Arch., 2021, p.100.
>
> [13] Kiayias, A., Russell, A., David, B. and Oliynykov, R., 2017, August. Ouroboros: A provably secure proof-of-stake blockchain protocol. In Annual International Cryptology Conference (pp. 357-388). Springer, Cham (section "5.2 Simulating a Trusted Beacon", pp. 380-381).
>
> [14] ​​Beimel, A., Omri, E. and Orlov, I., 2010, August. Protocols for multiparty coin toss with dishonest majority. In Annual Cryptology Conference (pp. 538-557). Springer, Berlin, Heidelberg.
>
> [15] Katz, J., Ostrovsky, R. and Smith, A., 2003, May. Round efficiency of multi-party computation with a dishonest majority. In International Conference on the Theory and Applications of Cryptographic Techniques (pp. 578-595). Springer, Berlin, Heidelberg.
>
> [16] Kshemkalyani, A.D. and Singhal, M., 2011. Distributed computing: principles, algorithms, and systems. Cambridge University Press (section "14.1.1.2 The Consensus Problem", page 503).

---

> > ### Comment · Reviewer_63QJ · 2021-11-22
> > **Response to authors' rebuttal**
> >
> > I acknowledge that I read the authors' rebuttal and the updated version; I find the rebuttal very detailed, and I thank the authors for the time and efforts they spent to reply to my comments, concerns, and questions and update the paper. Although the paper has improved, the main major concerns I had still exist
> >
> > Precisely, I still do not understand what the use-case of the setup the authors target in the paper is. Using general words like "practitioners", "computing power", ..etc. do not help. Who is performing training? Companies, hospitals, banks, universities, somebody else? I strongly advise the authors to think of a concrete example for their paper. Without this, it is hard to judge if the assumptions make sense or not. For example, I asked a question about how to achieve synchrony in the environment the authors mentioned, but I did not get an answer. I believe this happens simply because the environment itself is (or at least seems to be) not clear to the authors. Another critical example is the public dataset assumption. Once the reader understands the environment, they can judge if this assumption is valid or not.
> >
> > I am still against grouping Byzantine attacks into categories. The beauty of the Byzantine abstraction is that it is general enough to abstract all kinds of behavior/attacks. Putting it into groups breaks this rule. Even if you decide to do so, it is helpful to cite relevant papers which did the same (to be honest, I am not aware of any). I advise the authors to rewrite their proof in a more general way.
> >
> > Many parts of the paper are still unclear (as pointed out by other reviewers as well). I advise the authors to rethink the paper's organization and main content to make it more clear to the readers. The appendices do not help much if the main manuscript is unclear.
> >
> > The authors mentioned in their rebuttal "the protocols for Byzantine broadcast". What are these protocols? What is the Byzantine broadcast abstraction? Byzantine broadcast is a "family" of abstractions rather than a well-defined algorithm. Byzantine broadcast could be, for example, reliable or uniform reliable broadcast or simple majority voting with a leader election. The authors need to define clearly which algorithm they use and what the computation and communication complexities are in the best and worst cases. For example, leader election is impossible in asynchronous environments. Defining well the abstractions used helps the reader judge the proposed algorithm.
> >
> > The comparison with the coordinate-wise trimmed median and geometric median baselines is not fair because they are centralized approaches (according to the authors). I advise the authors to add comparisons with more relevant approaches (i.e., algorithms for decentralized Byzantine ML) as baselines.

---

> > > ### Author Response · Authors · 2021-11-24
> > > **Response to Reviewer 63QJ (Part 2/2)**
> > >
> > > > The authors mentioned in their rebuttal "the protocols for Byzantine broadcast". What are these protocols? What is the Byzantine broadcast abstraction? Byzantine broadcast is a "family" of abstractions rather than a well-defined algorithm. Byzantine broadcast could be, for example, reliable or uniform reliable broadcast [...] The authors need to define clearly which algorithm they use and what the computation and communication complexities are in the best and worst cases.
> > >
> > > We intentionally avoid terms "reliable broadcast" and "uniform reliable broadcast" because they often refer to the protocols tolerant only to crash failures (not Byzantine attacks) in literature. Instead, **we cite specific papers** with suitable protocols for Byzantine broadcast while introducing them in Section 2.3: _"To protect against this attack, distributed systems can use broadcast channels built on top of the protocols for Byzantine broadcast (Pease et al., 1980; Dolev & Strong, 1983; Hirt & Raykov, 2014)."_
> > >
> > > In the latest version of the paper, we have adjusted the wording to make it more clear that the reference leads to the specific protocol: _"To protect against this attack, distributed systems can use broadcast channels built on top of the protocol for Byzantine broadcast from Dolev & Strong (1983)."_
> > >
> > > Next, we refer to Appendix A.1.2 where **we define the notion of "the protocols for Byzantine broadcast"**, cite several protocols satisfying this definition, and explain their guarantees. Next, in Appendix B ("Network and compute overhead of BTARD-SGD"), we state the communication complexity of the protocol from Dolev & Strong (1983) (same for the best and the worst cases). Its computation complexity is the same, because the protocol does not involve any non-linear processing of the communicated data.
> > >
> > > In practice, both communication and computation complexity of this protocol is dominated by $O(d)$ communication involved in All-Reduce and $O(d)$ computation involved in calculating the gradients (where $d$ is the size of the gradient vector), since the models that benefit from distributed training usually contain at least tens of millions of trainable parameters. This is why we focus on the All-Reduce complexity in the main part of our paper.
> > >
> > > > The comparison with the coordinate-wise trimmed median and geometric median baselines is not fair because they are centralized approaches (according to the authors).
> > >
> > > This comparison is intentionally not fair, but only in the sense that **baselines have an unfair advantage** of having a trusted participant (the central server). Informally, the disadvantages of centralized approaches are (1) slower iterations, when bottlenecked by a central server, and (2) being inapplicable if no trusted server is available.
> > >
> > > To prove our point, we show that BTARD can match the **per-iteration** performance of centralized methods while being faster and more broadly applicable. To the best of our knowledge, no existing decentralized method can achieve the same final accuracy in presence of Byzantines.
> > >
> > > Furthermore, we note that BTARD can simulate the decentralized coordinate-wise median by running ButterflyClip with 0 steps (since it is initialized with the coordinate-wise median). In turn, the decentralized geometric median can be approximated by setting $\tau\rightarrow 0$ using the properties of CenteredClip (see Appendix D.1). However, these two methods are strictly inferior to BTARD as we demonstrate in Section 4. Namely, coordinate-wise median converges significantly slower, and geometric median has slightly worse performance while being more difficult to compute, as shown in Karimireddy et al. (2020).

---

> > > > ### Comment · Reviewer_63QJ · 2021-12-09
> > > > **Acknowledgement**
> > > >
> > > > I acknowledge I have read the author's final response.

---

> > > ### Author Response · Authors · 2021-11-24
> > > **Response to Reviewer 63QJ (Part 1/2)**
> > >
> > > Thank you for the feedback.
> > >
> > > > Precisely, I still do not understand what the use-case of the setup the authors target in the paper is. Using general words like "practitioners", "computing power", ..etc. do not help. Who is performing training? Companies, hospitals, banks, universities, somebody else? I strongly advise the authors to think of a concrete example for their paper.
> > >
> > > One concrete example of where BTARD-SGD could be useful is described in [1] (see references below). We cite this paper while motivating our work in Section 1. In this paper, the authors note that there are only a few pretrained language models available for the Bengali language (despite that it has over 230M native speakers who can benefit from the applications built using such models). The authors recruit 30 Bengali-speaking volunteers and 10 outside collaborators to pretrain a large language model in a distributed way using their hardware and cloud platform accounts. They use **only openly available data** (Wikipedia and the Bengali part of the OSCAR multilingual corpus) and train a model that beats state-of-the-art.
> > >
> > > In this experiment, any single participant could jeopardize the entire training run by sending incorrect updates, whether deliberately or due to a hardware error. However, this would not be possible in case of using BTARD-SGD.
> > >
> > > Other possible use cases are:
> > >
> > > - Several small companies making image editing apps can collaborate to train a large multimodal model (like OpenAI's DALL-E or CLIP) that can improve their products. They can use openly available large image-text datasets such as [2] and the combined computing power of the companies' clusters.
> > > - Several universities can collaborate to train a large language model with specific architecture for research purposes. They can use openly available large text datasets such as [3] and the combined computing power of their hardware.
> > >
> > > [1] Diskin, Michael, et al. "Distributed Deep Learning in Open Collaborations." arXiv preprint arXiv:2106.10207 (2021).
> > >
> > > [2] ​​Schuhmann, Christoph, et al. "LAION-400M: Open Dataset of CLIP-Filtered 400 Million Image-Text Pairs." arXiv preprint arXiv:2111.02114 (2021).
> > >
> > > [3] Gao, Leo, et al. "The Pile: An 800GB Dataset of Diverse Text for Language Modeling." arXiv preprint arXiv:2101.00027 (2020).
> > >
> > > > I asked a question about how to achieve synchrony in the environment the authors mentioned, but I did not get an answer. I believe this happens simply because the environment itself is (or at least seems to be) not clear to the authors.
> > >
> > > From an algorithmic standpoint, we addressed this question in Appendix B (the paragraph on “Synchronizaton points”). In short, BTARD **does not need synchronous networks** but instead relies on a small number of times where a peer needs to await a particular signal (or timeout) from another peer. In that regard, BTARD is not different from other known studies that achieve "synchronous" distributed training over the Internet such as [1].
> > >
> > > The only barrier used in BTARD that is not already present in non-Byzantine-tolerant training is that each peer needs to receive hashes from all peers before it reveals its private information for MPRNG. As we discuss in Appendix B, this additional barrier is implemented in a background thread, without interfering with the algorithm performance.
> > >
> > > From a technical point of view, we implement this with regular TCP connections and use an event loop based network layer (specifically, asyncio TCP server with uvloop). Each peer asynchronously awaits the necessary data from incoming TCP connection and verifies that it was signed with the correct signature before proceeding.
> > >
> > > > I am still against grouping Byzantine attacks into categories.
> > >
> > > As we discussed in our previous response, we do not limit Byzantines and do not assume that they adhere to a single attack type. We describe these attacks to provide reader with motivating examples of what byzantines **could** do and how this affects the system design of BTARD. In turn, our formal analysis does not require that Byzantine peers fall into one of the proposed categories.
> > >
> > > That said, thanks to Reviewer 63QJ’s comment, we now see that readers may be confused by the way we first refer to these attacks. To alleviate this, we have paraphrased the way these attacks are first introduced in Section 3.2 in the latest version of our paper.

---

> ### Author Response · Authors · 2021-11-11
> **Official Response to Reviewer 63QJ (Part 5/5)**
>
> > 17. The authors mentioned that m validators will validate the work of m peers. It seems from the evaluation section that there is a one-to-one mapping, in this case, i.e., each validator validates one peer. Please make this clear while describing the algorithm.
>
>
> We will add such a clarification to Section 3.1.
>
> > 18. How can one report a Byzantine node? This again requires consensus, right?
>
>
> Yes. As described in Algorithm 5, peers follow the broadcast channel to determine whether to ban the accused peer or ban the accuser for false allegation.
>
> > 19. It will be great if the authors can add to the paper an explanation of how CenteredClip works.
>
> Please refer to Appendix D.1.
>
> > 20. I suggest you better explain \tau in Equation (2). Currently, it is not clear in the text; I understood what it means only in the evaluation section.
>
> We will add a more detailed discussion to Appendix D.1.
>
> > 21. In Algorithm 2, Step 7: you either mean "broadcast" instead of "send" or you mean g_i[j] instead of g_i. The same fro Step 8. I believe you mean the latter though (given Step 9 is "MERGE" rather than "AGGREGATE").
>
> There are no mistakes of this kind in steps 7-8.
>
> First, the "broadcast" operation in the algorithm descriptions denotes the usage of the broadcast channel protocol described in Section 2.3. As explained above, this is different from sending the same data to each peer naively because this protocol involves extra communication between the peers to ensure that a sender could not break the consensus by sending different values to different peers. Due to this extra communication, we cannot use this protocol for large vectors (like the parts of the gradient vector in steps 7-8) if we need the communication overhead to be small (see details on overhead analysis in Appendix B).
>
> Algorithm 2 is a modified version of Butterfly All-Reduce (where naive averaging is replaced with the CenteredClip aggregation). As in the usual Butterfly All-Reduce, peer $i$ is aggregating the $i$-th part of the gradient vector from all participants and sends the result (denoted as $\hat g_i$) back to each participant. This is exactly what happens in step 7 (peer $i$ sends the same vector $\hat g_i$ to everyone) and step 8 (peer $i$ collects other aggregation results from everyone). Finally, the peer merges all aggregation results in step 9 to obtain the full aggregated gradient vector $\hat g$.
>
> We provide a scheme illustrating this process in the middle of Figure 1.
>
> > 22. A few important function definitions are missing: CenteredClip, ValidatePeer, VoteForBan, and Ban. Other functions are also missing, but they are less important.
>
> CenteredClip is described in Appendix D.1. We will provide explicit definitions for ValidatePeer, VoteForBan, and Ban in Appendix D.2.
>
> > 23. Footnote 4 is a bit off because the paper does not consider the heterogeneous case.
>
>
> This footnote is important to motivate why our work cannot be further generalized to the heterogeneous case. Also, it served as an advance response to questions from reviewers w1BH and ojVO.
>
> > 24. I do not buy the omission of "common low-magnitude attacks (Baruch et al., 2019; Xie et al., 2020; Allen-Zhu et al., 2021)". These are the SOTA attacks, and it is important to show how your algorithm behaves under these attacks.
>
>
> In our preliminary experiments, we found that those attack types behave as less efficient versions of high magnitude attacks, as they are designed to exploit vulnerabilities that BTARD does not have. However, we recognize that the paper would benefit from running additional experiments. We aim to provide them by November 20th.
>
> > 25. What is the attack in Figure 2 (first two figures)?
>
>
> These figures evaluate the training performance **without Byzantine peers**. They demonstrate that BTARD-SGD does not worsen the convergence speed and model quality in this case.
>
> > 26. The upper-middle plot of Figure 2 shows that the geometric median is almost as good as your algorithm. What is the benefit of your algorithm in this case?
>
> The benefit of BTARD over the geometric median and other baselines is that it can be performed in a decentralized setting, whereas the geometric median requires a trusted central server. Furthermore, the geometric median is notoriously difficult to compute, which is one of the reasons why methods like CenteredClip were created.
>
> > 27. Why does the coordinate-wise median diverge in the upper-left figure in Fig. 2?
>
> It does not diverge per se, but it needs almost 10 times more iterations to converge. This observation appears to coincide with Karimireddy et al. (2020), where coordinate-wise median converges slower and to a poorer optimum in several cases.

---

> ### Author Response · Authors · 2021-11-11
> **Official Response to Reviewer 63QJ (Part 4/5)**
>
> > 8. The validation step is not rigorous because you might be unlucky and get the Byzantine node to validate another Byzantine node in one iteration. Clearly, Byzantines can get away with their attack (without being caught) in this iteration. What are the practical/theoretical results of such a situation?
>
> We describe this and other special cases in Appendix C (before “protocol violations”). All our guarantees and experiments are adjusted for this.
>
> > 9. A Byzantine node can prove its honesty only to enter the system and then it can do whatever it wants.
>
> Please refer to Appendix F. If a byzantine peer enters a system and then starts attacking, such peer will be banned and will need to "prove its honesty" again. Since the "proving" requires a peer to run computations for a long amount of time, such peer will spend most of its time banned.
>
> > 10. The authors seem to rely on (Karimireddy et al., 2020) a lot. What are the main differences between this work compared to this work? Is it only PS architecture vs. decentralized architecture? If so, what are the additional challenges that you solved?
>
> Although the CenteredClip procedure from (Karimireddy et al., 2020) is an important part of our protocol, it is just one of many other important building blocks of our BTARD and BTARD-SGD. First of all, as one can see from the related work section, designing Byzantine-tolerant **decentralized** optimization methods with **strong theoretical guarantees** is a challenging task. As we explain in the second paragraph of Section 3.1 and at the beginning of Appendix D.3, a naive decentralized modification of the approach from (Karimireddy et al., 2020) is not robust to Byzantine attacks. Indeed, if we replace averaging by CenteredClip in Butterfly All-Reduce, we do not achieve robustness: Byzantine peers can arbitrarily modify gradients instead of honestly performing CenteredClip.
>
> This simple example highlights the huge difference between PS and decentralized setups. To circumvent this issue we have added several important modifications and verifications to the method: computation of hashes, Verification 1, 2, 3, and computation verifications after each step of BTARD-SGD. We rigorously analyzed the obtained algorithms and achieved the results that are better and more general than all previously known results in the decentralized case (and even better than some results in the centralized case) — see Table 2 in Appendix A.1.2.
>
> > 11. The federated learning (FL) example in the motivation does not fit nicely.
>
> We agree with this concern. While federated learning is close to our setup, we do not address it and it is not mentioned after Section 1. We will remove it in the nearest update.
>
> > 12. The authors chose the clipping value (\tau) based on out-of-band experiments. Can the authors add a discussion on how to choose a clipping value practically (without initial experiments)?
>
> We agree and will provide the additional discussion in the nearest update. In short, we choose $\tau$ based on the empiric variance of the local gradients. The std of local gradients was $\approx 1$ for the setup from Section 4.1 and $\approx 0.125$ for the ALBERT-large setup from Section 4.2. Our experiments suggest that the algorithm is robust in, at least, the 1-4x standard deviation range.
>
> In principle, it should be possible to automatically scale $\tau$ depending on the standard deviation. We leave this investigation to future work.
>
> > 13. What are the coordinate-wise trimmed median and geometric median baselines? It's clear how to use these aggregators with a central server yet, they are not well-defined in the decentralized case.
>
> In this case, we use centralized baselines.
>
> > 14. As the authors mentioned, they use only specific parts of MPC rather than a fully-fledged MPC protocol. I suggest the authors do not use this term and use specific terms of the specific algorithms they use instead.
>
> While most of the paper already refers to the specific protocols by their names and claims only the usage of "ideas used in MPC" (not fully-fledged MPC protocols), we agree that phrases like "we leverage secure multi-party computing" may be ambiguous. We will replace them in the nearest update.
>
> > 15. The expression "zero trust assumption" is a bit of a stretch. For example, you need at least 50% of honest peers.
>
> What we meant here is "zero trust security model", also known as the perimeterless security model. This model assumes that no specific peers can be fully trusted.
>
> > 16. What is the percentage of the validators (i.e., what is the inequality between m and n)? How does this percentage affect the correctness of the algorithm?
>
> In our experiments, we always use $1/16$ validators, which translates to 1 peer in Section 4.1 and 4.2 and 4 peers in Appendix I.3. We show that even with this number of validators, training recovers from the worst-case attack with almost half of the peers being Byzantine.

---

> ### Author Response · Authors · 2021-11-11
> **Official Response to Reviewer 63QJ (Part 3/5)**
>
> > 1. On relation to El-Mhamdi et al. (2020)
>
> We thank the reviewer for the reference. We will cite it and add the comparison to the next version of the paper. First of all, we use a more accurate claim in the discussion of the related work on decentralized approach to Byzantine robustness: the existing PS-free approaches either require full gradient computations, or peer-to-peer communications of the full vectors at each step, or have slow (theoretical) convergence guarantees.
>
> El-Mhamdi et al. (2020) do not provide non-asymptotic convergence rates making it impossible to provide a fair comparison with existing works and with our results as well. Therefore, it is unclear whether the usage of multiple servers speeds up the training or it just leads to overhead in the communications and computations. Indeed, in the described approach, the workers send their gradients to all servers at each iteration and receive parameters from all servers. This leads to a significant communication overhead in practice. Moreover, this method is based on median-based aggregation without additional verifications, which is proven to be a non-robust (vulnerable) aggregator in (Karimireddy et al., 2020).
>
> > 2. Since the algorithm requires n > 2f+1 then, in extreme cases, this attack leaves the system with only 1 node!
>
> Indeed, in the extreme case where almost 50% of peers are Byzantine, the training would progress with the same speed as with a single node. However, to the best of our knowledge, this is not worse than state-of-the-art results across Byzantine-tolerant systems. Most studies simply do not apply in this scenario (e.g. (El-Mhamdi et al., 2020) assumes that at most $1/3$ peers are Byzantine, (Karimireddy et al., 2020) needs at most $1/10$). As for the papers that support the same rate of Byzantines, they either operate at the same speed as one node or even slower.
>
> Based on the existing distributed computing projects such as Folding@home and cryptocurrencies, we estimate that the actual number of Byzantines is usually significantly smaller than 50%. For a system with almost 50% malicious peers, the fact that any benign peer leaving would compromise the entire system is usually a larger concern than the system's performance.
>
> > 3. Why does each peer require O(dn) in the PS architecture?
>
> We do not claim that each peer requires $O(dn)$ communication, but the centralized server does. As we increase the number of peers, the system will need $O(dn)$ time per step since the parameter server needs to ingest updates from $n$ peers. From another perspective, while a worker only needs to send and receive $O(d)$ data, its communication speed is inversely proportional to the total number of peers that share the bandwidth of the parameter server.
>
> > 4. The first sentence of Section 3.1 ("The core design principle behind our algorithm is that all types of Byzantine faults must have limited effect and a chance of being discovered.") is a bit weird. Usually, Byzantine attacks are posed as the strongest attack to diverge learning.
>
> Byzantine nodes can indeed do whatever they want: **no matter what they do**, BTARD ensures that Byzantine actions have an upper-bounded effect and a chance of being discovered. We thank the reviewer for noticing the potential misunderstanding and will fix it in the next version.
>
> > 5. It's not clear if clipping is effective against random attacks? I understand it will be robust against gradients with big magnitudes (like the attacks in the evaluation section) but what about the other attacks?
>
> This issue is addressed well in (Karimireddy et al., 2020). To summarize, clipping is indeed unable to detect low-magnitude attacks, so Byzantine-tolerant algorithms need to use additional checks. (Karimireddy et al., 2020) mitigates these attacks by using client momentum (“learns from history”). BTARD achieves the same results using validators (as described in Appendix C). Furthermore, as we summarize in Table 2, lower attack magnitude $\varepsilon$ results in better convergence guarantees.
>
> In short, a byzantine peer that performs a low-magnitude attack will be detected with probability 1 unless such peer stops after making a finite number of attacks. If a peer stops attacking before it is banned, the total "damage" from low-magnitude attacks is limited and will be negated by future steps.
>
> > 6. It's not clear how the butterfly method works?
>
> Please refer to "Li, Z., Davis, J. and Jarvis, S., 2017. An efficient task-based all-reduce for machine learning applications. In Proceedings of the Machine Learning on HPC Environments (pp. 1-8)".
>
> > 7. In Algorithm 2, a Byzantine can equivocate as follows. It can send different hashes (for the same part of the gradient) to different peers. … Is that a problem for your algorithm?
>
> The broadcast channel guarantees that Byzantine peers cannot send different hashes to different peers. We explain the broadcast channel properties in Appendix A.2.1.

---

> ### Author Response · Authors · 2021-11-11
> **Official Response to Reviewer 63QJ (Part 2/5)**
>
> > [of CenteredClip] I understand that the input to this algorithm is two gradients: the local one and the one gathered from all peers
>
> This is not the case. Each peer is responsible for aggregating a specific region of the gradient vector. CenteredClip is applied **inside** this procedure to aggregate $n$ vectors gathered from all peers (including the local vector). We refer to Algorithm 2 for details (here, step 6 shows that we compute CenteredClip using $n$ vectors previously received from peers).
>
> > One example is how the broadcast channels work in Section 2.3. The authors seem to describe consensus. Is that it? If yes, this concept is orthogonal to multi-party computation (MPC).
>
> As described in Section 2.3, broadcast channels are built on top of the protocols for Byzantine broadcast. While the problem of Byzantine broadcast is related to the consensus problem, they are not the same:
>
> In the consensus problem (e.g. as defined in [16]), each peer has its own initial value. All non-faulty peers must agree on the same (single) value. If all non-faulty peers have the same initial value, the agreed upon value must be equal to the initial ones.
> In the Byzantine broadcast problem, only one peer (the sender) has an initial value. All non-faulty peers must agree on its value. If the sender is non-faulty, the agreed upon value must be equal to the initial one.
>
> As explained in Section 2.3, we need broadcast channels to protect from malicious peers sending different values to each participant (thus addressing the issue from question 7).
>
> While Byzantine broadcast is not a part of MPC, many MPC protocols use it as a building block (e.g. [14, 15]), including the multi-party coin tossing protocol used in our method.
>
> > (1) it's very costly because all peers have to compute the gradient of the accused peer; a Byzantine can use this trick to make learning very slow
>
> While the Accuse procedure is indeed costly, it cannot be exploited efficiently to slow down training. As we describe in Section 3.1 and elaborate in Appendix C, the procedure is designed in such a way that it will always eliminate at least one Byzantine peer: either the accused peer or the accuser (in case of a false allegation).
>
> > (2) The procedure relies on consensus, which is also costly.
>
> BTARD does indeed rely on broadcasting (and hence, consensus of benign peers). However, an important feature of BTARD is that it only broadcasts scalar values whose size does not depend on the model size to minimize the communication overhead of broadcast protocols (see analysis in Appendix B). In practice, we have found that at our scales (16-64 participants) this operation can be performed in background and requires negligible extra communication when compared to averaging millions of trainable parameters.
>
> Moreover, both the broadcast and multi-party RNG protocols can be easily replaced with modern more scalable alternatives should the need arise. There are several decentralized systems [12, 13] that generate unpredictable shared random numbers every few seconds, which is sufficient for our case (Table 1 in [13] provides an overview of existing techniques including their guarantees, communication and computation costs). As for the broadcast channel, we discuss its scalable implementations in Appendix B.
>
> > the cost of sampling the random direction z among all peers.
>
> To generate a shared vector $z$ from a scalar seed $r$ (as defined in Appendix D), peers can use any standard pseudo-random number generator. Generating $z$ from a known seed has a negligible cost: for instance, generating one for ALBERT-large takes $30 \pm 1.2$ ms on the same T4 GPU that we use in our experiments. For the sake of completeness, we will include this in our complexity analysis.
>
> Below, we address the 27 points from the review in their original order.

---

> ### Author Response · Authors · 2021-11-11
> **Official Response to Reviewer 63QJ (Part 1/5)**
>
> We thank the reviewer for their detailed and high-effort review. Some of their questions led us to clarify parts of the explanation. Overall, the review acknowledges both the value of our theoretical results and the importance of scaling to large models.
>
> The reviewer also asks an impressive number of questions and we do our best to address them. We strongly believe that the reviewer's concerns can be addressed with only minor revisions to the paper. To keep our discussion organized, our response will follow the same structure as the original review.
>
> > public dataset assumption. Doesn't that make the problem trivial? Can you give examples in which this assumption is practical?
>
> We respectfully disagree. From a practical standpoint, training many state-of-the-art models (e.g. large language models) requires immense computational resources [1, 8]. Most practitioners simply do not have enough computing power to train this model in a reasonable time. However, these models are typically trained on large freely available datasets [6, 7], and there are several systems that allow training them by combining the effort of multiple independent parties or even volunteers [2, 3, 4, 5]. Most of these papers acknowledge that they need security against malicious participants.
>
> As we argue in Section 2.2, Byzantine-tolerant training in this setup is challenging because it has strict computation and communication constraints. A Byzantine-tolerant training strategy should not be slower than training the model independently on each peer. Many of our algorithmic choices were introduced specifically because we had to make it efficient.
>
> > Furthermore, many parts of the main algorithm require synchronous networks.
>
> More specifically, BTARD-SGD requires two points of synchronization per round: (1) before All-Reduce for gradient aggregation, as any synchronous SGD algorithm, and (2) after All-Reduce, for random number generation. Furthermore, the second synchronization can be performed in background while peers accumulate gradients for the next step. The third and final point of synchronization may arise if one peer accuses another of being Byzantine. However, such an accusation will result in at least one Byzantine peer being banned, so this extra synchronization can only happen a limited number of times.
>
> In practice, most computationally-intensive training pipelines often train with very large batches [9, 10, 11] and hence, relatively infrequent steps. Prior work [2] demonstrated that this alone can make it feasible to train with volunteers over the internet.
>
> That said, we agree that it is important to highlight the need for synchronization steps in the paper. We will upload a revised version of the paper that clearly addresses synchronization by November 20.
>
> > The third uncommon assumption is about adversary capabilities. I do not understand the need for the second paragraph on page 6 (which describes the possible attacks). Byzantines should be able to do whatever they want
>
> We do not put any restrictions on what Byzantines can do. Nevertheless, Byzantines can still be grouped into the specified categories depending on their behavior (whether they deviate from the protocol or not, whether they send correct gradients or not, etc.). This is intended to provide a high-level, intuitive explanation of why our algorithm is robust to all possible types of dishonest behavior.
>
> > There are essential parts in the paper that are not described clearly and left to the appendices.
>
> Indeed, we defer some important details to the appendix (e.g. detailed algorithm, parts of analysis, some experiments), and we are willing to defend our choice. For instance, while it would be great to have the detailed algorithm right away, it occupies 3.5 pages and would take up even more when we attempted to fit it in the main paper. Despite our best effort, it is impossible to make it more concise without sacrificing accuracy or clarity.
>
> For this reason, we chose to explain the intuition and core ideas of BTARD in the main paper, while deferring a more detailed description to the appendix. However, we do our best to add references from the main paper to the appropriate parts of the appendix.

---

### Official Review · Reviewer_w1BH · 2021-11-09

**Correctness:** 3
**Technical Novelty And Significance:** 3
**Empirical Novelty And Significance:** 2
**Recommendation:** 6
**Confidence:** 3

**Main Review:**

+The paper develops a careful protocol utilizing cryptographic primitives (MPC), unlike similar works that mainly aim to eliminate byzantine behavior by comparing the gradients provided by various nodes. The utilization of  MPC is novel, and the provable convergence guarantee is a strength of the paper.
+/- While the paper's protocols work only in the case where nodes are able to sample i.i.d. from the overall data set (the "homogenous" setting), the paper refers to recent lower bounds that preclude byzantine tolerant algorithms in the heterogenous setting (at least in the worst-case).
- The implementation of the cryptographic primitives to keep the byzantine behavior in check requires a number of all-to-all communication exchange. While the lack of a centralized trusted node is expected to make the protocols more expensive, a key missing ingredient of the paper is an explicit accounting for this communication cost. I believe it will greatly improve the contribution if this cost is explicitly characterized and reported.
- The algorithm explanation could improve in terms of clarity. For instance, the paper refer's to a butterfly all reduce procedure, which seems to suggest that each node receives the sum of all the numbers (gradients) that are transmitted in the step. However, the descriptions in Algorithms 1,2, only include "Send" and "Receive" primitives, and the algorithms in the appendix do not seem to do aggregation as a part of the communication primitives - it is done separately and explicitly in the centeredclip procedure. This can be explained more clearly. Note that the peer-to-peer communication (i.e., n^2 different messages being sent) also has an impact on the communication cost (referred to above).
- The paper seems to provide distinct converge behavior for the case where the number of byzantine adversaries is explicitly known. A clear/intuitive explanation of why the behavior is distinct can enhance the paper.


**Summary Of The Paper:**

The paper develops a byzantine resilient algorithm for distributed training via SGD without a trusted centralized node. The key idea is that it improves upon recent decentralized algorithms by using cryptographic primitives. In particular, a multiparty random number generator is used to randomly verify subsets of devices in each iteration, enabling the algorithm to learn byzantine behavior that deviates sufficiently from the protocol. The paper provides rigorous theoretical result in terms of convergence guarantee, with a rate that depends on the number of byzantine adversaries. The derived rate depends on the fraction of nodes that are byzantine (capped by 50%, which is standard in this literature). In cases where no node is adversarial, the derived convergence rate matches that of the standard algorithms.

**Summary Of The Review:**

Overall solid piece of work. It is worth considering the paper modulo addressing the comments above.

---

> ### Author Response · Authors · 2021-11-10
> **Official Response to Reviewer w1BH**
>
> We thank the reviewer for their feedback. We appreciate the positive evaluation of our work: Reviewer w1BH acknowledges the novelty of the proposed protocol, rigorousness of our theoretical results, and positions our submission as a “solid piece of work”.
>
> The reviewer also raises several questions and gives useful comments in their review. We address each of them below.
>
> > While the paper's protocols work only in the case where nodes are able to sample i.i.d. from the overall data set (the "homogenous" setting), the paper refers to recent lower bounds that preclude byzantine tolerant algorithms in the heterogenous setting (at least in the worst-case).
>
> We would like to emphasize that this is not a weakness of our approach but rather an inevitable limitation that we explicitly mention in the paper: Byzantine tolerance is impossible in the heterogeneous case (one cannot achieve any predefined accuracy of the solution of the optimization problem with heterogeneous functions available on the clients). We explain this result in Appendix E.2.
>
> Nevertheless, our paper addresses an important problem — it proposes the first scalable decentralized method robust to Byzantine attacks with rigorous convergence guarantees. Such a method is necessary for the setups where open collaborations (e.g. academic institutions and/or volunteers) pool their computational resources together to train a model that is too large to be trained by any of the participants alone. No prior work has proposed a method with these properties (as discussed in Section 2.2 and Appendix A1.2).
>
> > The implementation of the cryptographic primitives to keep the byzantine behavior in check requires a number of all-to-all communication exchange. While the lack of a centralized trusted node is expected to make the protocols more expensive, a key missing ingredient of the paper is an explicit accounting for this communication cost. I believe it will greatly improve the contribution if this cost is explicitly characterized and reported.
>
> We account for this communication cost in Appendix B. We will add references leading there to the main part of the paper to make it easier to find this appendix.
>
> > The algorithm explanation could improve in terms of clarity. For instance, the paper refer's to a butterfly all reduce procedure, which seems to suggest that each node receives the sum of all the numbers (gradients) that are transmitted in the step. However, the descriptions in Algorithms 1,2, only include "Send" and "Receive" primitives, and the algorithms in the appendix do not seem to do aggregation as a part of the communication primitives - it is done separately and explicitly in the centeredclip procedure. This can be explained more clearly.
>
> We recognize that the algorithm is complex and are willing to incorporate any suggestions on improving the clarity of the explanation.
>
> We would like to clarify that All-Reduce protocols (including Butterfly All-Reduce) are special communication-efficient protocols that allow a group of peers to collectively aggregate (e.g. average) vectors stored on each peer (as explained in Section 2.1). The building blocks of such protocols are “Send” and “Receive” operations involving peer-to-peer communication (without any local computations on the machines). Next, since naive averaging is not robust to the Byzantines, we replace the average computation in Butterfly All-Reduce with the computation of a robust estimator called CenteredClip (these computations do not involve any communication themselves). The output of CenteredClip is computed on each machine using the information from other workers. We reflect these details in Algorithms 2 and 4 and elaborate on other building blocks of BTARD and BTARD-SGD in Appendix D.
>
> > The paper seems to provide distinct converge behavior for the case where the number of byzantine adversaries is explicitly known. A clear/intuitive explanation of why the behavior is distinct can enhance the paper.
>
> We thank the reviewer for the feedback and will add a small clarification to Section 3.2. The case when the number of attacking peers is known at each step is simpler since one can explicitly use this knowledge (i.e., $\delta$ at each step) to better adjust the clipping level in CenteredClip. When the exact $\delta$ is used at each iteration instead of its upper bound, CenteredClip returns more accurate estimates of the mean. This fact explains why the convergence guarantees are better in the case when the number of attacking peers is known at each iteration. However, this is an unrealistic assumption and we provide the convergence results in this case for the sake of scientific curiosity. The discovered discrepancy between the two cases highlights the importance of designing new robust aggregation rules independent of any parameters. This is an interesting direction for future work.

---

### Official Review · Reviewer_yAgj · 2021-11-12

**Correctness:** 3
**Technical Novelty And Significance:** 2
**Empirical Novelty And Significance:** 2
**Recommendation:** 5
**Confidence:** 3

**Main Review:**

While there are a large number of papers on Byzantine robustness in the parameter server setting, there are relatively few papers for the decentralized setting. In this sense, the paper is timely and relevant. However, the paper lacks in giving sufficient details, which makes it difficult to understand the contributions and novelty. Detailed comments are as follows.

Major comments:

1. How can one ensure that the policy on page 6 where a worker can eliminate one other worker along with themselves, cannot be exploited by Byzantine workers? Can it happen that a Byzantine worker b eliminates an honest worker h, and ensures that another honest worker h’ eliminates b. In this case, (b,h) and (b,h’) will be eliminated. In other words, Byzantine worker b could eliminate two honest workers. It needs a proof that this type of attacks are not possible. Moreover, since the setup is decentralized how consensus happens to determine if a pair of workers to be eliminated. It will be important to give more details.

2. What is the computation and communication complexity at each peer for BTARD? Validators need to recompute the gradients, and in addition, in case if a peer accuses another peer, all other peers need to recompute the gradient. This suggests huge computation overhead. Would this recalculation of gradients outweigh the benefits of parallelism? It is crucial to quantify computation and communication costs explicitly.

Moreover, how much additional communication cost is incurred by using secure broadcast and MPRNG? Appendices A.2.1 and A.2.2 treat these primitives at a very high level. It would be helpful to select a suitable protocol from the literature, and give enough details.

3. In experiments, baseline methods include coordinate-wise median and geometric median, How are these computed in the decentralized setting? It will be important to give more details.

4. How much improvement the proposed BTARD algorithm gives as compared to baseline methods? From Figures 2 and 3, it seems like the improvements are very marginal. Can the authors quantify the improvements in terms of percentage increase in accuracy (or decrease in loss), and add a discussion on comparison?

5. The authors mention that they omit common (and powerful) attacks in [Baruch et al. 2019, Xie et al. 2020, Allen-Zhu et al. 2021] as these attacks are designed to bypass certain checks, which are not used by the proposed algorithm. The proposed algorithm does not use variance and magnitude checks does not mean that these attacks will fail. These attacks were proposed to show inefficiency of several well-known Byzantine robust schemes such as Krum and coordinate-wise median. So, it will be important to test BTARD against some of these attacks.

Other comments:

1. The paper assumes that training is performed on a public dataset, which can be accessed by all peers. It would be helpful to justify this assumption. In the Byzantine robustness literature, many papers consider data parallelism, where data is distributed across peers. In such a setup, it would not be possible to validate by recalculating gradient. It would be helpful to add a discussion on this assumption.

2. The paper does not formally introduce the decentralized training setup being considered. How many peers in total, and how many are Byzantine? Can any peer talk with any other peer? Can Byzantine peers collude? A few details are given in Sec. 3.2 after the proposed methods. It will be important to describe these details up front.

3. The authors do not give any description or details about Algorithms 1-3 in the main text. It is difficult to understand the algorithms, and takes a long time for the reader.


**Summary Of The Paper:**

The paper considers the problem of decentralized training in the presence of Byzantine peers. The paper proposes a Byzantine-tolerant algorithm which uses randomly chosen peers as validators. A validator recomputes the gradient and ‘accuses’ a peer if the computation is not correct. Towards this end, the paper proposes to leverage two primitives from secure Multi-Party Computation (MPC) literature: secure broadcast and multi-party random number generator. For robust aggregation, the paper relies on Centered Clipping proposed recently in [Karimireddy et al. 2020].

**Summary Of The Review:**

The paper considers a challenging problem of Byzantine-tolerant learning in the decentralized setup (without a parameter server). However, the paper lacks in many details and does not quantify costs and gains of the proposed algorithm.

---

> ### Author Response · Authors · 2021-11-17
> **Official Response to Reviewer yAgj (Part 2/2)**
>
> > 4. How much improvement the proposed BTARD algorithm gives as compared to baseline methods? From Figures 2 and 3, it seems like the improvements are very marginal. Can the authors quantify the improvements in terms of percentage increase in accuracy (or decrease in loss), and add a discussion on comparison?
>
> The plots comparing BTARD with the "vanilla" All-Reduce and other aggregation techniques demonstrate their convergence **without attacks**. They are intended to show that the difference (i. e., a **negative effect** of BTARD on the model's quality and convergence speed) **is marginal** compared to the "vanilla" All-Reduce. Indeed, BTARD with $\tau = 1$ (stronger clipping) reaches only 0.6% smaller accuracy in Figure 2, and BTARD-Clipped-SGD with $\tau = 0.125$ makes loss even better (2.3% smaller) in Figure 3.
>
> These results are already discussed in the last paragraphs of Section 4.1 and 4.2: _"we note that our method does not worsen the speed of convergence but introduces a minor negative effect on the final test accuracy"._ We will report the specific numbers there and improve the captions of the convergence plots to clarify that there are no attacks involved.
>
> The plots demonstrating the attacks do not include the "vanilla" All-Reduce and the coordinate-wise/geometric median baselines because they were shown to be vulnerable in Karimireddy et al. (2020). In turn, while CenteredClip with PS and client momentum is Byzantine-robust, it does not converge in our setting with almost half of all peers being Byzantine. In particular, experiments in Karimireddy et al. (2020) only consider 20% Byzantine peers for the same attack types.
>
> > 5. The authors mention that they omit common (and powerful) attacks in [Baruch et al. 2019, Xie et al. 2020, Allen-Zhu et al. 2021] as these attacks are designed to bypass certain checks, which are not used by the proposed algorithm. The proposed algorithm does not use variance and magnitude checks does not mean that these attacks will fail.
>
> We agree and aim to provide additional experiments by November 20th.
>
> Next, we respond to the other comments by the reviewer:
>
> > 1. The paper assumes that training is performed on a public dataset, which can be accessed by all peers. It would be helpful to justify this assumption. In the Byzantine robustness literature, many papers consider data parallelism, where data is distributed across peers.
>
> We justify this assumption in footnote 4 (page 6) and discuss it in Appendix E.2 in detail. The paper referred there proves that one cannot achieve any predefined accuracy of the solution of the optimization problem with heterogeneous functions available on the clients even if the dataset dissimilarity is bounded. Although the authors of that paper show that under the additional assumption holding for some over-parameterized models one can circumvent the given lower bound, the corresponding result typically implies that $\delta$ should be too small.
>
> Therefore, we emphasize that this assumption is not a weakness of our approach but rather an **inevitable limitation** that we explicitly state in the paper.
>
> > 2. The paper does not formally introduce the decentralized training setup being considered. How many peers in total, and how many are Byzantine? Can any peer talk with any other peer? Can Byzantine peers collude? A few details are given in Sec. 3.2 after the proposed methods. It will be important to describe these details up front.
>
> In Section 3.2, we state that the share of Byzantines $\delta < 1/2$. Further, we state that we assume Byzantines to be omniscient as defined in Karimireddy et al. (2020).
>
> We recognize that describing our training setup and the attacker model is important, so we will move the paragraph describing that to the beginning of Section 3.1 in the nearest update. Also, we will explicitly clarify that peers can talk with any other peer and Byzantines can collude.
>
> > 3. The authors do not give any description or details about Algorithms 1-3 in the main text. It is difficult to understand the algorithms, and takes a long time for the reader.
>
> We describe and motivate each building block of Algorithm 1 in Section 3.1. This includes paragraphs dedicated to ButterflyClip (Algorithm 2) and the Accuse procedure (Algorithm 3).
>
> We agree that the paper would benefit from having more explanations and a detailed algorithm right away. However, the detailed algorithm occupies 3.5 pages, so we were unable to fit all that in the main paper due to the space constraints without sacrificing accuracy or clarity. Because of that, we chose to explain the intuition and core ideas of BTARD in the main paper, deferring the details to Appendix D. We did our best to add references from the main paper to the appropriate sections of the appendix.

---

> > ### Comment · Reviewer_yAgj · 2021-11-21
> > **Responses were pretty helpful, but there are a few more questions.**
> >
> > Thanks to authors for their responses, and for the updated paper. This addresses several of my questions. I still have some more questions:
> >
> > 1. In the notation used in Appendix B, what is the communication cost per user for the standard all-reduce SGD? For BTARD protocols, I am confused how the communication cost is O(d + n^3) and not O(dn + n^3)? Each peer needs to send their model updates to all other peers, right? Can the authors give more details?
> >
> > 2. How do the authors compute 10% computation overhead (and 13% compute overhead for low-magnitude attacks)? Even if each peer selects one validator, the computation cost is doubled as compared to all-reduce SGD, right? Can the authors please provide more details on this?
> >
> > 3. The paper evaluates the performance of BTARD protocols on known attacks, but there are newer attacks possible which exploit the BTARD algorithm. Consider the case of 16 peers, with 7 Byzantine peers as considered in the Experiments. Just by accusing one honest peer in a coordinated manner, it seems like Byzantine peers can eliminate 7 honest peers in the first few training rounds. In this way, the system only works with 2 (honest) peers in total for the rest of the rounds. This would slow down the rate of convergence since the effective total batch size is only twice the local batch size (as opposed to 9 times the local batch size, if Byzantine peers simply send all-0 gradient as an example of a low-impact attack). How would the rate of convergence compare empirically with the baselines considered in this case?

---

> > > ### Author Response · Authors · 2021-11-22
> > > **Reply to other questions by Reviewer yAgj**
> > >
> > > Thank you for the feedback.
> > >
> > > > 1. In the notation used in Appendix B, what is the communication cost per user for the standard all-reduce SGD? For BTARD protocols, I am confused how the communication cost is O(d + n^3) and not O(dn + n^3)? Each peer needs to send their model updates to all other peers, right?
> > >
> > > Standard bandwidth-optimal versions of All-Reduce SGD (including the ones based on Butterfly All-Reduce) involve the communication cost of only $O(d)$ **per user** at each step. We mention that at the end of Section 2.1. For instance, Butterfly All-Reduce works as follows:
> > >
> > > 1. Each peer splits its gradient vector (of size $d$) into $n$ equal parts (of size $O(d / n)$ each).
> > > 2. Each peer sends the $i$-th part of its vector to peer $i$ (that is, it sends the 1st part to peer 1, the 2nd part to peer 2, etc.).
> > > 3. At this point, if a peer has a number $j$, it has received the $j$-th parts of all gradient vectors from other peers. The peer averages them to obtain the $j$-th part of the aggregated gradient vector and sends the result back to all other peers.
> > > 4. Each peer receives the $i$-th part of the aggregated gradient vector from peer $i$ (that is, the 1st part from peer 1, the 2nd part from peer 2, etc.).
> > > 5. Each peer merges all received parts to obtain the full aggregated gradient vector.
> > >
> > > In this protocol, each peer only sends $n$ source vectors of size $O(d / n)$ at step 2 and $n$ resulting vectors of size $O(d / n)$ at step 3. Also, each peer receives the same amount of data. Therefore, the total communication cost is $O(d)$ per user.
> > >
> > > To achieve scalability and communication efficiency, BTARD uses a similar procedure, where the naive averaging at step 3 is replaced with CenteredClip (we describe the resulting procedure in Section 3.1, in particular, in Algorithm 2). Therefore, this part of the algorithm has the communication cost of only $O(d)$ per user as well.
> > >
> > > > 2. How do the authors compute 10% computation overhead (and 13% compute overhead for low-magnitude attacks)? Even if each peer selects one validator, the computation cost is doubled as compared to all-reduce SGD, right?
> > >
> > > If we consider the setup with $m$ validators, the whole system **jointly chooses** $m$ validators at each training step. This is done using MPRNG that allows us to generate **common unpredictable random numbers** for the whole system. The $m$ peers chosen to serve as validators do not need to compute their own gradients (i.e., their computing power is dedicated to validation). We state that in paragraphs 3-4 of Section 3.1.
> > >
> > > Thus, if we have $n$ peers in total, we dedicate only the share $\frac{m}{n}$ of the total computing power for validation. This is $\frac{1}{16} = 6.25\\%$ for the attacks from Section 4 and $\frac{2}{16} = 12.5\\%$ for the worst-case low-magnitude attack from Appendix I.4.
> > >
> > > > 3. [...] Consider the case of 16 peers, with 7 Byzantine peers as considered in the Experiments. Just by accusing one honest peer in a coordinated manner, it seems like Byzantine peers can eliminate 7 honest peers in the first few training rounds. In this way, the system only works with 2 (honest) peers in total for the rest of the rounds.
> > >
> > > Such attacks are indeed possible (we mention them as "reputation attacks" in Section 3.1 and discuss them in Appendix C). In the extreme case with almost half of the peers being Byzantine, the convergence would indeed slow down empirically (in terms of wall time), and the training will progress with the same speed as with 1-2 nodes.
> > >
> > > However, to the best of our knowledge, this is not worse than state-of-the-art results across Byzantine-tolerant systems. Most studies simply do not apply in this extreme scenario (e.g. El-Mhamdi et al. (2020) assume that at most $\delta = 1 / 3$ of peers are Byzantine, and Karimireddy et al. (2020) need $\delta \le 1 / 10$). Other papers supporting $\delta$ close to $1 / 2$ operate at the same speed as one node or even slower.
> > >
> > > Moreover, based on the existing distributed computing projects such as Folding@home and cryptocurrencies, we estimate that the actual number of Byzantines is usually significantly smaller than 50%. For a system with almost 50% malicious peers, the fact that any benign peer leaving would compromise the entire system is usually a larger concern than the system's performance.
> > >
> > > Finally, our paper describes a heuristic protocol allowing new peers to join the training (see Section 3.3 and Appendix F). It allows inviting new participants to contribute to the training if the collaboration decides that the remaining resources are not enough. This procedure ensures that even if the attackers manage to re-join, the system will be safe from Byzantine and Sybil attacks as long as honest peers hold more computing power than the attackers.

---

> > > > ### Comment · Reviewer_yAgj · 2021-11-22
> > > > **Thanks for the clarifications**
> > > >
> > > > Thanks for responding to my questions. I understand the communication costs. However, each peer as well as validator(s) also need to download data samples from the public dataset, right? This will add to the communication costs. Of course, the number of model parameters $d$ is often much larger than the size of a data sample. Having said that, it would be important to mention this point.
> > > >
> > > > When $m$ peers and $m$ validators are chosen in each step, it is not clear if it is enough to simply assume that Byzantine peers will ‘eventually’ be detected. Byzantine peers can hamper convergence until they get  detected. I think the authors are implicitly assuming that the number of rounds is much larger than $n$. This may be a natural assumption, but needs to be implicitly stated if indeed such an assumption is needed. For example, suppose the case that 1 peer and 1 validator are selected. Consider an attack in which one Byzantine peer sends incorrect gradient with probability p, and other Byzantine peers send correct gradients but would not correctly validate this malicious peer if chosen as a validator. Then, an analysis on the probability with which the Byzantine peer would be detected is needed. I think this would roughly be O$((1-\delta)p/n)$, which could be very small.
> > > >
> > > > How does the theoretical convergence rate(s) compare to the decentralized SGD (without Byzantine attacks)? When reputation attack is considered, the theoretical convergence rate would be equal to that of the decentralized SGD with $(1-2\delta)$ peers (all honest). I wonder if this is reflected in the convergence rate expressions.
> > > >
> > > > Overall, the proposed protocols are interesting but they seem to crucially rely on the assumption that each peer can access the entire dataset. This enables the protocols to use honest users as validators, which is not possible in the more conventional distributed/decentralized learning setup with data parallelism. This key assumption needs to be explicitly and clearly stated in the Abstract and Introduction; it looks like this assumption is currently mentioned only in Sec. 3. (Otherwise, the reader only reading abstract and introduction is likely to get an impression that the paper gives protocols for Byzantine-robust decentralized learning in the more conventional setting.)

---

> > > > > ### Author Response · Authors · 2021-11-25
> > > > > **Response to Reviewer yAgj (Part 2/2)**
> > > > >
> > > > > > How does the theoretical convergence rate(s) compare to the decentralized SGD (without Byzantine attacks)? When reputation attack is considered, the theoretical convergence rate would be equal to that of the decentralized SGD with $(1 - 2  \delta)$ peers (all honest). I wonder if this is reflected in the convergence rate expressions.
> > > > >
> > > > > Thank you for the question. First of all, to prevent possible misunderstanding, we would like to notice that our theoretical results for BTARD-SGD are closer to the ones known for **Parallel SGD** (with parameter server), since it has the same convergence guarantees as **All-Reduce SGD**. In turn, the name **Decentralized SGD** refers to Gossip SGD in optimization literature (e.g., see the details in [1] from the references below). As we explain below, Gossip SGD and Parallel SGD have different complexity guarantees.
> > > > >
> > > > > Next, our theoretical results hold for all types of attacks **including the reputation attacks**. In the particular case of reputation attack, the analysis can be shown to recover the rate of Parallel SGD with $(1-2\delta)n$ honest peers. Since in our theoretical analysis we assume that $\delta \leq 0.1(n-m)$ (e.g., see Theorems E.2 and E.3 for the analysis in the non-convex case), the reputation attack does not spoil the complexity results (up to small numerical factors). Therefore, in this regime, the **reputation attack is not the worst one** in terms of the effect on the convergence. We note that our assumption on $\delta$ used in the theoretical analysis is still reasonable and can hold in practice. SOTA results like ones from Karimireddy et al. (2020) use similar bounds on $\delta$.
> > > > >
> > > > > Finally, we notice that in all considered cases the first two terms of our complexity results for BTARD-SGD (see Table 1 for the summary) match the complexity of Parallel SGD without Byzantine attacks. Moreover, when $\varepsilon$ is small enough we recover the complexity of Parallel SGD without Byzantine attacks as well since the third terms in our rates have better dependence on $\varepsilon$ than the second ones. Since Parallel SGD has better convergence guarantees than Decentralized SGD, our method is theoretically superior to Decentralized (Gossip) SGD in these setups. To be precise, let us consider non-convex problems under uniformly bounded variance assumption. In this case, the complexity of Parallel SGD is (in the notation from Table 1)
> > > > >
> > > > > ${\mathcal{O}}\left( \frac{L\Delta_0}{\varepsilon^2} + \frac{L\Delta_0\sigma^2}{n\varepsilon^4} \right),$
> > > > >
> > > > > while Decentralized SGD has the complexity (in the homogeneous case)
> > > > >
> > > > > ${\mathcal{O}}\left( \frac{L\Delta_0}{\varepsilon^2}  + \frac{L\Delta_0\sigma^2}{n\varepsilon^4} + \frac{L\Delta_0\sigma}{\sqrt{p}\varepsilon^3} \right),$
> > > > >
> > > > > where $p$ is the spectral gap of the communication graph (e.g., see the details in [1]). The quantity $p$ depends on the communication graph. When each node has $O(1)$ neighbors, $p \sim \frac{1}{n}$ (2-d torus network), so, the third term is proportional to $\sqrt{n}$ in this case. In contrast, the complexity of our BTARD-SGD (when the number of attacking peers is not known) is
> > > > >
> > > > > ${\mathcal{O}}\left( \frac{L\Delta_0}{\varepsilon^2}  + \frac{L\Delta_0\sigma^2}{n\varepsilon^4} + \frac{n^2\delta\sigma^2}{m\varepsilon^2} \right).$
> > > > >
> > > > > As one can see from the above comparison, the third term in our complexity bound has better dependence on $\varepsilon$ than the corresponding term in the complexity bound for Decentralized SGD by a large factor $\frac{1}{\varepsilon}$. This means that for small enough $\varepsilon$ BTARD-SGD has better complexity than Decentralized SGD. Moreover, if $n/m = {\mathcal{O}}(1)$, $\delta = {\mathcal{O}}(n^{-1/2})$ (number of Byzantines is not larger than $\sqrt{n}$), and $\sigma \sim L\Delta_0$, then the third term in the complexity bound for BTARD-SGD is better than corresponding term from the complexity bound of Decentralized SGD for any $\varepsilon$.
> > > > >
> > > > > A similar comparison can be conducted for convex and strongly convex cases as well. We will add these details to the final version of the paper.
> > > > >
> > > > > > Overall, the proposed protocols are interesting but they seem to crucially rely on the assumption that each peer can access the entire dataset. [...] This key assumption needs to be explicitly and clearly stated in the Abstract and Introduction; it looks like this assumption is currently mentioned only in Sec. 3.
> > > > >
> > > > > Thank you for pointing this out. We have updated our paper so that the introduction explicitly states this assumption several times. Currently, we are not able to modify the abstract, but we will state this assumption there as well in case our paper gets accepted.
> > > > >
> > > > > **References:**
> > > > >
> > > > > [1] Koloskova, A., Loizou, N., Boreiri, S., Jaggi, M., & Stich, S. (2020, November). A unified theory of decentralized SGD with changing topology and local updates. In International Conference on Machine Learning (pp. 5381-5393). PMLR.

---

> > > > > > ### Comment · Reviewer_yAgj · 2021-11-30
> > > > > > **Thanks, I am changing my score in response to the discussion**
> > > > > >
> > > > > > Thanks for the responses. It would be useful to add the details on distributed vs parallel vs decentralized setups with appropriate references as [1] above. (BTW, I am not sure if a peer can generate data on the fly as it would not be possible to share the data with a validator.)
> > > > > >
> > > > > > Based on the discussion with the authors, I am raising my score from 3 to 5. The paper essentially considers Byzantine-robust distributed learning (without parameter server) on public datasets where each peer can access the entire dataset. There does not seem to be any other paper considering this particular  setup. It would be good not to oversell the paper by comparing the contributions against conventional distributed (with or without parameter server, parallel, and decentralized) setups using data parallelism. The authors have revised the manuscript to respond to my concerns as well as those from the other reviewers. However, I believe it would be worth substantially revising the paper to clearly mention the setup and key techniques. Experiments showing concrete costs in terms of number of bytes communicated and/or wall-clock time would significantly improve the contributions.

---

> > > > > ### Author Response · Authors · 2021-11-25
> > > > > **Response to Reviewer yAgj (Part 1/2)**
> > > > >
> > > > > Thank you for the feedback.
> > > > >
> > > > > > However, each peer as well as validator(s) also need to download data samples from the public dataset, right? This will add to the communication costs. Of course, the number of model parameters d is often much larger than the size of a data sample. Having said that, it would be important to mention this point.
> > > > >
> > > > > Downloading data samples on each step is indeed one of the options, but not the only one. In other cases, peers can download all data before training or generate the data on the fly (as long as it is i.i.d.).
> > > > >
> > > > > We do not restrict the way of getting the data and do not consider it a part of our algorithm, leaving the related communication and computation costs out of the scope of our analysis. We have clarified that in Appendix B.
> > > > >
> > > > > > When $m$ peers and $m$ validators are chosen in each step, it is not clear if it is enough to simply assume that Byzantine peers will ‘eventually’ be detected. Byzantine peers can hamper convergence until they get detected. I think the authors are implicitly assuming that the number of rounds is much larger than $n$. This may be a natural assumption, but needs to be implicitly stated if indeed such an assumption is needed.
> > > > >
> > > > > Let us clarify this place. If a Byzantine peer deviates from the protocol, this can be detected either instantly or by an honest validator. The first case is trivial for the analysis, so, let us consider the second one without loss of generality. If a Byzantine peer deviates from the protocol at iteration $k$, this deviation/Byzantine attack is revealed with a certain probability larger than $m/n$ as we explain in our proofs, e.g., in the proof of Theorem E.2 (Appendix E.3.3, page 35, the text _“If a Byzantine peer deviates from the protocol ...”_ and what follows after that) or in other proofs of our convergence results. Therefore, in expectation, the total number of protocol deviations before the ban is upper bounded by $n/m$ for each peer and for any possible strategy of the attacks (including the one described by the reviewer). The upper bound does not depend on the behavior of Byzantine peers and can be made small enough via increasing $m$.
> > > > >
> > > > > Next, we emphasize that **we do not assume that the total number of iterations $K$ is larger than $n$**: it can be any positive number. However, if the goal is to find the solution of optimization problem with accuracy $\varepsilon$ (where $\varepsilon$ depends on the assumption about convexity of the problem), then the required number of iterations does depend on $n$ explicitly and we do not hide it in our bounds: the third terms in our complexity bounds (see Table 1) are either proportional to $n$ or $n^2$. However, if $\varepsilon$ is not too small, $m$ is such that $n/m$ is not large, and $\sigma^2$ is small enough, then our upper bounds on $K$ do not imply that $K \geq n$. Overall, our bounds contain all the required information about the complexity of the algorithm and do not hide any dependencies except dependencies on the numerical constants.
> > > > >
> > > > > > For example, suppose the case that 1 peer and 1 validator are selected. Consider an attack in which one Byzantine peer sends incorrect gradient with probability p, and other Byzantine peers send correct gradients but would not correctly validate this malicious peer if chosen as a validator. Then, an analysis on the probability with which the Byzantine peer would be detected is needed. I think this would roughly be $O((1 - \delta) p / n)$, which could be very small.
> > > > >
> > > > > In this example, the Byzantine can indeed be detected only with **full** probability $O((1 - \delta) p / n)$ at each step. However, as we explain above, the conditional probability of being detected is lower bounded by $m/n$ when a Byzantine peer violates the protocol. Therefore, the theoretical guarantees stated above hold for this example as well.
> > > > >
> > > > > Informally, in BTARD, Byzantine attacks at one step have only limited effect (due to our clipping approach), so sending wrong gradients with probability $p < 1$ decreases the attack strength. We demonstrate this empirically in Figure 6 from Appendix I.1 that corresponds to $p = 0.1$ (one can compare it with Figure 2 that corresponds to $p = 1$). The attack strength is further decreased if only a small share of Byzantines send wrong gradients (as demonstrated in Figure 7).
> > > > >
> > > > > As a result, one peer sending wrong gradients with a small probability $p$ just **cannot significantly harm the convergence**, even though it may remain undetected for a long time when only 1 validator is used.

---

> ### Author Response · Authors · 2021-11-17
> **Official Response to Reviewer yAgj (Part 1/2)**
>
> We thank the reviewer for their feedback. We appreciate them noting that a problem of Byzantine robustness in a decentralized setting is "timely and relevant" and giving comments on improving the paper's clarity.
>
> We begin with responding to the major comments:
>
> > 1. How can one ensure that the policy on page 6 where a worker can eliminate one other worker along with themselves, cannot be exploited by Byzantine workers? Can it happen that a Byzantine worker b eliminates an honest worker h, and ensures that another honest worker h’ eliminates b. In this case, (b,h) and (b,h’) will be eliminated. In other words, Byzantine worker b could eliminate two honest workers. It needs a proof that this type of attacks are not possible. Moreover, since the setup is decentralized how consensus happens to determine if a pair of workers to be eliminated.
>
> Both $\text{Accuse}(i, j)$ and $\text{Eliminate}(i, j)$ imply that peer $i$ uses the broadcast channel to declare its intent to ban peer $j$. All peers collect such messages during a training step and process them at the end of the step in some specific order (e.g. sorted by $(\text{type}, \text{public\\_key}_i, \text{public\\_key}_j)$, where $\text{type} \in \\{\text{Accuse}, \text{Eliminate}\\}$ and $\text{Accuse} < \text{Eliminate}$). If processing one of the messages results in banning peer $p$, further messages involving $p$ are ignored regardless of the $p$'s role. This way, it is impossible for a Byzantine to eliminate more than one honest peer along with itself. Peers reach consensus since their decisions on banning someone are based solely on the messages from the broadcast channel (sorted in the common order) and the calculations with identical results.
>
> We agree that these details are important and will add them to Appendix D.2 in the nearest update of the paper.
>
> > 2. What is the computation and communication complexity at each peer for BTARD? Validators need to recompute the gradients, and in addition, in case if a peer accuses another peer, all other peers need to recompute the gradient. This suggests huge computation overhead. Would this recalculation of gradients outweigh the benefits of parallelism?
>
> We analyze the communication complexity and computational overhead in Appendix B (we will add references leading there to make it easier to find this appendix).
>
> In all setups from our experiments, BTARD withstands all attacks with a relatively small share of peers (1/16) chosen to be validators. As such, the validators' computation overhead is under 10% during normal training. In turn, accusing another peer indeed leads to all peers recomputing the gradients. However, such an accusation always results in at least one Byzantine being banned (as described in Section 3.1, paragraph 4), so these extra computations are performed only a limited number of times. Therefore, our computational overhead is much less than the benefits of parallelism in practical scenarios.
>
> > Moreover, how much additional communication cost is incurred by using secure broadcast and MPRNG? Appendices A.2.1 and A.2.2 treat these primitives at a very high level. It would be helpful to select a suitable protocol from the literature, and give enough details.
>
> We estimate the BTARD communication complexity in Appendix B. To obtain this estimate, we discuss the communication complexity of a suitable broadcast channel protocol from Dolev & Strong (1983). Also, we mention other results that may be more efficient in practice.
>
> Next, we provide details on a suitable MPRNG protocol from Blum (1983) in Figure 4. It shows that this protocol requires each peer to only broadcast 3 scalars, so its communication cost is marginal (compared to sending at least tens of millions of trainable parameters).
>
> We will extend Appendices A and B with a text description of the protocol from Figure 4 and an explicit discussion of its communication cost.
>
> > 3. In experiments, baseline methods include coordinate-wise median and geometric median, How are these computed in the decentralized setting?
>
> In this case, we use centralized baselines. We will clarify that in Section 4.2.

---

### Author Response · Authors · 2021-11-17
**General Response to Reviewers**

We thank the reviewers for taking the time to study our work and give the corresponding feedback.

We appreciate the reviewers acknowledging the value of our results, including the relevance of Byzantine-robustness in a decentralized setting (yAgj), the importance of scaling to large models (63QJ), the novelty of our approach (w1BH, 38p5), and the value of our theoretical results (w1BH, 63QJ). Reviewer w1BH describes our submission as a _“solid piece of work”_, and Reviewer 38p5 writes that it is _"well written and easy to follow"_.

We did our best to respond to reviewers' questions and concerns. Also, we have prepared an updated version of the paper with minor changes improving clarity, fixing potential misunderstandings, and extending discussions provided in the appendix. This version is now available in OpenReview (new additions are highlighted in green).

We summarize the changes below and mention reviewers who requested them. Note that **footnote 4** (mentioned in the discussions) **became footnote 2**, since we have moved the paragraph detailing our assumptions to the beginning of Section 3.1.

**Changes in the main part of the paper**

- **(yAgj, w1BH)** We added a reference to Appendix B "Network and compute overhead of BTARD-SGD" (Section 3.1).
- **(yAgj, 63QJ)** We clarified that we use the centralized versions of the coordinate-wise and geometric median baselines (Section 4.2).
- **(yAgj, 63QJ)** We explicitly clarified that the upper-left plots in Figures 2 and 3 show convergence without attacks and quantified the difference between BTARD and the baselines (Section 4).
- **(yAgj, 63QJ)** We moved the paragraph detailing our setup and assumptions to the beginning of Section 3.1 and explicitly clarified that each peer can _"communicate with any other peer"_ and Byzantines _"can arbitrarily deviate from our algorithm"_ and _"are able to collude with each other"_.
- **(w1BH)** We explained the reasons for the distinct convergence behavior under the assumption that participants know the number of Byzantines attacking on each iteration (bottom of page 6).
- **(63QJ)** We cited El-Mhamdi et al. (2020) in Section 2.2 (and added a detailed comparison to Appendix A.1.2).
- **(63QJ)** We removed the mentions of federated learning and improved the usage of the MPC-related terminology.
- **(63QJ)** We clarified that (a) the validators and the validated peers form a one-to-one mapping and (b) peers chosen to be validators do not need to compute their own gradients (Section 3.1).

**Changes in the appendix**

- **(yAgj)** We added details on Accuse() and Eliminate() implementation (Appendix D.2).
- **(yAgj)** We added a text description of the suitable MPRNG protocol from Figure 4 and an explicit discussion of its communication cost (Appendices A.2.2 and B).
- **(63QJ)** We added a discussion on the BTARD's synchronization points and the cost of sampling the random direction $z$ using MPRNG (Appendix B).
- **(63QJ, ojVo)** We added formal definitions for Split, ValidatePeer, Ban, and other auxiliary functions (Appendix D.1).
- **(63QJ)** We explained the effect of $\tau$ from Equation (2) in Appendix D.1.

**On additional experiments**

We also agreed to provide the experiments extending the set of attacks tested against BTARD.

We are currently running these experiments and aim to provide the results by November 20. We kindly ask the reviewers to provide feedback on our responses and the new version of the paper in the meantime.

---

### Author Response · Authors · 2021-11-21
**Results of Additional Experiments**

Dear reviewers,

We have extended the set of attacks tested against BTARD with the SOTA low-magnitude attacks from Baruch et al. (2019), Xie et al. (2020), Allen-Zhu et al. (2021), as requested by Reviewers yAgj and 63QJ.

We report the new results and discuss them in Appendix I.4 in the latest version of our paper (we also added a reference leading there from the main part of the paper).

We kindly ask you to provide feedback on our responses and the new version of the paper.

---

### Decision · Program_Chairs · 2022-01-20

**Decision:**

Reject

**Comment:**

This paper presents promising and ambitious work in the context of Byzantine-tolerant learning in the decentralized setup. The reviews raised several critical points of concern: completeness of technical derivations and details, experimental results and comparisons to other work, excluding several state of the art attacks. The reviewers provided with a generous amount of feedback, that should be incorporated in the paper before publication. Unfortunately the camera ready timeline is quite sort for such an extensive feedback to be integrated in this paper.  The authors are urged to resubmit after taking into account all the suggestions that were made during this review cycle.